# Global high-resolution estimates of the UN Human Development Index using satellite imagery and machine learning

Luke Sherman [1,6], Jonathan Proctor [2,6] ✉, Hannah Druckenmiller [3,4], Heriberto Tapia[5] & Solomon Hsiang [1,4]

The United Nations Human Development Index, which incorporates income, education and health, is arguably the most widely used alternative to gross domestic product. However, official country-resolution estimates (N=191) limit its use. We build on recent advances in machine learning and satellite imagery to produce and distribute global estimates of the Human Development Index for municipalities (N=61,530) and a 0.1° × 0.1° grid (N=819,309). To construct these estimates, we develop and validate a generalizable downscaling technique based on satellite imagery that allows for training and prediction with observations of arbitrary size and shape. We show how our estimates can improve decision-making and that more than half of the global population was previously assigned to the incorrect Human Development Index quintile within each country due to aggregation bias. We publish the satellite features necessary to increase the spatial resolution of any other administrative data that is detectable via imagery.

The Human Development Index (HDI) is widely used by policymakers and academics to summarize three key dimensions of wellbeing: the population's health, human capital, and standard of living[1–4]. A more comprehensive measure of wellbeing than income or wealth alone[2,3,5], HDI is used to categorize countries by their level of human development, which, in turn, can determine allocations of global resources[6]. However, the United Nations Development Program (UNDP) releases official global estimates of HDI annually only at the highly aggregated national level (N=191), preventing the use of the indicator in applications that require subnational information. Thus, measures of income remain the dominant metric for evaluating development progress within countries, in part because they are more readily available.

In an effort to address this, non-UN researchers[7] recently processed extensive household survey data in order to produce HDI estimates for first-level administrative units–i.e., provinces and states, hereafter called "provinces" (N = 1739). These efforts have substantially advanced our understanding of global development patterns, but

province-level measures nonetheless remain too coarse for many modern policy applications where local information is needed, such as community-level targeting of aid[8,9]. Furthermore, the reliance of current HDI estimates on slow, infrequent, and costly ground-based data collection limits the usability of HDI for most practical applications other than cross-national rankings.

The combination of satellite imagery and machine learning (SIML) is increasingly used to predict socioeconomic variables at fine spatial resolution[10–18]. This approach enables information that is expensive to obtain through ground surveys to be estimated at low cost. While SIML estimates do not replicate ground surveys exactly[19–21], the quality of SIML estimates is now high enough that it can assist targeting of aid and program evaluation in remote communities where alternative sources of information are unavailable[8,10,20,22,23].

Here, we produce global estimates of HDI at the second administrative level – i.e., municipalities and counties, hereafter called "municipalities" (N=61,530) and for a global 0.1° × 0.1° (approximately

[1]Global Policy Laboratory, Stanford Doerr School of Sustainability, Stanford University, Stanford, CA, USA. [2]Food and Resource Economics, Faculty of Land and Food Systems, University of British Columbia, Vancouver, BC, Canada. [3]Division of Humanities and Social Sciences, California Institute of Technology, Pasadena, CA, USA. [4]National Bureau of Economic Research, Cambridge, MA, USA. [5]Human Development Report Office, United Nations Development Programme, New York, NY, USA. [6]These authors contributed equally: Luke Sherman, Jonathan Proctor. ✉e-mail: jon.proctor@ubc.ca

10km by 10km) grid. We construct these estimates by combining information from prior provincial estimates[2] with global daytime and nighttime satellite imagery[24,25]. Our approach builds on recent advances in machine learning[12,26] to develop a general method that learns the relationship between imagery and an outcome of interest (here, HDI) using data from any set of political boundaries. We can then use that relationship to estimate the outcome for any other set of boundaries. Importantly, our method works for spatial units of arbitrary shape and size, so models can be trained on coarse-resolution outcome measurements and make predictions at finer resolution. We apply this method to transform provincial HDI measures into finer resolution estimates. While other such "downscaling" approaches typically rely on either theoretically informed relationships or simple dasymetric masks[27,28] our approach to making predictions at finer resolution than the source labels uses machine learning and satellite imagery to identify complex spatial relationships between imagery and an outcome of interest (Fig. 1).

## Results

Our results have six sections. First, we outline the theoretical approach to downscaling. Second, we train and evaluate a global model for HDI

at the province-level using aggregates of satellite features. Third, we implement multiple tests to validate that this model is skillful. Fourth we examine performance for the components of HDI separately. Fifth, we generate high-resolution global HDI data using this procedure, and we evaluate how these estimates compare to existing aggregated estimates. Finally, we illustrate how these high-resolution estimates could alter decision-making when targeting aid.

## 1. A general approach to downscaling administrative data using satellite imagery and machine learning

The ability of SIML systems to promote development is limited by the paucity of suitable observations for model training[10]. This limitation is partly due to the design of modern SIML methods, since large quantities of administrative data are available, but existing systems are generally not designed to make use of them. To date, SIML approaches for predicting human outcomes have standardized the structure of both the training labels and corresponding imagery so that the unit of analysis is a regular spatial structure, such as a square. For example, many systems use convolutional neural networks (CNNs)[11,13,16], which tend to perform well on diverse computer vision tasks. CNNs, however, typically require images to be a constant size and shape, such as 224 x

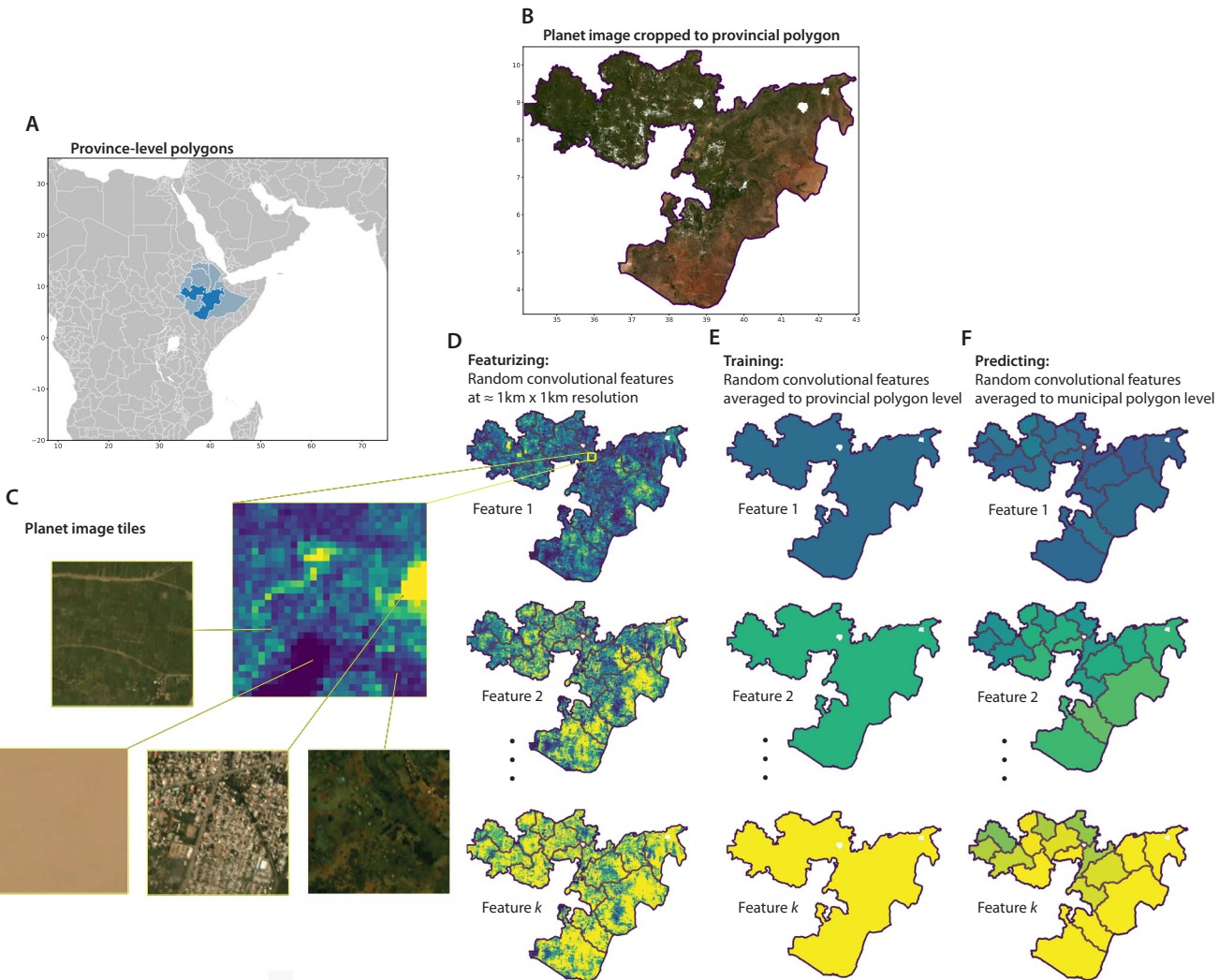

**Fig. 1 | The Multi-task Observation using Satellite Imagery and Kitchen Sinks (MOSAIKS) approach transforms satellite imagery for each administrative polygon into a vector of image features.** **A** The location of Oromia, an example province (ADM1 unit) within Ethiopia. **B** A composite of Planet imagery over Oromia in 2019. **C** A sample of 0.01° x 0.01° image tiles. **D** Three examples of MOSAIKS random convolutional features over Oromia; each pixel shows the feature value for a single

0.01° × 0.01° image ($\mathbf{X}_{tile}$). **E** The corresponding aggregation of these MOSAIKS features to the provincial polygon (ADM1) level for model training ($\overline{\mathbf{X}}_{province}$). **F** Aggregation of these same MOSAIKS features to the municipal polygon (ADM2) level for fine-resolution prediction of HDI ($\overline{\mathbf{X}}_{municipality}$). See Figure S1 for an illustration of how MOSAIKS features are calculated and used to predict HDI. The municipal shapefile used here is from geoBoundaries[65], published under a CC-BY 4.0 license.

224 x 3 pixels in the case of the commonly used ResNet-18[29]. This restriction has caused prior studies to rely on coarse approximations for linking irregularly shaped labels to corresponding imagery, for example, by averaging polygon labels that overlap with the square image[12,14]. Such procedures can introduce considerable error when administrative polygons are much larger or smaller than the chosen square size. This is particularly relevant for HDI, for which data is globally available only for nations or provinces, which tend to be irregularly shaped and vary greatly in spatial extent. For example, the largest provincial polygon in our data is the Far Eastern Federal District of Russia, which is over 6 million $km^2$, and the smallest is Banjul of Gambia, which is 7 $km^2$. Developing a robust and widely applicable SIML system that can be trained on inputs that correspond with such diverse administrative structures requires an alternative strategy.

In an ideal setting, we would solve for a function that could directly map a single satellite image "tile" (e.g., 1km × 1km) to the corresponding HDI for the same tile

$$HDI_{tile} = f(satellite\_image_{tile}) + \epsilon_{tile} \quad (1)$$

where $\epsilon$ is the component of HDI that is not measurable with imagery. In theory, Eq. (1) could be solved directly with many learning approaches, such as a CNN[30], but this is infeasible in practice because tile-level data on HDI (i.e., the left-hand side of Eq. (1), $HDI_{tile}$) does not exist. Instead, we observe only aggregated estimates of HDI over politically-defined regions ($HDI_{country}$ or $HDI_{province}$) that correspond with large and irregular agglomerations of image tiles. For the SIML system described by $f(\cdot)$, this creates a mismatch between the spatial structure of inputs (image tiles) and outputs (administrative regions).

We solve this problem by converting image tiles into a generalizable set of descriptive variables or "features," $\mathbf{X}_{tile}$, such that $f(\cdot)$ can be structured as linear in these features,

$$f(satellite\_image_{tile}) = \boldsymbol{\beta} \cdot \mathbf{X}_{tile}, \quad (2)$$

where $\boldsymbol{\beta}$ is a vector of weights (i.e., coefficients). Specifically, we construct a basis for the imagery such that outcomes of interest are well-represented by linear combinations of the basis vectors. This allows aggregate administrative measures of HDI to project onto corresponding aggregations of tile-level features with the same weights that would be recovered if the problem had been solved using only tile-level data. Thus, we learn the model

$$HDI_{province} = \boldsymbol{\beta} \cdot \underbrace{\left( \frac{1}{N} \sum_{tile \in province} \mathbf{X}_{tile} \right)}_{\bar{\mathbf{X}}_{province}} + \epsilon_{province} \quad (3)$$

and recover the same weights $\boldsymbol{\beta}$ that we would have recovered had we directly solved Eq. (1) using the linearization in Eq. (2). See Supplementary Information S4 for an empirical validation of the scale-invariance of model weights using this approach. Note that $\bar{\mathbf{X}}_{province}$ is simply the vector of average tile-level features within a province. The weights $\boldsymbol{\beta}$ can then be used to generate predictions for arbitrary aggregations of tiles. We use these $\boldsymbol{\beta}$ to downscale HDI to the municipality level ($\boldsymbol{\beta} \cdot \bar{\mathbf{X}}_{municipality} = \widehat{HDI}_{municipality}$) and the tile level ($\boldsymbol{\beta} \cdot \bar{\mathbf{X}}_{tile} = \widehat{HDI}_{tile}$).

The benefits of linearizing this problem have been understood in general terms, since linear models of basic scalar image properties (e.g., "greenness"[31] or nighttime lights[32]) have been widely used to downscale administrative-level data. However, to our knowledge, it has not been shown that such linearization is possible and skillful for the types of featurizations that capture complex spatial structures in imagery.

Here we demonstrate that such a skillful linearization can be achieved by embedding rich image information using the Multi-task Observation using Satellite Imagery and Kitchen Sinks (MOSAIKS) approach[12]. Converting images into MOSAIKS features ($daytime\_satellite\_image \rightarrow \mathbf{X}_{MOSAIKS}$) provides a structured representation of the unstructured information within the satellite image that performs well in linear models. These features summarize the joint distribution of color and textures within daytime tri-band optical imagery (see "Methods" 1 and Figure S1). We concatenate these features with features that summarize nighttime lights of locations[33,34] ($nighttime\_satellite\_image \rightarrow \mathbf{X}_{NL}$) to construct a linear model that downscales HDI using only satellite data (see Methods 4, 5 and 6).

## 2. Predicting province-level HDI using satellite imagery

Using province-level administrative HDI data for training (as in Equation (3) and Fig. 1), we find that predictions made using linear aggregates of daytime and nighttime satellite image features, anchored to known country means, explain 96% of the variation in global provincial HDI values (Fig. 2A, denoted "full variation performance"). Specifically, we train a model to predict provincial HDI deviations from each country mean and then add the known country mean to the predicted provincial deviations. We take this "mean-anchored" approach, because it reflects how SIML may be used to augment existing HDI data in practice (see "Methods" 9 and Supplementary Information S6.1 for a discussion of mean-anchoring). Since most HDI variation is across-countries rather than within-countries (Fig. 2A, B), much of this performance predicting provincial variation is driven by the measured country means that the predictions are anchored to.

The primary value of incorporating satellite imagery is to explain local-scale variation in HDI. As a first test of the model's ability to explain local variation, we evaluate model performance predicting deviations in provincial HDI from the country mean. We find that model predictions explain 52% of this within-country provincial variation in HDI (Fig. 2B; see "Methods" 7 for a discussion of performance metrics). This indicates that SIML-based provincial predictions of HDI add substantial fine-resolution information to existing national measures. Importantly, models trained on provincial deviations from the country-level HDI have higher performance predicting such deviations than models trained directly on the provincial values themselves (Table S1 col. 4). Intuitively, model weights are optimized to explain the smaller within-country variation in the demeaned model, rather than the larger across-country deviations (Fig. 2A, B). We use such "within-country" models as our primary model specification.

## 3. Validating downscaling of data below the province-level

We cannot directly evaluate the performance of municipality-level or grid-level HDI predictions worldwide because such highly-resolved estimates have not been previously constructed. Nonetheless, we test the performance of our downscaling technique in three ways that allow predictions to be directly compared to "ground truth" at finer resolution than the training data. First, we directly compare our municipality-level HDI predictions to census-derived estimates in three countries where these data are available: Indonesia, Brazil, and Mexico[9,35,36]. Second, we train a model relating satellite imagery to the International Wealth Index (IWI) at the province level, and then construct downscaled predictions of IWI at the resolution of Demographic and Health Surveys (DHS) clusters where granular IWI measurements are available[37]. The IWI is an alternative development indicator to HDI that omits measures of education and health. Third, we train a model to predict nighttime lights (NL), a common proxy for economic wellbeing[33,38–43], using features constructed exclusively from daytime satellite imagery, and test whether our approach can downscale NL. Mirroring the structure of our HDI analysis, we train a model using only NL labels aggregated to the province level, and then evaluate predictions of NL at the municipality level. No test can directly validate the

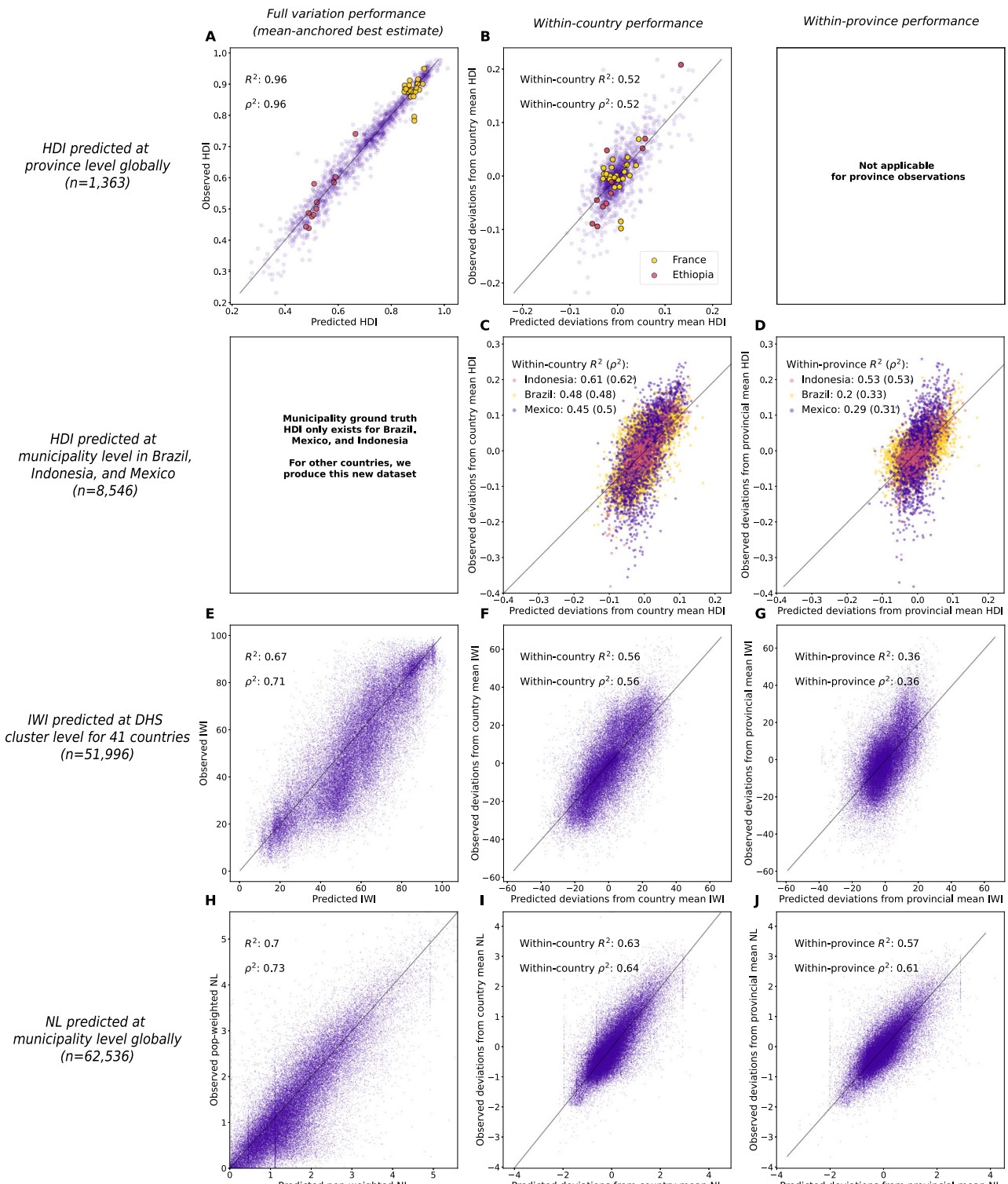

**Fig. 2 | MOSAIKS models perform well predicting socioeconomic indicators, including at downscaled resolution. A** Observed and predicted HDI at the province level. Note that the within-country variation is smaller than the across-country variation, as illustrated by France, in yellow, and Ethiopia, in pink. **B** The same as (**A**), evaluating within-country variation. Provincial deviations from the country mean for France and Ethiopia are now centered at 0, and the model is evaluated on how well it can differentiate provinces that are relatively well and worse-off within countries. **C**, **D** Observed and predicted municipal HDI data in Mexico, Brazil and Indonesia. **E**–**G** Observed and predicted International Wealth Index (IWI) at the Demographic and Health Surveys (DHS) cluster level. **H**–**J** Observed and predicted nighttime lights (NL) at the municipality level. The vertical streaking in (**H**–**J**) are caused by countries that have very spatially dense municipalities (Supplementary Information S6.3). Predictions are anchored to known country or provincial means (Methods 9). All predictions are for the year 2019.

performance for downscaling HDI globally, since the data necessary for such a test do not exist; however, all three of these large-scale tests taken together document the effectiveness of our downscaling strategy in general (using global socioeconomic data similar to HDI) and for HDI in particular (using municipal HDI data for three countries).

When evaluating estimates made at finer resolution than that of the training data, we mean-anchor downscaled estimates to the known provincial mean. This approach produces the best possible estimates by using the satellite-based model to explain within-province variation, which is previously unknown, and the known provincial values to explain the across-province variation.

As a direct evaluation of HDI downscaling performance, we compare municipal HDI predictions from the satellite-based model trained on provincial HDI deviations from the country mean to municipal HDI derived from census-based calculations in Mexico[9], Brazil[36], and Indonesia[35] (Methods 2). In Mexico, downscaled HDI predictions explain 45% of the municipal HDI variation overall (Fig. 2C) and 29% of the within-province variation (Fig. 2D). In Brazil, HDI predictions explain 48% of municipal HDI variation overall, and 20% of the within-province variation. And in Indonesia, HDI predictions explain 61% of the municipal HDI variation overall and 53% of the within-province variation. It is encouraging that HDI predictions align better with census-based measures for Indonesia than for Mexico or Brazil because measures for Indonesia are from 2019, which aligns with our satellite-based predictions of within-province HDI variation, while measures for Mexico and Brazil are from 2010. These older HDI measures could differ from our satellite-based predictions in part due to changes in HDI since the measurements were taken. Additionally, some portion of the misalignment between satellite-based predictions and survey estimates likely arises as a result of errors in the survey data itself, a widely recognized issue[10] that we are unable to assess. Together, these results indicate that our method for SIML-based downscaling improves our understanding of the spatial distribution of HDI in these three example countries. We emphasize that these data are not a complete substitute for survey-based estimates and that our model could not be trained or evaluated without ground-truth surveys.

We test the ability of our approach to downscale IWI internationally by training a model on province-level aggregates of IWI and then predicting IWI across DHS clusters. This is a more difficult task than predicting municipality-level values and an equally difficult task as predicting at the $0.1° \times 0.1°$ grid-level, since DHS clusters tend to be even finer resolution than municipalities and about the same size as the HDI grid (DHS cluster $\approx 180$ km$^2$, municipality $\approx 2000$ km$^2$, grid tile $\approx 120$ km$^2$). Models are trained on 862 provincial observations within 85 countries and evaluated at 51,996 DHS clusters (Table S1). Analogous to our approach with HDI, models are trained on province-level deviations from country-level means and predictions are re-centered to match the observed province-level mean, as these values are known (see "Methods" S6.2). Downscaled IWI predictions explain 67% of the variation in IWI across all DHS clusters (Fig. 2E) and 56% of the variation in IWI across DHS clusters within countries (i.e., of cluster deviations from the country mean, Fig. 2F). Importantly, this approach is also able to predict 36% of the variation in IWI within the provincial units that it was trained on (Fig. 2G). This result demonstrates the ability of our downscaling approach to generate skillful global-scale predictions at resolutions higher than the training data, and also its ability generalize to measures other than HDI.

To further evaluate our approach in a global test, we train a model on aggregate provincial NL and evaluate predictions at municipal resolution. NL are not a direct measure of human welfare; however, they are generally correlated with income and other development indicators[39–41,44,45]. NL have even been used along with population to construct development indicators correlated with HDI (e.g.,[42]). NL are particularly useful here because they allow us to design a validation test where true subnational values are known worldwide. Mirroring our

HDI process, we train the model using provincial deviations from the country mean, and then construct municipal values as the predicted municipal deviations from the country mean plus the known country mean (Methods S6.3). Downscaled NL predictions capture 70% of municipal variation in NL globally (Fig. 2H), 63% of the municipal variation within countries (Fig. 2I), and, most importantly, 57% of the variation across municipalities within provinces (Fig. 2J). These results further reinforce the ability of our approach to downscale global province-level data and underscore its generalizability to other non-HDI outcomes. Unlike the two downscaling experiments above, this experiment relies entirely on features generated using only daytime imagery (Figure S2).

Comparisons to municipal data on HDI, IWI, and NL indicate that models trained on provincial data can explain 20-57% of within-province municipal variation. Collectively, these three experiments demonstrate that our approach effectively combines coarse socioeconomic measurements with satellite data to produce skillful estimates at spatial resolutions finer than the province-level training data.

## 4. Additional evaluation of model performance

One motivation for using MOSAIKS features to downscale HDI is their ability to predict a diversity of ground-based measures. This is particularly relevant for predicting HDI, since it is constructed from components that capture human health, education, and income. To consider which components of HDI are best captured by our estimates, we retrain models to predict each component of HDI separately. We find that MOSAIKS models explain 92% (5%) of the full (within-country) variation in provincial life expectancy, 93% (51%) of mean years of schooling, 90% (27%) of expected years of schooling, and 97% (56%) of gross national income per capita (GNIpc) (Table S3). While the components of HDI do tend to be correlated (Table S4), these results indicate that instead of just capturing income, predictions of HDI using satellite imagery maintain the ability to capture multiple dimensions of human wellbeing. These results also help explain what aspects of human development satellite features are able to capture, indicating their ability to explain local variation in education and income as well as their difficulty predicting aspects of health. Predictions of HDI made from combinations of its individually predicted components perform nearly identically to the direct predictions of HDI used throughout this analysis.

Further model evaluation including model performance across global regions, and the value of combining daytime and nighttime imagery are discussed in Supplementary Information S2. While it remains untestable how accurately the approach can downscale HDI globally, we find that the approach can predict substantial global variation in HDI and its components, and that it can downscale both HDI in three countries and variables related to HDI globally with accuracy.

## 5. Global municipality-level and grid-level estimates of HDI

We use our model for within-country HDI (from Results Sections 2–3) to estimate HDI in 2019 for 61,530 municipalities and 819,309 $0.1° \times 0.1°$ grid tiles (Fig. 3), the finest resolutions at which HDI has been estimated globally (Methods 10). We make these municipal and grid-level estimates of HDI publicly available for download at mosaiks.org/hdi. We also similarly produce and make available estimates of the individual components of HDI.

Our high-resolution estimates enable a substantially more detailed understanding of human development compared with national and provincial measures (Fig. 3A, B vs. C, D). Both municipal and grid-level estimates reveal within-province heterogeneity of HDI that was previously un-resolved (Figures S3, S4). The gridded HDI estimates tend to be higher along major roadways, especially at the intersection of roadways (Figure S5). Borders, such as between Turkey, Georgia, Armenia, Azerbaijan, Iraq and Iran, are less apparent in the

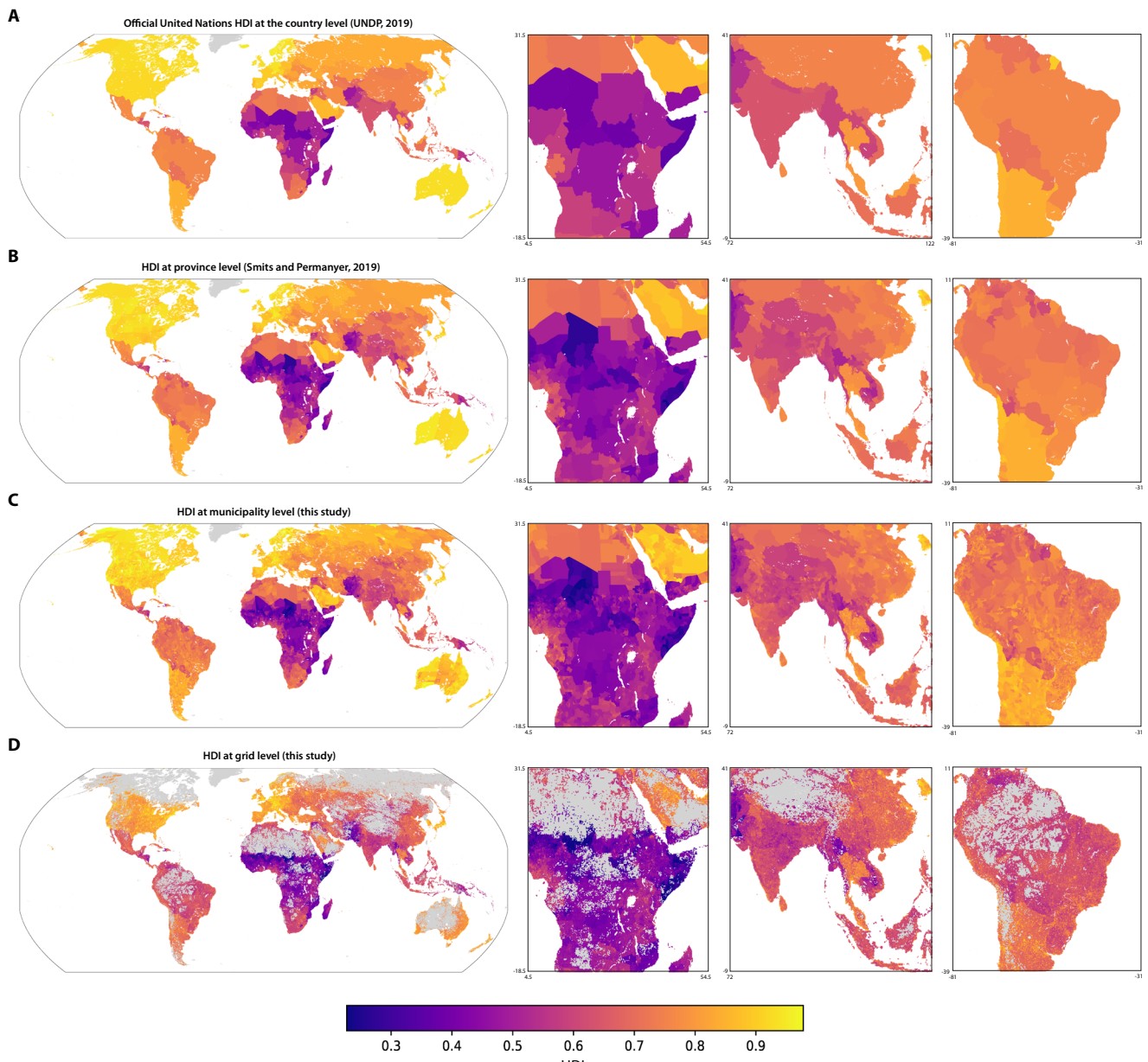

**Fig. 3 | Global HDI estimates at the municipal and grid levels. A** Official United Nations HDI at the country level[4] **B** HDI data at the province level from Smits and Permanyer[7]. **C** Municipal level estimates of HDI produced here. **D** Grid level estimates of HDI at the 0.1° by 0.1° (approximately 10km by 10km) level produced here. Gray in the grid-level estimates indicates land area believed to be unsettled[60]. All data shown are for the year 2019. The municipal shapefile used here is from geoBoundaries[65], published under a (CC-BY 4.0 license.).

fine resolution estimates, indicating a greater continuity in human development across space than in the provincial maps. Higher HDI levels in the city centers and lower HDI levels in the suburbs of capital cities such as Moscow, Russia and Antananarivo, Madagascar are also visible in the municipal and grid estimates, but obscured in the provincial estimates. The contribution of environmental features to human development is illustrated in eastern Pakistan and northwestern India, where human development is higher in the plains bordering the Indus River and its tributaries, and lower in neighboring deserts. Similarly, within Sonora and Sinaloa in Mexico, coastal areas show higher human development than inland regions. This local heterogeneity in HDI indicates that uniform assignment of HDI to populations based on their country or province of residence is inaccurate because it groups together populations with very different levels of human development.

We use our estimates to quantify the degree of aggregation bias that occurs when using only province-level estimates. Aggregation bias occurs here because small units (i.e., grids or municipalities) are assigned the HDI of a larger unit (i.e., a province), which does not reflect local conditions. For example, a small urban region, where HDI is high, embedded in a province that also contains large, less developed rural areas, will be assigned a HDI level that is too low when provincial measures are used. We quantify how frequently such misassignment occurs within countries by assigning populations to a quintile of HDI within their national HDI distribution based on provincial, municipal, or grid-level estimates. We then evaluate how frequently the province-level estimates agree with the more highly resolved estimates (Fig. 4).

We find that a majority of the global population (58% using municipal estimates and 65% using grid estimates) is assigned to a

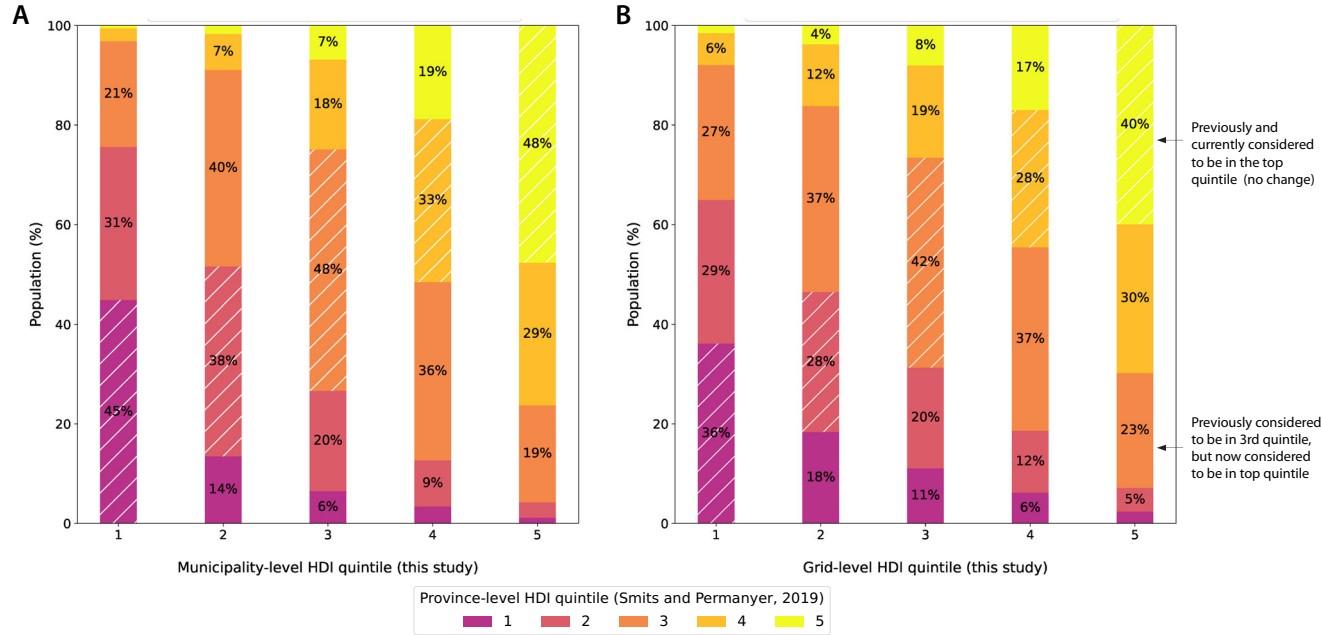

**Fig. 4 | Municipal and grid-level estimates of HDI assign more than half of the global population to a different within-country HDI quintile than provincial estimates. A** Shows the difference in estimated HDI quintile, within countries, using provincial vs. municipal data. **B** shows the same analysis using grid-level HDI estimates. Colors show the estimated HDI quintile using provincial data from[7], where yellow is high human development and purple is low human development. All data is from 2019. Bins along the *x*-axis show estimated HDI quintiles using the municipal and grid level data produced in this study. Hatch marks indicate no change in quintile assignment using municipal data. When provincial data do not allow for the creation of five distinct bins, the population is assigned first to the middle bin, followed by the neighboring bins. For example, if a country does not have any province-level data, the entire population is assumed to be in the middle quintile for that country. For this reason, a greater fraction of the global population is assigned to the middle quintile (33%) than to the outer quintiles when using the provincial data. Note: in validation tests, average within-country $R^2$=0.51 and $\rho^2$=0.53 for municipal HDI in Brazil, Mexico, and Indonesia.

different within-country HDI quintile compared to when using provincial estimates. For example, of the population measured to be in the bottom two HDI quintiles using provincial estimates, 8.5% are reassigned to the top two HDI quintiles using municipal estimates and 12.9% using grid-estimates. Grid-level estimates reveal larger amounts of aggregation bias due to their finer resolution. Based on our grid-level estimates, we estimate that 20.4% (21.0%) of the global population is one quintile lower (higher) than assigned using provincial estimates, and 9.5% (9.3%) are two quintiles lower (higher).

## 6. Illustrative application: targeting policy in Mexico

To explore how our HDI measures could improve the efficiency of development policies, we conduct an illustrative targeting exercise within Mexico. We simulate how a geographically targeted policy based on provincial vs municipal HDI data (Fig. 5A, B) might achieve different outcomes, noting that previous work has shown that more spatially granular targeting can produce meaningful welfare gains[13,46–49]. We study Mexico because "ground truth" estimates of HDI enable us to benchmark performance[9].

The use of municipality-level data improves the number of program recipients correctly targeted. Supposing that the program director aims to provide assistance to the 10% of individuals with the lowest HDI, accuracy of program targeting increases by 11.4% percentage points (from 32.3% to 43.7%) when using municipal data (Fig. 5E, assuming the standard deviation of individual HDI within each municipality is 0.1, see "Methods" 11). Use of the municipal data also results in a much greater geographic dispersion of targeted municipalities (Fig. 5C, D). Evaluation of targeting performance using a receiver operating characteristic curve[8] gives similar results (Fig. 5F, Methods 11). Replicating this analysis for Indonesia also gives consistent results (Figure S10).

This application illustrates, in a hypothetical scenario, how fine-resolution HDI estimates could improve targeting of resources, even though these fine-resolution estimates are imperfect. Users of these fine-resolution estimates should consider their specific policy context to determine whether these fine-resolution estimates, which explain 29% of the within-province HDI variation in Mexico, 20% in Brazil, and 53% in Indonesia, provide sufficient additional information to be useful in their setting. Moreover, we acknowledge that there exist other datasets that could alternatively be used for wealth- or income-based targeting (e.g., [13]). Still, these estimates may be particularly valuable in data poor settings, where traditional data sources such as surveys are less available and reliable[10]; only about half of the world's lowest-income countries have conducted a census in the last decade[11,22]. These estimates may also be useful in cross-country settings where a globally consistent metric of welfare is desired.

## Discussion

We produce and make publicly available global-scale, high-resolution estimates of HDI, enabling the use of broad-based measures of well-being for local decision-making and policy prioritization. We achieve this by developing an approach for generating spatially granular development indicators using SIML models trained on spatially aggregated and inconsistently structured administrative data. Since many forms of non-HDI data are also available only for administrative regions, we believe this generalizable method will increase the range of outcomes that can be used as labels in SIML models, which are currently constrained by limited training data[10].

Our strategy is motivated by the limited resolution of available training data for HDI, but our results do not exhibit obvious compromises in performance relative to alternatives that exploit high-resolution labels. A related benchmark in this literature achieves $\rho^2 = 0.63$–$0.67$ predicting a wealth index when training and evaluating at the DHS cluster resolution[11,50]. Though we train at the provincial level, not the DHS cluster level, our performance predicting DHS cluster IWI is competitive with this previous analysis ($\rho^2 = 0.71$; Table

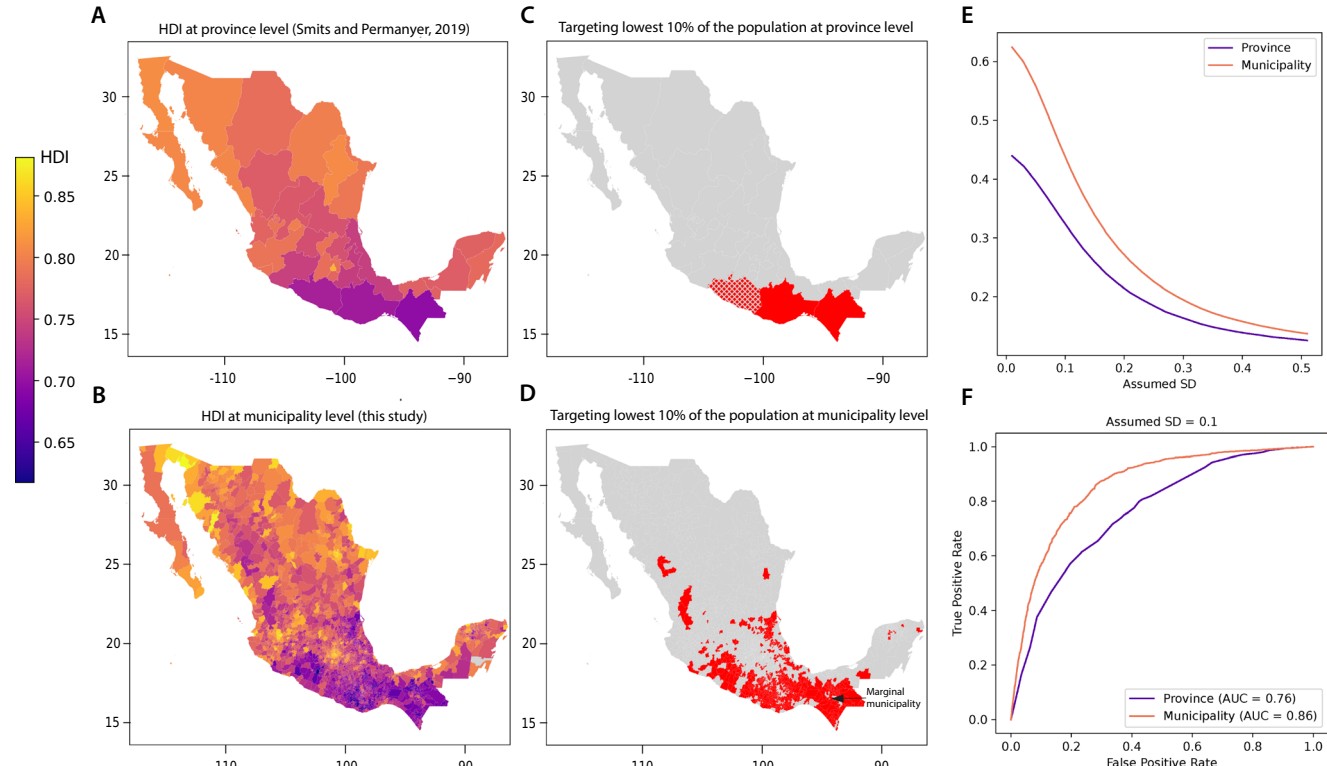

**Fig. 5 | Spatially granular HDI measures can improve decision-making. A** HDI for 2019 at the province level of observation[7] **B** HDI estimates for 2019 at the municipality level produced in this paper. **C** Lowest HDI provinces that would be targeted until 10% of the country's population is reached. **D** Lowest HDI municipalities that would be targeted until 10% of the country's population is reached. Hashing in (**C**) and (**D**) shows the marginal province and municipality that would be partially targeted. **E** Targeting accuracy (true positive rate) as a function of the assumed standard deviation of HDI within each municipality. **F** Receiver operating characteristic (ROC) curves illustrate the degree of improvement in area under the curve (AUC) that comes with targeting at the municipality level relative to the province level (assumed SD of 0.1 within each municipality). The municipal shapefile used here is from geoBoundaries[65], published under a CC-BY 4.0 license.

S1, col 1). Another benchmark in this literature achieves an $R^2$ of 0.54 predicting RWI, which is conceptually identical to what we call within-country IWI, at the same DHS cluster level[13] (metric taken from a replication of their Figure 3B). Again, despite training at the provincial level rather than the DHS cluster level we achieve a within-country $R^2$ of 0.56 predicting IWI at the cluster level. Direct comparison to both of these benchmarks, however, is complicated by differences in training and evaluation methodologies including our focus on explaining within-province variation and our corresponding mean-anchoring approach.

This approach to producing HDI estimates can be viewed as a type of regression-based technique for small area estimation using satellite imagery[51,52]. Though data limitations in this setting—including the lack of fine-resolution global surveys of HDI—preclude the direct implementation of most commonly used small area estimation techniques (e.g.,[53]), this approach follows the general small area estimation framework of combining limited measurements of the variable of interest (provincial HDI) with universally available auxiliary data informing the variable of interest (satellite imagery) to make accurate and precise fine-resolution predictions. A key difference between more traditional small area estimation techniques and ours is that we use a global-level auxiliary dataset that does not suffer from the challenge of construct validity across countries, although this comes with a tradeoff of needing to assume some degree of consistency between imagery and conditions on the ground. A recent comparison of poverty maps for Malawi produced using i) a small area estimation approach combining survey data and census data and ii) an alternative approach combining survey data and geospatial indicators including satellite imagery found that the two approaches produced very similar estimates, with a correlation exceeding 0.9[54]. This, paired with the performance of satellite

imagery predicting HDI shown here, indicates that satellite imagery can be an effective substitute for census information when producing fine-resolution estimates of human wellbeing in settings where census data is not available. In locations where reliable census data is available, satellite imagery could serve as a complementary source of information. We note also that additional sources of auxiliary data could be used in future research to improve these estimates (e.g., Supplementary Information S5 and Table S5).

One important property of the errors in our estimates, common in machine learning[12,20,21], is that our predictions exhibit lower variance than the true values. For example, in Mexico the standard deviation of our satellite-derived estimates is approximately half that of census-derived values. Survey and other traditional approaches to data collection remain critical to informing the state of global human development and complement satellite models, which cannot be trained or evaluated without ground-truth measures.

A benefit of our approach is its mathematical transparency[55], relative to deep learning alternatives; however, like most computer vision approaches, it is a challenge to explain why our model generates a particular prediction. To explore which specific phenomena are correlated with our HDI estimates, we examine how predicted HDI values associate with other variables globally, an approach similar to some prior efforts to evaluate model predictions[52]. We find that municipalities with higher road density (full variation $\rho^2$=0.24, within-country $\rho^2$=0.14) and building density (full variation $\rho^2$=0.07, within-country $\rho^2$=0.16) tend to have higher predicted HDI (Figure S8, Supplementary Information S3). Population has a weak association with HDI across the globe (full variation $\rho^2$=0.02), but within countries, more densely populated municipalities tend to have higher HDI ($\rho^2$=0.33). In contrast, cooler countries tend to have higher HDI (full

variation $\rho^2$=0.12), but temperature has little association with HDI within countries ($\rho^2$=0.0). Terrain ruggedness, forest cover, crop cover, and rainfall have little to no association with our HDI estimates. Collectively this analysis indicates that our HDI estimates capture image information related to built infrastructure and the density of human settlements, but also indicates that there is residual image information beyond these factors that contribute to the HDI estimates. Together the variables analyzed above explain 32% of the full variation in municipal HDI estimates and 33% of the within-country variation in a multiple linear regression analysis.

Policymakers will face a number of complex considerations before using these data in practice. Broadly, policymakers can benefit from algorithmically-informed decisions that are transparent and explainable[56] and there may be costs to using data for decision-making that is fundamentally harder to interpret and explain than alternative decisions made using traditional survey data. In addition, policymakers will have to decide whether and how it is appropriate to use a global dataset that we have only been able to directly validate in three countries. Third, policymakers may benefit from technical evaluations that interpret the performance here so that they can assess the marginal benefits of using these data for their specific use-case, recognizing that reasonable people may disagree about whether the data provided here achieve high enough performance to merit its use for specific applications. Future work should investigate improvements in remote sensing model interpretability that can support transparent policy-making and aid in understanding potential sources of systematic bias[55,57,58].

We emphasize that the approach described here can be used to predict a wide variety of labels for which country, province, and/or municipality level labels exist. To facilitate this use, we make the features used in this analysis publicly available at the country, province, and municipality level via mosaiks.org/hdi[24]. We offer these features aggregated to these administrative-unit levels using both area and population weights. Each of these files is relatively small, ≈ 3 GB or less, and thus possible to process on a desktop computer. For comparison, the global set of features at 1km resolution is ≈ 3 TB and the raw imagery is ≈ 30 TB. We hope that researchers and decision-makers can leverage these features, along with their own administrative datasets, to produce new downscaled estimates of socially-relevant outcomes. Such spatially granular data may create new opportunities for achieving global development goals.

## Method

### 1. Overview

To transform satellite imagery into descriptive features that exhibit high performance in linear models, we build on the recent development of MOSAIKS, an approach that achieves performance competitive with CNNs using an unsupervised image embedding combined with a linear ridge regression model[12]. The linearity of ridge regression enables the scale invariance of our downscaling approach[12]. MOSAIKS random convolutional features combined with ridge regression have been shown to be skillful at solving diverse prediction problems —such as forest cover, population, elevation, and house price— using only imagery as inputs and using only a single linear specification. This property makes MOSAIKS a particularly appealing approach for predicting HDI, which is constructed from multiple development indicators. Each MOSAIKS feature for a tile describes the similarity between the satellite image and a smaller patch of imagery, and is calculated as a nonlinear transformation of the image's pooled convolution with a random sub-image from the sample[12,59]. For example, a feature whose patch was sampled from a city might inform how urban an image is, while a feature whose patch was sampled from farmland might inform how agricultural an image is. Together, thousands of MOSAIKS features form a basis that can skillfully describe the rich structure contained within large imagery datasets through simple linear combinations of the features. For details on how MOSAIKS captures visual information from satellite imagery see Figure S1, as well as the Methods sections "Theoretical foundations" and "Convolutional random kitchen sinks", and the Supplementary Information section "Alternative interpretations relating MOSAIKS to kernels and CNNs" of ref. 12.

To compute local HDI via SIML, we transform a dataset of global Planet imagery ( ≈ 5m resolution) into a set of 4000 general-purpose MOSAIKS features ($X$) for 0.01° × 0.01° tiles ( ≈ 1km × 1km; Fig. 1)[24]. We supplement these MOSAIKS features with features that flexibly characterize the distribution of nighttime lights in each tile (Methods 5). While past measures of economic and human wellbeing have tended to focus on nighttime lights alone[38–43], inclusion of high-resolution daytime imagery improves model predictions of HDI, especially in regions with low HDI (Supplementary Information S2). Visual imagery may improve performance by better resolving HDI variation in areas without electrification that are nearly dark, or by better differentiating between highly populated areas with lower HDI and less populated areas with higher HDI, which can have similar brightness in nightlight data[16]. We then learn a model that is linear in these features ($\beta \cdot \mathbf{X}$) and use this linear model to estimate HDI at high resolution. While we focus this analysis on the downscaling of HDI, this approach is generalizable to other types of administrative data associated with irregularly-shaped political units.

For both training and prediction, we average image features for administrative polygons using population weights (Fig. 1D, grid-scale predictions follow a similar procedure)[60]. This results in one vector of image features for each province and municipality in the world. To learn the relationship between the image features and HDI, we train a model on province-level HDI labels and aggregated province-level image features (Fig. 1E). We then predict municipality-level HDI using the municipality-level image features (Fig. 1F), and we predict 0. 1° × 0. 1° grid HDI using features for that grid.

Throughout this analysis we use the term "province", the abbreviation "ADM1" to refer to first-level administrative regions; and "municipality", "ADM2" to refer to second-level administrative regions, though the terminology for these units varies by country. For example, "state" and "county" are the designations used for ADM1 and ADM2 units in the United States.

Our fine-resolution global HDI estimates were produced in collaboration with researchers at the Human Development Report Office of UNDP (Supplementary Information S1), but data released with this paper should not be considered official United Nations indicators.

### 2. Label data: HDI

The United Nations Human Development Index is a composite measure used to assess a country's average achievements in three key dimensions of human development: health, education, and standard of living[4]. Health is measured by life expectancy at birth; education is evaluated using a combination of mean years of schooling for adults and expected years of schooling for children; and standard of living is assessed through gross national income per capita (GNIpc), adjusted for purchasing power parity. The HDI applies a logarithmic transformation to GNIpc to account for the diminishing benefits of increased income. Each of these components is normalized on a scale from 0 to 1, and the HDI is calculated as the geometric mean of the three dimension indices, giving equal weight to each dimension. This method allows the HDI to reflect not just economic wealth, but also broader aspects of well-being. In 2019, the average HDI across countries was 0.72, with a standard deviation of 0.15, a minimum of 0.36 in Somalia and a maximum of 0.96 in Switzerland. (Fig. 3A).

National-level HDI data originate from the UNDP Human Development Data Center and are updated every year[61]. UNDP uses data

from the World Bank, UNESCO, UNICEF, DHS, UN Stats, and other organizations to create these national-level indicators[4].

Province-level data on HDI and its components, as well as the associated provincial shapefile, come from the Global Data Lab (GDL) Subnational HDI Database V7.0[7,62]. We omit 3% of the observations, which do not match with the associated GDL shapefile. The resulting province-level HDI dataset contains 1739 provincial observations from 159 countries. Additionally, we include 20 country-level observations that do not have subnational province units (e.g., Qatar). For all model training and evaluation we use provincial HDI data from 2019, the same year as the MOSAIKS image features.

We compare our municipal HDI estimates with census-derived municipal estimates for HDI, where these data exist. For Indonesia, time series estimates of HDI at the municipality level are made publicly available by Badan Pusat Statistik (BPS), the statistics agency of Indonesia[63]. We use estimates from 2019, which makes these, to our knowledge, the only available municipal HDI data that come from the same year as our satellite imagery.

For Brazil, the Human Development Report Office of the United Nations Development Program has derived HDI data at the municipality level using census data[36]. The most recent year these data are available is 2010, which we use in this analysis. The census-derived data have a different mean than the 2019 HDI data that we use elsewhere in this analysis; though, this has no influence on the within-country or within-province evaluation metrics (Fig. 2C, D) because predictions and observations are demeaned at the country and province level, respectively, before the metrics are calculated.

For Mexico, census derived estimates of HDI are available for the year 2010, constructed by ref. 9. As with Brazil, these data have a different mean than the 2019 HDI data; though, again, that has no influence on the within-country or within-province evaluation metrics.

When certain components of HDI are not directly available at the municipal level, producers of these datasets use close proxies. For example the BPS calculates municipal HDI data in Indonesia using data on real expenditure rather than GNIpc[35]. And when calculating municipal HDI in Mexico, ref. 9 use the child survival rate rather than life expectancy at birth for the health index and an asset index rather than GNIpc for the standard of living index. Likewise, provincial HDI estimates are also constructed using proxies when direct measurements of HDI components are not available[7]. Discrepancies in how HDI is calculated at the municipal and provincial levels may contribute to differences between our satellite-based HDI estimates and these survey-based HDI estimates.

### 3. Label data: IWI

IWI data also come from GDL. These data are publicly available at the country and province levels and we use these data for 2019. GDL also provided us IWI data at the DHS cluster level, which are not publicly available. We use cluster-level IWI estimates from 2012 through 2019 in this analysis. We drop observations that do not overlap a parent province polygon and for which no imagery is available. This results in 51,996 DHS cluster observations from 41 countries.

We match all label data to time-constant satellite image features from 2019. Because these features are not contemporaneous with all labels, our results present a conservative estimate of the ability of SIML to measure HDI globally. Given that HDI variation is substantially larger over space than time, however, perfectly contemporaneous measures would likely improve performance only modestly.

### 4. MOSAIKS features

We create daytime image features using Planet's Surface Reflectance Basemaps product from 2019, which has a pixel resolution of 4.77m × 4.77m at the equator. These quarterly mosaics are processed by Planet to minimize cloud cover, balance color across seasons, and remove seams from images[24,25]. We use data from quarter 3 because it

corresponds with less ice coverage in the northern hemisphere and less cloud cover in the tropics. We follow the methods described in Rolf et al. to generate a set of 4000 task agnostic daytime image features using random convolutional features[12,24]. Two thousand of these features use a patch size of 4 × 4 × 3 pixels, and the other two thousand use a patch size of 6 × 6 × 3 pixels. The third dimension of the patch size refers to the number of color bands (i.e., red, green, and blue) that are available in Planet imagery. We selected these patch sizes because they maximized performance across three non-HDI prediction tasks: predicting nightlight intensity, road length, and forest cover at the global level. We tested patch sizes ranging from 3 × 3 × 3 to 10 × 10 × 3 and found that using a combination of two different patch sizes (with 2000 patches each) outperformed using a single patch size (with 4000 patches) across all three tasks.

We create features for all land tiles with available imagery on a global 0.01° × 0.01° equal-angle grid, amounting to ≈ 151 million feature vectors in total[24]. Features become more sparse above 60° latitude, due to a lack of available imagery. Fig. 1C shows individual images spanning 0.01° × 0.01°, along with their corresponding MOSAIKS feature values.

We create polygon-level feature vectors by averaging values across the feature tiles associated with each polygon. Each administrative polygon is represented by a single vector of 4000 daytime image features. We assign each feature tile to the administrative polygon that contains its centroid. For small municipal and DHS polygons that do not contain any tile centroids, we represent the polygon by the nearest feature tile. When averaging, we weight by population using data from the Global Human Settlement Layer (GHS-POP)[60]. We use the GHS-POP data product for the year 2020 at 30 arcsecond (0.008°) resolution.

### 5. Nighttime light features

We create non-linear NL features from the Visible Infrared Imaging Radiometer Suite (VIIRS) average masked data product[33]. We use the V2.1 annual composite from the year 2019. The data has global coverage at 15 arcsecond (0.004°) resolution. Radiance units are expressed as $nW/cm^2/sr$. We assign the small number of pixels with negative radiance values to have a value of zero.

We create features that flexibly characterize the distribution of the NL data using indicator variables that represent whether the radiance value of each NL pixel falls into each of 21 bins. The first bin represents radiance values of zero. The next 20 bins represent radiance values that fall into evenly spaced quantiles of the global distribution of positive NL values. Analogous to aggregating the daytime imagery features to the polygon level, we calculate the population-weighted average value for each of the 21 bins for a given polygon. These polygon-level NL features denote the fraction of each polygon's population that is covered by NL values represented by each of the 21 bins. This approach allows NL to associate non-linearly with the outcome variables in our linear models.

We compute average nighttime light features for polygons by weighting by population in our main analysis. Thus, changes in VIIRS grid cell sizes caused by latitude do not affect these feature values[64], since total population in each cell is measured independently of pixel area. We also verify that population-weighted features, which should have higher performance than area-weighted features because HDI is measured on a per-person basis (not per-area) do indeed have higher performance. For within-country provincial performance, $R^2$ = 0.52 using population-weighted features, and $R^2$ = 0.24 using area-weighted features, which were constructed accounting for the change in the VIIRS grid cell size with latitude.

Use of the "average masked" VIIRS data product should generally have background, biomass burning, and aurora radiance removed[34]. To the extent there is still unmasked flaring in the NL data, we anticipate that the use of population-weighting when constructing the NL

features minimizes this bias, as oil producing areas are typically sparsely populated.

## 6. General model specification

All models are trained at either the country or province level and use either the 4000 MOSAIKS daytime imagery features, the 21 NL features, or both. We train models using a five-fold cross-validation procedure with basic ridge hyper-parameter tuning. Data are split by country during cross-validation to account for spatial autocorrelation and to ensure that the model is predicting provincial outcomes when no observations from within the same country have been observed. We apply a clipping procedure that restricts model predictions to the minimum and maximum value observed in the training data. Hyperparameter tuning is done with this clipping procedure. We allow for a different hyper-parameter between the MOSAIKS and NL feature sets, though this has only a minor impact on our results.

The general linearized model, representing Eq. (3), that we implement is

$$Y = \beta_0 + \beta_1 \mathbf{X}_{MOSAIKS} + \beta_2 \mathbf{X}_{NL} + \epsilon \qquad (4)$$

Where $Y$ is used to refer to the HDI, IWI, or NL labels interchangeably. We use this same model but predict each outcome separately. $\mathbf{X}_{MOSAIKS}$ is the matrix of daytime MOSAIKS features and $\mathbf{X}_{NL}$ is the matrix of nightlight features. We learn $\beta_0$, $\beta_1$, and $\beta_2$ using ridge regression, following Rolf et al. [12]. When predicting the NL outcome, we always exclude the $\mathbf{X}_{NL}$ feature matrix. For each outcome, we report performance for models trained at the country, province, and within-country levels (Table S1). Model training is further detailed in Supplementary Information S6.1.

## 7. Performance metrics

For each model specification, we report two metrics, both of which are used in the literature. The coefficient of determination ($R^2$), used to evaluate related models by ref. [13] and others, describes the accuracy of the raw model predictions and is a direct measure of model skill. The square of the correlation coefficient ($\rho^2$), used by ref. [16] and others, scores performance after allowing model predictions to be linearly rescaled before they are compared to observed values. We calculate both of these metrics when evaluating the full variation in labels and when decomposing the variation in labels into components that are visible within-countries or within-provinces (in contrast to between-countries and between-provinces).

The "full variation" performance metrics describe how well we estimate subnational HDI globally when we use all information available to us. To calculate full variation performance, we calculate $\rho^2$ and $R^2$ on the predicted and observed values of subnational HDI directly. This evaluates the ability of model predictions to capture the total variation in the observed values–i.e., variation across countries, across provinces within countries, and across municipalities or DHS clusters within provinces. Because most of the variation in HDI and other outcomes is between countries (Fig. 2A, B), a large portion of the model's full variation performance comes directly from the mean-anchoring procedure (when it is used). Thus, the full variation performance metrics do not precisely evaluate the model's ability to predict local variation in isolation, and so they are not our preferred evaluation metric for understanding model performance within countries. Instead, we focus our analysis on within-country and, when applicable, within-province performance.

The "within-country" performance measures the amount of variation in the provincial deviations from the country mean that can be explained by the model. This metric evaluates the ability of the model to explain local variation in the outcome by removing large-scale variation in the outcome across countries in the demeaning step before predictions and observations are compared. To calculate

within-country performance, we calculate $\rho^2$ and $R^2$ after demeaning predictions and observed values at the country level (i.e., after subtracting the predicted and observed country average value from each predicted and observed data point, respectively).

The "within-province" performance metric evaluates the ability of the model to explain hyper-local variation in the outcome, such as which DHS clusters within each province have higher or lower IWI. It does this by removing all between-province variation in the predicted and observed values before they are compared. To calculate within-province performance, we calculate $\rho^2$ and $R^2$ after demeaning predictions and observed values at the province level.

## 8. Model evaluation

When reporting model performance at the same resolution as training (Fig. 2 top row, Table S1 upper section, and Table S3), we evaluate predictions from the validation folds of a five-fold spatial cross-validation procedure in which models are trained and evaluated on data from non-overlapping sets of countries. This enables more observations to be used when evaluating model performance. We also evaluate models on a held-out test set of countries that were not included when tuning the HDI model. Before analysis, we set aside ≈ 20% of the provincial HDI data to be used as a final evaluation test set by randomly sampling 35 countries and their respective provinces. Evaluation on this test set was conducted after all hyper-parameter tuning and analysis decisions were made. We find that performance is not meaningfully different in the validation and tests sets, which indicates that the models evaluated on the validation folds did not over-fit to the data (Table S2).

In the downscaling experiments, we evaluate performance using fine-resolution municipal or DHS cluster observations that were not used for model training or tuning. After tuning the model using cross-validation we retrain the model using the optimal hyper-parameters on all the province or country observations before predicting at downscaled resolution.

## 9. Mean-anchoring

In our primary within-country model, we anchor our estimates to country or province-level means depending on the experiment. Estimates from models trained on provincial or national observations in "levels" (Table S1) are never mean-anchored.

The procedure for anchoring to the country mean is illustrated in the top row of Fig. 2, where we evaluate performance at the same resolution as model training. Our within-country model is trained to predict within-country anomalies, so in order to predict HDI in "levels" (Fig. 2A), we add back the known country average HDI (Equation S2). This procedure enables us to calculate full variation performance (Fig. 2A) but has no impact on the reported within-country performance (Fig. 2B).

In the downscaling application, our goal is to produce the best possible estimates at fine resolution. Thus, when producing downscaled estimates of IWI and HDI, we anchor our predictions to the observed provincial value of the outcome (Equation S8). This recentering procedure impacts the full variation performance and the within-country performance but does not impact the within-province performance (Table S1). Note that we anchor municipal NL estimates to country means, rather than provincial means, because we find that this improves the estimates (Supplementary Information S6.3). Mean anchoring and model evaluation are further detailed in Supplementary Information S6.1.

## 10. Producing downscaled estimates of HDI

To produce fine-resolution estimates of HDI, we take the local estimates of HDI that we predict using satellite imagery and mean-anchor them to province-year values from 2019. This approach uses existing provincial values to explain provincial HDI over space and time, and

then supplements that with information on local (i.e., within-province) variation in HDI predicted by the satellite imagery.

We use the within-country model specified in Equation S1a to estimate HDI at the municipality level, using a municipality (ADM2) shapefile from geoBoundaries[65]. We anchor municipality estimates by centering predicted deviations on the observed province-level HDI value for each year, following the procedure for downscaled IWI predictions (Equation S8). We do not release HDI estimates for municipalities that cannot be linked to a parent province with a province-level HDI estimate from Smits and Permanyer[7] because there is not a known provincial value to anchor on.

To produce 0. 1° × 0. 1° estimates of HDI, we similarly use the within-country model specified in Equation S1a. We make predictions using MOSAIKS features at 0. 1° × 0. 1° resolution. This results in gridded estimates of HDI at approximately 10km$^2$ resolution. We mask out locations where humans are not believed to be settled based on GHS-POP[60] (keeping areas with population > 0) and then mean-anchor our tile estimates such that the population-weighted average of the grid tiles within each province matches known provincial HDI values.

### 11. Targeting applications

In Fig. 5 we evaluate how access to more granular HDI data improves the number of program recipients correctly targeted. We assume the goal is to provide a uniform transfer to those at or below the tenth percentile of the HDI distribution. Following Smythe and Blumenstock (2022)[8], the program is geographically targeted and all individuals within targeted regions receive the same transfer, a practice used to reduce administrative costs[46,66]. If the administrator has access to only provincial HDI measures, then eligibility is determined at the province-level. Alternatively, using our municipal estimates, the administrator is able to target the program at the municipality level.

We evaluate performance using the census-derived municipality-level HDI measurements[9]—which are not used to train our model. These are the same census-derived measures used to evaluate performance in Fig. 2C, D and discussed above (Methods 2). Because these estimates are municipality-level aggregates, we simulate HDI for individuals—i.e., the targets of the policy—by imposing additional assumptions about the distribution of HDI across individuals within a municipality. We assume HDI within each municipality has a truncated normal distribution (bounded between 0 and 1) that is centered around the census-derived municipality mean (Figure S6).

The degree to which municipal HDI measures improve targeting performance relative to provincial measures depends somewhat on the assumed dispersion of individual HDI values within municipalities. Assuming lower (higher) dispersion leads to larger (smaller) absolute—though similar proportional—gains (Fig. 5E and Figure S6).

When evaluating performance using a receiver operating characteristic curve (ROC), (Fig. 5F) the aim is still to provide assistance to the 10% of individuals with the lowest HDI, but the fraction of the total population targeted is modified to examine how the true positive (individuals with low HDI correctly given aid) and false positive (individuals with HDI above the desired cutoff incorrectly given aid) rates change accordingly. Moving from left to right on the $x$ axis in Fig. 5F implies a greater number of people receiving assistance. The area under the curve (AUC) shows the efficiency of the targeting. The AUC increases by 0.1 (+13% from 0.76 to 0.86) when municipal data are used instead of provincial data, indicating improved targeting performance when the budget constraint is varied.

As an additional robustness check, we repeat this policy targeting experiment using Indonesia municipality data[63] and observe a similar increase in AUC under the same assumptions (11% from 0.62 to 0.69). This is shown in Figure S10.

### Reporting summary

Further information on research design is available in the Nature Portfolio Reporting Summary linked to this article.

## Data availability

All data used in this analysis, other than the DHS cluster-level IWI data from the Global Data Lab, is from free, publicly available sources. Details on how to access data for replication can be found a (github. com/Global-Policy-Lab/hdi_downscaling_mosaiks). HDI estimates are available at (mosaiks.org/hdi).

## Code availability

Replication code is available at (github.com/Global-Policy-Lab/hdi_downscaling_mosaiks).

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

## Acknowledgements

We thank Tamma Carleton, Pedro Conceição, Esther Rolf, and seminar participants at The Workshop in Environmental Economics and Data Science, the WIDER Development Conference (Bogotá), IDinsight, and the American Geophysical Union Fall Meeting for their valuable feedback. We also thank Tamma Carleton for contributions to Fig. S1. We thank the Global Data Lab for sharing DHS cluster-level data on the International Wealth Index. We thank Iñaki Permanyer and Arie Wahyu Wijayanto for helping us access census-derived municipality data in Mexico and Indonesia, respectively. And we thank Betina Barbosa and Thais Delarisse for facilitating access to municipal-level HDI data for Brazil. The project benefited from input and assistance from Jeanette Tseng, Jessica Katz, and Trinetta Chong. This work was supported by the Human Development Report Office of the United Nations Development Program (S.H., Contract No. RLA/HDR02022-12). Additional support for this work comes from the National Science Foundation Graduate Research Fellowship Program (L.S., Grant No. DGE 2146752), the Harvard University Center for the Environment and Harvard University Data Science Initiative (J.P.), the Sustainability Accelerator at the Stanford Doerr School of Sustainability (L.S., S.H.), and AI for Earth supported by Microsoft and National Geographic (S.H., Grant No. NGS-57697T-18). Any opinions, findings, and conclusions or recommendations expressed in this material are those of the authors and do not necessarily reflect the views of the National Science Foundation.

## Author contributions

Conceptualization: L.S., J.P., H.D., H.T., S.H. Methodology: L.S., J.P., H.D., H.T., S.H. Formal analysis: L. S. Writing–original draft: L.S., J.P., H.D., S.H. Writing–review and editing: L.S., J.P., H.D., H.T., S.H.

## Competing interests

The authors declare no competing interests.
