## [Transparent Peer Review file · Nature Communications]

Global high-resolution estimates of the UN Human Development Index using satellite imagery and machine learning

Corresponding Author: Dr Jonathan Proctor

This manuscript has been previously reviewed at another journal. This document only contains information relating to versions considered at Nature Communications. Mentions of the other journal have been redacted.

Version 0:

Reviewer comments:

Reviewer #1

(Remarks to the Author)

In the article “Global high-resolution estimates of the United Nations Human Development Index using satellite imagery and machine learning,” the authors present a MOSAIKS-based approach to estimate the Human Development Index (HDI) at municipal and grid-level spatial resolutions using satellite imagery. Their methodology involves a downscaling technique that utilizes province-level HDI data to train the model, enabling predictions at finer spatial scales for municipalities of varying shapes and sizes, and also a grid level with 0.1x 0.1 degree. This work aims to address critical gaps in sub-national HDI data, uncovering disparities often overlooked by traditional methods. The study holds potential for bridging global development data gaps, with implications for informed policy-making and equitable resource allocation, particularly in underserved regions, emphasizing the importance of high-resolution HDI estimates. If this paper is accepted, their estimates are likely to be widely used in future research and policy-making, potentially affecting populations influenced by their outcomes. Therefore, this study requires a very high standard of rigor and thorough validation. However, we believe that this article needs significant revisions before publication in Nature Communications. We would like to begin the first round of review by sharing the eight primary concerns. Once these concerns have been adequately addressed, we believe it will be appropriate to proceed with a detailed review of the methods and their specific aspects.

1.1. Comprehensive review on related studies and comparison needed

As the author describes in the paper, there have been numerous efforts to measure socio-economic indicators related to the Sustainable Development Goals (SDGs) using satellite imagery. For instance, prior studies have focused on estimating indicators such as Gross Domestic Product (GDP) [1], poverty levels [2], and asset wealth [3]. This paper focuses on the measurement and downscaling of the Human Development Index (HDI), presenting the first global estimates of HDI at the second-level administrative units. However, an additional consideration arises as follows:

Their main contribution is the first instance of estimating HDI (and its components) using satellite imagery on a **global** scale. However, there have been several attempts that made a prediction for specific areas, such as country[4] or regional[5] level. [4] measured HDI for county regions corresponding to ADM2 level in China, and [5] measured HDI for the East Java region in Indonesia.

Especially, in a prior study [5], a more diverse range of remote sensing data sources was employed to measure HDI, rather than relying solely on MOSAIKS. For example, NDVI (Normalized Difference Vegetation Index), NDWI (Normalized Difference Water Index), and NDBI (Normalized Difference Built-up Index) were derived from Sentinel-2A bands to provide information on vegetation, water, and built-up areas, respectively. Additionally, datasets such as MODIS and Sentinel-5P were utilized. In contrast, the authors of the current paper conducted regression using MOSAIKS combined with VIIRS nightlight imagery, with results presented in Tables S1–S3. Notably, nightlight imagery was the sole baseline used for methodology validation. To enhance the novelty and robustness of their findings, experiments incorporating other globally available remote sensing datasets, such as MODIS or Sentinel, are recommended. Alternatively, the authors should explicitly address why such data were excluded, citing specific limitations such as temporal or spatial resolution constraints.

- [1] Engstrom, R., Hersh, J., & Newhouse, D. (2022). Poverty from space: Using high resolution satellite imagery for estimating economic well-being. *The World Bank Economic Review*, 36(2), 382-412.
- [2] Jean, N., Burke, M., Xie, M., Davis, W. M., Lobell, D. B., & Ermon, S. (2016). Combining satellite imagery and machine learning to predict poverty. *Science*, 353(6301), 790-794.
- [3] Yeh, C., Perez, A., Driscoll, A., Azzari, G., Tang, Z., Lobell, D., ... & Burke, M. (2020). Using publicly available satellite imagery and deep learning to understand economic well-being in Africa. *Nature communications*, 11(1), 2583.
- [4] Zhang, X., Xu, J., Zhong, S., & Wang, Z. (2024). Assessing uneven regional development using nighttime light satellite data and machine learning methods: evidence from county-level improved HDI in China. *Land*, 13(9), 1524.
- [5] Ramadhan, R., & Wijayanto, A. W. (2023). Integrating Satellite Imageries and Multiple Geospatial Big Data for Granular Mapping of Spatial Distribution of Human Development Index in East Java, Indonesia. In *Proceedings of The International Conference on Data Science and Official Statistics (Vol. 2023, No. 1, pp. 274-295)*.

2.1. Lack of comprehensive validation

One of the critical weaknesses of this article is its lack of comprehensive validation for the downscaling methods. Validating the granular-level HDI estimates is crucial, as the approach relies on HDI "estimates" data from first-level administrative units as the primary training dataset. These first-level HDI estimates are not ground-truth data but rather modeled outputs derived from extensive household survey data. Consequently, the secondary estimates generated through this approach require robust and rigorous validation to ensure their reliability and applicability.

In Section 2, the authors employ two strategies to validate their method: comparing their HDI estimates with survey data from Mexico and testing their model on two other wealth-related indicators, the International Wealth Index and Nighttime Lights. As computer vision analysts in the satellite imagery domain, we understand the inherent challenges of evaluating grid-level satellite imagery features due to the scarcity of high-resolution ground-truth datasets across diverse countries. However, these two strategies are insufficient to fully establish the credibility of their results and their utility for policy-making. First, validating the estimates using data from only one country is inadequate. While high-resolution HDI ground-truth data is rare, there are countries that publish components of the HDI at the municipal level, which could enhance the robustness of the validation. For instance, several East Asian countries report 'years of schooling' at the municipal level, and many high-income countries (e.g., Canada) publish municipal-level data on life expectancy at birth. While constructing fully comprehensive ground-truth HDI datasets may not be feasible, these individual indicators can serve as valuable proxies for evaluating the model's performance more comprehensively across different geographic and socioeconomic contexts. Second, we are not fully convinced that testing the model on other wealth-related indicators can provide sufficient theoretical or empirical evidence to validate the model's utility for HDI estimation. While these indicators may correlate with wealth or development, they do not directly measure the multidimensional components of HDI, such as education, life expectancy, and income. As a result, such tests fail to rigorously demonstrate the model's ability to accurately capture the complexities of HDI and its applicability for real-world policy-making.

2.2. Lack of time-series validation

The paper appears to validate the estimates from 2012 to 2021 individually by year (Figure S8). However, since the results span from 2012 to 2021, additional validation is needed to assess how well the model captures temporal changes. Specifically, the validation should assess whether the year-to-year variations in the HDI estimates align with actual HDI data. This involves verifying the increases or decreases in HDI over time against reliable ground-truth data to ensure that the model accurately captures temporal trends and dynamics. As previous studies have argued [1], validating time-series changes and temporal predictions from satellite imagery is inherently more challenging than estimating a single time point. For instance, if the performance at a single time point is 0.6, the performance across two time points may decrease due to textual differences between satellite images taken in different years and months.

2.3. Downscaling of the composite HDI index versus individual components

Another issue of concern is the (main) manuscript's methodological focus on downscaling the HDI as a composite index, rather than separately downscaling its individual components and subsequently aggregating them into a single estimate. As evidenced in Table S4 of the manuscript, the components of the HDI demonstrate low correlations and distinct distributions within countries. For example, the Pearson correlation between expected years of schooling and life expectancy is only 0.1. This indicates that treating the HDI as a unified index may obscure critical spatial variations in its underlying dimensions, potentially reducing both the precision and interpretability of the estimates. While the authors provide the performance of their model for each HDI component in the supplementary materials (Table S3), we argue that these results should be elevated to the main manuscript to emphasize their significance and address these methodological concerns. Also, in our view, while the HDI index is meaningful in itself, it is even more important to understand the different forms of development across various dimensions. For instance, even with the same HDI score, one region may exhibit a high education index but low life expectancy, while another may show high income but low education scores. These varying patterns of development require different policy responses. Therefore, we believe that producing granular-scale estimates for each individual component of HDI would be more valuable, as it would allow for targeted interventions.

- [1] Burke M. et al. (2021). Using satellite imagery to understand and promote sustainable development. *Science*, 371, eabe8628.

3.1. Baseline comparisons

The paper assumes that the model predictions explain a fairly significant portion of HDI variations. For example, it reports that model predictions explain 52% of within-country provincial variation in HDI. However, the authors do not discuss whether explaining 52% (or other numbers) of the variation is sufficient to consider the model reliable for policy applications. This omission raises questions about the adequacy of the model's performance for informing HDI-related policies. One key challenge in evaluating this performance is the lack of a baseline comparison with other models capable of estimating HDI

or related indices. While there may not be a prior model specifically designed for HDI estimation using satellite imagery and machine learning (SIML), there are existing SIML approaches that could potentially be adapted for HDI prediction. For example, Relative Wealth Index (RWI) in Mexico constructed from Meta [1] can be used for the granular-level baseline comparisons in Table S1. Establishing such a baseline would provide critical context for interpreting the reported performance and determining its practical significance.

[1] Chi, G., Fang, H., Chatterjee, S., & Blumenstock, J. E. (2022). Microestimates of wealth for all low-and middle-income countries. *Proceedings of the National Academy of Sciences*, 119(3), e2113658119.

3.2. Ablation study: Excessive Contribution of Nightlight Data to the Full Model's Performance

As described in the ridge regression, MOSAIKS and nightlight are the two primary features used for estimating HDI in a given region. Table S1-S3 showcase the model performance with ablations, where each component is excluded. However, upon reviewing the data in Table S1, it appears that the NL standalone model shows comparable performance to the full model in terms of ρ^2 and R^2 . In some cases the NL model even overperforms the full model. For example, when predicted at municipality level in Mexico, within-country performance of MOSAIKS+NL presents 0.5 ρ^2 (0.45 R^2) while that of NL is 0.58 ρ^2 (0.5 R^2). In light of these results, the authors are encouraged to provide evidence supporting the meaningful contribution of MOSAIKS in the full model.

4.1. Same weights β for linearization at tile, province, municipality levels.

The model assumes that the weights (β) representing the relationship between satellite imagery features and HDI are consistent across different spatial scales, such as tiles, provinces, and municipalities. However, this assumption—that the same β applies regardless of scale—does not appear to have been explicitly validated in this study. Given the differences in spatial aggregation and scale, it is critical to test whether this assumption holds, as it directly impacts the robustness and reliability of the model's estimates.

4.2. Methodological novelty

Another methodological issue is the heavy reliance of this study's model on the MOSAIK framework, which was previously published in *Nature Communications* [1]. While it is noteworthy that this paper applies the approach to estimating different datasets (i.e., HDI) and provides the first global estimates at the second administrative level, it is difficult to identify significant methodological advancements compared to the earlier work. The primary novelty lies in the application context rather than in the development of the modeling approach itself, raising questions about the extent of its contribution beyond the adaptation to a new target variable.

If the methodological advancements are not substantial, it raises the question of whether the work would be better suited for publication as a scientific data paper rather than as a full research article. In such a case, the focus should be on rigorously validating the dataset to ensure its credibility and utility for future research and applications, rather than emphasizing methodological innovation.

[1] Rolf, E. et al. (2021). A generalizable and accessible approach to machine learning with global satellite imagery. *Nature Communications*, 12(1), 1-11.

(Remarks on code availability)

Reviewer #2

(Remarks to the Author)

I co-reviewed this manuscript with one of the reviewers who provided the listed reports. This is part of the *Nature Communications* initiative to facilitate training in peer review and to provide appropriate recognition for Early Career Researchers who co-review manuscripts.

(Remarks on code availability)

Reviewer #3

(Remarks to the Author)

The paper "Global high-resolution estimates of the United Nations Human Development Index using satellite imagery and machine learning" contains the first granular estimates of the UN's Human Development Index. The paper develops a downscaling technique that uses provincial estimates to generate estimates at a more granular level. The paper is very well-written, to the point, and contains numerous relevant and convincing robustness checks. The authors anticipated and addressed many of my questions related to the interpretability of the model, its performance over time, its performance by HDI dimension, and whether the relationship between the features and the outcome is geographically stable. I have three comments, which I think could help improve the relevance and applicability of the model.

1. Use of multiple imputation techniques.

The authors point out that the predictions have a much lower variance than the true distribution, which is also evident from several of the panels of Figure 2. If I am not mistaken, this means that the model systematically underpredicts the extent to which non-poor provinces have poor municipalities, as well as the extent to which poor provinces have non-poor municipalities. This seems to be an important issue given that perhaps the main purpose of the paper (as illustrated with the

targeting example in Mexico) is to uncover such situations. Multiple imputation and small area estimation techniques have existed for decades to deal with the issue that the variance of the predicted values is too small. Could the authors use such techniques to recover more plausible predictions? I realize that small area estimation applications predict at the same level as which ground-truth data are available, so the authors are in a different situation here. Yet surely there are some assumptions about the error term one could make to recover more realistic predictions, perhaps utilizing the Mexican data where the true distribution of municipal estimates is available. If not attractive, and no attractive alternative exists, please explain why.

2. Discussion of advantages/disadvantages for targeting relative to alternative approaches

The policy relevance of the paper clearly comes from improved within-country targeting of resources, as cross-country targeting of resources tends to rely on country-level data. Yet it is unclear when countries armed with these estimates would use them for targeting resources. Countries with administrative data would probably use those for targeting, while countries with recent census and survey data would probably rely on small-area estimation techniques to generate granular measures of well-being (I realize small area estimation techniques rarely if ever have the HDI on the left-hand side, but they could have any multidimensional measure of well-being that can be recovered from surveys). What are the cases the authors envision the data generated could be applied? Perhaps in countries with no census, no survey data, no admin data? Or do you think this method works better than small-area estimation techniques? If so, what is the evidence to that end? This paper might be of interest for the discussion: <https://doi.org/10.1016/j.jdeveco.2024.103352>

3. Interpretation of the model's error

The authors conduct numerous different and relevant checks to test the performance of their model. The overarching challenge is that an ideal assessment of the model would require true granular HDI estimates which, except for in one circumstance, do not exist. The one genuine test of the error they can make suggests that the performance is moderate at best (Figure 2D), with an R^2 of 0.29, while the other one that comes close (2G) is likewise not too great. I think the authors ought to place greater emphasis on the point that the model fundamentally cannot be evaluated in a first-best manner and that second-best approximations suggest that one-third of the relevant variance can be predicted by the model.

(Remarks on code availability)

Reviewer #4

(Remarks to the Author)

Summary

The manuscript by Sherman et al. presents a novel method to downscale the United Nations Human Development Index (HDI) using a combination of high-resolution daytime and nighttime satellite imagery and machine learning. The study addresses the critical issue of aggregation bias in traditional HDI measurements by demonstrating that a significant fraction of the global population is misclassified when relying solely on country- or province-level data. Validation using survey-based estimates in Mexico and comparisons with alternative socioeconomic indicators (e.g., the International Wealth Index and nighttime lights) support the robustness of the proposed approach. Overall, the work offers a promising avenue to support more spatially precise policymaking and contributes substantially to the literature on satellite-based socioeconomic measurements.

Major Comments

1. Introduction:

The paper assumes how HDI is constructed at the national and municipal levels, but it would be beneficial to explicitly mention the range of HDI (0 to 1), its variation across countries, and the weights assigned to each component (education, health, and income).

In Mexico, municipal-level HDI estimation relies on the population census for education and health, but the income variable is estimated using small area estimation techniques (Elbers, Lanjouw, and Lanjouw 2003). These estimations are inherently uncertain. It would strengthen the paper to contrast the proposed method with these techniques, highlighting not only the limitations of census data for small areas but also the challenges in obtaining statistically valid income estimates at the municipal level.

2. Methodological Clarity:

While the authors discuss the MOSAIKS method, a clearer explanation with a more accessible example would enhance understanding. Figure 1 provides a visual representation, but the role of colors and the integration of socioeconomic variables could be explained more explicitly. A simplified schematic or a step-by-step breakdown of Figure 1 would aid readers unfamiliar with MOSAIKS in grasping how the method links coarse provincial data with fine-resolution predictions.

The paper could delve deeper into the comparison between the proposed method and small area estimates (SAE). It would

be valuable to discuss whether the ML-based downscaling approach offers advantages over SAE, particularly in terms of variance and accuracy.

The paper does not expand on the consistency of estimates in well-being over time. Specifically, it would be insightful to investigate instances where the official HDI shows an increase, while alternative measures like SIML suggest a decrease (or viceversa). Are there such cases?

(Remarks on code availability)

The link works and seems to be very informative: github.com/Global-Policy-Lab/hdi_downscaling_mosaiks
I did not install the files, but it seems that everything is reproducible.

Version 1:

Reviewer comments:

Reviewer #1

(Remarks to the Author)

I appreciate the authors' efforts and the potential value of high-resolution HDI-related data. I have carefully evaluated the authors' rebuttal and revised materials in light of my initial review. While I appreciate the authors' efforts, the key concerns remain insufficiently addressed. In my judgment, addressing these issues would require substantial new work and re-framing that goes beyond a reasonable revision and would effectively constitute a new study.

Here are three issues that I am not still convinced by the manuscript and authors' responses.

1. Low model performance

Model performance at the target spatial scales is too low to support the main claims and intended uses. The manuscript validates municipality-level predictions using province-level HDI data from three countries (Indonesia, Brazil, Mexico). The authors focus on "explaining within-province variation" (manuscript, line 330), and this is the "preferred metric in (their) paper because (they) are interested in explaining fine-resolution variation in HDI within provinces that has yet to be previously estimated at global scale" (response, pp. 15-16).

However, the reported within-province predictive performance averages about $R^2 \approx 0.34$ (Brazil 0.20, Mexico 0.29, Indonesia 0.53). This is substantially below what is commonly seen and expected in the satellite-based socioeconomic prediction literature for policy-relevant applications (typically $R^2 \geq 0.5$). Given this level of performance and the scale mismatch in validation, the headline claims—particularly regarding widespread misclassification of HDI quintiles and improved decision-making—are not sufficiently substantiated.

In addition, a simple, available baseline outperforms the authors' approach in at least one country. In my first-round review, I requested a within-province baseline using Meta's Relative Wealth Index (RWI) for Mexico (Chi et al., 2022). The authors responded that it was not possible "to calculate a within-province metric" (response, p. 16). Alternatively, I simply computed the within-province relationship between municipal MHDH and the mean RWI by municipality in Mexico ($N = 2,456$; excluding Bacalar due to missing MHDH). MHDH data was accessed through the website of authors. This baseline achieved $R^2 = 0.364$ at the municipality level, exceeding the authors' reported value for Mexico ($R^2 = 0.29$). This indicates that a simple, readily available baseline can match or outperform the proposed approach on the stated task.

2. Lack of Validation for time-series estimates

During the first round of review, I asked the authors to provide proper validation of time-series estimates on HDI. However, the authors stated that "we use the satellite imagery to downscale annual first level administrative HDI estimates to higher spatial resolution, but do not predict or evaluate changes in the within-province distribution of HDI over time. (Response, p. 11)" and "This approach assumes that the spatial pattern of HDI within provinces does not change over time; relaxing this assumption gives similar results (Supplementary Information S3)."

Nevertheless, the manuscript proposes to release annual maps for 2012–2021. In practice, when this manuscript is published in Nature Communications, there are high possibilities for some readers to use these data for time-series comparisons and policy evaluation despite the authors' guideline in the manuscript. Without any temporal validation (e.g., consistency across years, sensitivity to shocks, stability of rankings, calibration of year-to-year changes), this might create a substantial risk of misuse and misinterpretation. Given that the study's primary objective is to characterize fine-scale spatial variation in HDI, what is the rationale for releasing annual maps for 2012–2021?

3. Methodological novelty

I raised the heavy reliance of this study's model on the MOSAIK framework, which was previously published in Nature Communications (Rolf et al., 2021). However, the authors disagree with my concern with several methodological contributions they assert. The authors argue novelty in training and predicting on irregularly shaped and sized polygons to

leverage coarse administrative labels. “Training and predicting on irregularly shaped and sized polygons, which is generally not possible using existing SIML approaches. This addresses a key limitation of the existing SIML literature -- the dearth of available training data -- by allowing models to be trained on coarse but plentiful administrative data. (response p. 22)”. Unfortunately, the similar ideas have been already proposed such as Han et al. (2020).

Other touted advances—training on fine-scale local variation, using a fixed-effects specification, and expanding MOSAICS features from ~1% to near-global coverage (response p. 22)—appear to be incremental refinements and engineering scale-up rather than a substantive methodological contribution. The manuscript’s empirical demonstrations do not, in my view, establish a new methodological paradigm beyond what has been introduced in the literature.

References

- Chi et al. (2022). Microestimates of wealth for all low-and middle-income countries. *Proceedings of the National Academy of Sciences* 119, e2113658119
- Han, S. et al. (2020). Lightweight and Robust Representation of Economic Scales from Satellite Imagery. *Proceedings of the AAAI Conference on Artificial Intelligence*, 34(01), 428-436. <https://doi.org/10.1609/aaai.v34i01.5379>
- Rolf, E. et al. (2021). A generalizable and accessible approach to machine learning with global satellite imagery. *Nature Communications*, 12(1), 1-11.

(Remarks on code availability)

Reviewer #3

(Remarks to the Author)

Thank you to the authors for replying to my last set of questions. I have just two follow-up questions which I think can be handled relatively easily.

1. I am still struggling a bit with the fact that the ranking of predicted values within a country will be biased against poor municipalities in non-poor regions and vice versa. If I am not mistaken, this systematic bias would not arise if the predictions were made in levels instead of deviations from mean values. I understand such predictions would perform worse in terms of R^2 , but perhaps the exclusion/inclusion errors for targeting would be better? It should be easy for the authors to reject this hypothesis with their Mexican, Brazilian, and Indonesian data. If the AUC indeed is worse under this approach, I don't think any further changes are needed to the manuscript on this topic.

2. Thanks to the authors for clarifying when the proposed data can be useful for targeting. I agree that the application illustrates how fine-resolution HDI estimates can improve targeting of policies that would otherwise use coarser provincial measures. Yet I am still unaware of any cases where coarser HDI estimates were used for targeting. It would be helpful if the authors could cite a handful of such cases to document that this is tackling a widespread policy issue. Even if such cases do not exist, I think the fine-resolution HDI estimates have tremendous value, but perhaps more in terms of providing vital data for the research community and increasing our understanding of human wellbeing globally, but less for targeting purposes.

(Remarks on code availability)

Reviewer #4

(Remarks to the Author)

NA

(Remarks on code availability)

Version 2:

Reviewer comments:

Reviewer #1

(Remarks to the Author)

I appreciate the authors' considerable efforts in addressing my comments and for providing well-prepared and detailed responses. The exchange during the review process has been very insightful, and I have gained a clearer understanding of the authors' objectives and interpretations within the manuscript. Their thoughtful replies have also clarified several points of potential misunderstanding on my part. Nonetheless, some points of concern remain, which I believe arise from differences in interpretation and perspective between the authors and myself.

I raised three concerns: low model performance, lack of validation for time-series estimates, and methodological novelty. I appreciate that the authors have addressed the second concern by removing the time-series estimates from the manuscript, which might be a difficult decision. On the first and third concerns, I can partly agree with the authors' position. I

acknowledge that this is largely a difference in perspective rather than a factual dispute. I believe this issue from the first and last concerns could be addressed constructively in the manuscript itself. I recommend that the authors explicitly describe the potential limitations and concerns in the Discussion section.

1a. Low Model Performance (general performance)

Authors' responses:

- "the assertion that novel research in this domain is only publishable if $R^2 \geq 0.5$ is both arbitrary and inaccurate." (Second Responses, p. 2)
- "The Reviewer incorrectly presents the "within-province performance" R^2 metric, which is not the most relevant metric for evaluating within-country HDI distributions, particularly when comparing to prior literature that has been evaluated using a within-country performance metric." (Second Responses, p.2)
- "Finally, the Reviewer's suggestion that the model performance presented is of too low quality to be of interest or use to policy-making applications is not correct." (Second Responses, p.2)

Reviewer's comments:

Firstly, my intention was not to suggest an official or rigid threshold (e.g., $R^2 \geq 0.5$) as a requirement for publication. Rather, my point is that within today's mature SIML research landscape, performance metrics at or above this level have become commonplace, and are thus a natural reference point for evaluating new contributions.

The examples cited by the authors (Jean et al., 2016; Rolf et al., 2021; Yeh et al., 2020) are indeed important milestones in the SIML domain, despite their modest performance metrics. However, it is important to recognize that these studies were pioneering works, published at a time when the combination of satellite imagery and machine learning was still emerging. Their significance lay primarily in opening new research directions rather than in achieving high predictive accuracy per se. In recent literature, the performance achieved is typically higher—for example, Ahn et al. (2023) reported Spearman's $\rho = 0.77$ for grid-level economic development estimates and $R^2 = 0.84$ at a district level (comparable to Figure 2.C of the present manuscript). Similarly, Zheng et al. (2025) reported R^2 values between 0.62–0.83 for within-country wealth index predictions across multiple African nations.

Given the maturity the SIML field has now achieved, expectations for model performance and robustness are naturally higher, especially in high-impact venues. Novel contributions today might reasonably be held to more stringent performance and validation standards than the early works from the 2010s or early 2020s.

Secondly, I acknowledge that my earlier comparison between the authors' within-province performance metric and the within-country performance metrics used in prior literature may have been inappropriate, given the differences in spatial resolution and study objectives. I accept that direct comparison across these metrics can be misleading. However, in this case, the manuscript's stated novelty lies in its downscaling approach and its ability to explain fine-resolution, "within-province variation" in HDI. If this level of granularity represents the core methodological contribution, then it is reasonable to expect the within-province performance itself to be sufficiently strong to justify publication in a high-impact journal such as *Nature Communications*. While earlier studies evaluated their models using "within-country metrics" because that matched their research scope, they achieved performance levels that were competitive for their chosen resolution and metric at the time. Similarly, if the focus here is at the "within-province level", then comparable competitiveness should be expected within that resolution context—both in terms of predictive accuracy and demonstration of robustness.

My biggest concern still lies in the cases such as Brazil with 0.2 within-province R^2 although the average for countries is 0.34 (within-province) or 0.51 (within-country). This means the model might explain only about 20% of the variance at this resolution in a certain country. I am concerned that in policy contexts, such low performance might limit practical usefulness or lead to misinterpretation if the uncertainty is not adequately communicated.

Lastly, I did not mention that the model performance is too low to be of interest to policy-making applications. I mentioned it is "too low to support the main claims and intended uses." (Second review comment 1) In fact, this research topic is highly pertinent, and I agree it could be of significant interest to institutions such as UNDP and other stakeholders. The manuscript clearly addresses a socially important issue, and the potential impact in guiding development policy is substantial. However, while institutional interest (e.g., UNDP's collaboration) is encouraging for dissemination and uptake, it does not, in itself, validate the model's accuracy or suitability for the stated applications without further performance improvement or careful qualification of limitations.

In summary for the first concern (low performance), in the context of recent SIML studies, I would generally expect higher performance than what is reported in the manuscript. While the 0.5 threshold I have referred to in earlier comments is not an official requirement, current literature suggests that this is not an unusually high bar to meet. That said, I acknowledge the authors' perspective that performance at the "within-province" resolution can still be acceptable depending on research aims and context.

On balance, I believe the level of performance reported here is acceptable, provided that the Discussion section explicitly addresses potential issues of interpretation that may arise from these accuracy levels. In particular, I recommend that the manuscript comment on the practical risks and limitations in policy-relevant cases such as Brazil, where the within-province R^2 is approximately 0.20.

References

- Ahn, D., Yang, J., Cha, M. et al. A human-machine collaborative approach measures economic development using satellite imagery. *Nat Commun* 14, 6811 (2023). <https://doi.org/10.1038/s41467-023-42122-8>
- Zheng, Z., Burke, M., Ermon, S., Kilic, T., Lee, R., Lobell, D.B., Newhouse, D. and Wu, T., 2025. Dynamic, High-Resolution Wealth Measurement in Data-Scarce Environments. World Bank.

1b. Low Model Performance (comparison with Chi et al.)

Authors' responses:

- "the Reviewer's proposed comparison is to regress municipal HDI in Mexico directly on the RWI constructed by Chi et al., thereby training and evaluating a model on the municipal data. This, however, is incomparable with all of our estimates because the Reviewer's procedure skips the difficult-but-essential downscaling part of the problem. [...] From the perspective of the problem that the manuscript is solving, the Reviewer's proposed test "cheats" by training the model directly on the unobservable municipal data." (Second Responses, p. 4)

Reviewer's comments:

I appreciate the authors' detailed response to my suggestion regarding comparison with Chi et al.'s RWI in Mexico. I understand that from the authors' methodological perspective, using RWI to estimate municipal-level HDI may be considered to "cheat" the difficult but essential downscaling step, and is therefore not directly comparable to the approach presented in the manuscript.

However, I would like the authors to also consider this issue from the perspective of an end user, such as a policymaker in Mexico. In a real policy setting, the priority is often to use the most accurate and validated local-level statistic available — whether produced by downscaling or more direct regression — in order to identify disadvantaged regions and allocate resources effectively. For example, RWI is available at the grid level and can serve as a strong predictor of municipal HDI without requiring downscaling in contexts where municipal data are already available.

In countries such as Mexico, Brazil, and Indonesia, where municipal-level HDI data are accessible, it is entirely feasible and valid for end users to use direct regression-based approaches or other high-correlation proxies without addressing the downscaling problem. In contrast, in countries without municipal-level HDI, the quality and trustworthiness of downscaled estimates become far more important. In my view, downscaled HDI in such countries should be considered for policymaking only if adequate validation is presented, comparable to the validation done for Mexico, Brazil, and Indonesia.

I recognize that this represents a difference in perspective: the authors argue that validation in three countries is sufficient to infer reliability in others without municipal data, whereas my position is that validation should be demonstrated country-by-country before such data are used for policy purposes in a certain country. I believe that this concern needs to be mentioned at somewhere in the Discussion, so that readers understand the potential limitations.

Authors' responses:

- "Reviewer's proposed procedure would not be implementable in the other 176 countries where we produce HDI estimates. Thus the proposed baseline is not valid since it cannot be implemented globally, and the explicit purpose of the paper is to produce global fine-resolution estimates of HDI, which currently do not exist." (Second Responses, p. 5)

Reviewer's comments:

I acknowledge the authors' point that my proposed RWI-based procedure would not be implementable in the other 176 countries for which they produce HDI estimates, whereas their downscaling approach enables the creation of global, fine-resolution HDI estimates — a product that, to my knowledge, does not currently exist.

However, I would like to underline that this represents a difference in perspective rather than a disagreement over technical details. From the authors' viewpoint, producing global coverage in itself is a sufficient justification for the value of the dataset, even if country-specific validation is limited to a subset of nations. From my perspective, global fine-resolution HDI estimates are of limited policy utility in any given country unless they are accompanied by adequate validation for that specific national context.

Authors' responses:

- "Third, if the Reviewer's proposed approach were assumed to represent a valid comparison (which we do not believe is correct), then its implementation by the Reviewer is neither complete nor fair. [...] Replicating the procedure proposed by the Reviewer results in a within-province performance of $R^2 = 0.50 / 0.26 / 0.37$ for predicting municipal HDI in Indonesia / Brazil / Mexico using Meta's RWI" (Second Responses, p. 5)

Reviewer's comments:

I appreciate the authors' clarification regarding my earlier suggestion and the replication of the procedure I proposed. As noted, my second-round R^2 results were based solely on municipal-level HDI estimates in Mexico, as that was the only case for which I had access to the necessary data. The authors' replication demonstrates that, in Mexico, R^2 using Meta's RWI is higher ($R^2 = 0.31$) than with the authors' approach. However, when considering the average within-province R^2 values across the three countries assessed — 0.38 for RWI compared to 0.39 for the authors' method — I acknowledge that, on balance, the authors' approach yields slightly better average performance than this potential baseline. Therefore, I agree that, in the context of these three countries, the presented method performs at least as well or marginally better than the proposed RWI-based alternative.

Moreover, I agree with the authors that using RWI for HDI estimation can be considered an "apples-to-oranges substitution" (Second Responses, p.6), since it bypasses the downscaling process central to the manuscript's methodology. At the same time, I would note that all SIML-derived HDI estimates — regardless of modeling approach — inherently represent "apples-to-oranges" substitutions when compared with official HDI statistics constructed from household survey data. Both RWI-based and other SIML-based methods are proxies, and performance differences are therefore best understood as degrees of alignment to survey-based ground truth, not as direct measurements.

Authors' responses:

- "Third, the Chi et al. estimates are not global, and only cover low and middle income countries, which limits the

applications that they can be used for.” (Second responses, p. 6)

Reviewer’s comments:

I respectfully disagree with the authors’ view that the Chi et al. estimates have limited utility because they are not global and cover only low- and middle-income countries. As noted in much of the SIML literature, satellite-based socioeconomic estimates are often most valuable precisely in low- and middle-income countries, where conventional statistics are sparse or outdated. In such contexts, these proxy measures can fill critical data gaps and support evidence-based decision-making. Moreover, as I mentioned in my first-round review, there are municipal-level data for HDI-related indicators (e.g., literacy rate, life expectancy) in several high-income countries, such as South Korea. Other countries in the Global North typically have both the resources and institutional capacity to conduct municipal- or district-level surveys for HDI or component statistics. As such, the added value of satellite-based downscaled HDI estimates may be greatest in settings where official local-level data do not already exist.

2. Lack of Validation for time-series estimates

Thank the authors again for removing the time-series estimation component from the manuscript. I recognize that this may have been a difficult decision, given the potential appeal of time-series data. I believe this revision improves the clarity and focus of the paper in its current form. I also hope that the authors will consider developing and validating time-series HDI estimates in future studies, as such work could make a valuable contribution to the field once robust validation methods are available.

3. Methodological novelty.

Authors’ responses:

- “The Reviewer’s statement above is removed from context and only describes the training based on irregular polygons as if that were the only point of the paper, ignoring the difficult-but-central downscaling.” (Second response, p. 8)

Reviewer’s comments:

The authors state in their response that my earlier comment “is removed from context” and portrays training on irregular polygons as the only point of the paper, “ignoring the difficult-but-central downscaling.” (Second Response, p. 8) I respectfully disagree with this characterization. In the main manuscript, the authors themselves clearly present the ability to utilize irregularly shaped administrative data as a key methodological advance:

“This limitation is partly due to the design of modern SIML methods, since large quantities of administrative data are available, but existing systems are generally not designed to make use of them.” (lines 47 – 48)

“This restriction has caused prior studies to rely on coarse approximations for linking irregularly shaped labels to corresponding imagery, for example, by averaging polygon labels that overlap with the square image” (lines 54 – 56)

In their First Response (p. 22), the authors again emphasize:

“Training and predicting on irregularly shaped and sized polygons, which is generally not possible using existing SIML approaches. This addresses a key limitation of the existing SIML literature -- the dearth of available training data -- by allowing models to be trained on coarse but plentiful administrative data.” (First response p. 22)

Across these statements, the irregular-polygon approach is positioned as a methodological innovation in its own right, not solely as a supporting detail within the downscaling framework. My earlier comments on novelty were therefore based on the authors’ own framing of this capability as a central contribution, alongside the downscaling methodology.

After reading the Second Response, I acknowledge that Han et al.’s aim and the authors’ aim differ: the authors seek “to address the problem of downscaling administrative data that originates from irregular polygons” (Second Response, p. 8), while Han et al. aim to “obtain essential spatial representation for any given district from high-resolution satellite imagery based on deep neural networks” (Han et al., 2020, abstract). Nonetheless, Han et al. appear to address the challenge of training (at least) on irregularly shaped and sized polygons—not necessarily within a downscaling approach—through “learning the critical spatial characteristics of arbitrary size areas and representing such characteristics in a fixed-length vector.” (Han et al., abstract)

Finally, while the authors also mentioned other advances—training on fine-scale local variation, use of a fixed-effects specification, and expansion of MOSAICS features from ~Y1% to near-global coverage (First Response, p. 22)—I still regard these as incremental refinements and engineering scale-up, rather than major methodological breakthroughs.

Again, I acknowledge this difference in view remains primarily a matter of perspective. At this stage, I believe further back-and-forth would likely restate the same arguments, and I therefore leave it to the editor or other reviewers to determine whether the manuscript meets the journal’s standards for publication.

Han, S. et al. (2020). Lightweight and Robust Representation of Economic Scales from Satellite Imagery. Proceedings of the AAAI Conference on Artificial Intelligence, 34(01), 428-436. <https://doi.org/10.1609/aaai.v34i01.5379>

(Remarks on code availability)

Response to Reviewer Comments for “Global high-resolution estimates of the United Nations Human Development Index using satellite imagery and machine learning”

Reviewer comments in *blue italics*.

Our replies in black.

Reviewer #1 (Remarks to the Author):

In the article “Global high-resolution estimates of the United Nations Human Development Index using satellite imagery and machine learning,” the authors present a MOSAICS-based approach to estimate the Human Development Index (HDI) at municipal and grid-level spatial resolutions using satellite imagery. Their methodology involves a downscaling technique that utilizes province-level HDI data to train the model, enabling predictions at finer spatial scales for municipalities of varying shapes and sizes, and also a grid level with 0.1x 0.1 degree. This work aims to address critical gaps in sub-national HDI data, uncovering disparities often overlooked by traditional methods. The study holds potential for bridging global development data gaps, with implications for informed policy-making and equitable resource allocation, particularly in underserved regions, emphasizing the importance of high-resolution HDI estimates. If this paper is accepted, their estimates are likely to be widely used in future research and policy-making, potentially affecting populations influenced by their outcomes. Therefore, this study requires a very high standard of rigor and thorough validation. However, we believe that this article needs significant revisions before publication in Nature Communications. We would like to begin the first round of review by sharing the eight primary concerns. Once these concerns have been adequately addressed, we believe it will be appropriate to proceed with a detailed review of the methods and their specific aspects.

We thank the reviewer for their thoughtful and insightful review. Please find our responses to these eight primary concerns below.

1.1. Comprehensive review on related studies and comparison needed

As the author describes in the paper, there have been numerous efforts to measure socio-economic indicators related to the Sustainable Development Goals (SDGs) using satellite imagery. For instance, prior studies have focused on estimating indicators such as Gross Domestic Product (GDP) [1], poverty levels [2], and asset wealth [3]. This paper focuses on the measurement and downscaling of the Human Development Index (HDI), presenting the first global estimates of HDI at the second-level administrative units. However, an additional consideration arises as follows:

*Their main contribution is the first instance of estimating HDI (and its components) using satellite imagery on a **global** scale. However, there have been several attempts that made a*

prediction for specific areas, such as country[4] or regional[5] level. [4] measured HDI for county regions corresponding to ADM2 level in China, and [5] measured HDI for the East Java region in Indonesia.

Especially, in a prior study [5], a more diverse range of remote sensing data sources was employed to measure HDI, rather than relying solely on MOSAICS. For example, NDVI (Normalized Difference Vegetation Index), NDWI (Normalized Difference Water Index), and NDBI (Normalized Difference Built-up Index) were derived from Sentinel-2A bands to provide information on vegetation, water, and built-up areas, respectively. Additionally, datasets such as MODIS and Sentinel-5P were utilized. In contrast, the authors of the current paper conducted regression using MOSAICS combined with VIIRS nightlight imagery, with results presented in Tables S1–S3. Notably, nightlight imagery was the sole baseline used for methodology validation. To enhance the novelty and robustness of their findings, experiments incorporating other globally available remote sensing datasets, such as MODIS or Sentinel, are recommended. Alternatively, the authors should explicitly address why such data were excluded, citing specific limitations such as temporal or spatial resolution constraints.

[1] Engstrom, R., Hersh, J., & Newhouse, D. (2022). Poverty from space: Using high resolution satellite imagery for estimating economic well-being. *The World Bank Economic Review*, 36(2), 382-412.

[2] Jean, N., Burke, M., Xie, M., Davis, W. M., Lobell, D. B., & Ermon, S. (2016). Combining satellite imagery and machine learning to predict poverty. *Science*, 353(6301), 790-794.

[3] Yeh, C., Perez, A., Driscoll, A., Azzari, G., Tang, Z., Lobell, D., ... & Burke, M. (2020). Using publicly available satellite imagery and deep learning to understand economic well-being in Africa. *Nature communications*, 11(1), 2583.

[4] Zhang, X., Xu, J., Zhong, S., & Wang, Z. (2024). Assessing uneven regional development using nighttime light satellite data and machine learning methods: evidence from county-level improved HDI in China. *Land*, 13(9), 1524.

[5] Ramadhan, R., & Wijayanto, A. W. (2023). Integrating Satellite Imageries and Multiple Geospatial Big Data for Granular Mapping of Spatial Distribution of Human Development Index in East Java, Indonesia. In *Proceedings of The International Conference on Data Science and Official Statistics* (Vol. 2023, No. 1, pp. 274-295).

We thank the reviewer for the comment. These referenced contributions are now all cited in the manuscript (e.g., lines 38-39). We especially thank the reviewer for the reference to Ramadhan et al. (2023), which led us to municipality-level HDI data that we were previously unfamiliar with. We use this data to further validate our HDI estimates, as discussed in our response to the reviewer's second comment below.

To address the reviewer’s concern, we now implement the reviewer’s suggestion to incorporate other globally available remote sensing datasets into the analysis. Following Ramadhan et al. (2023), we add three indices that the reviewer suggests (NDVI, NDWI and NDBI from the Sentinel satellite) to our model. We describe this analysis in the discussion on lines 331-344:

Here, we use satellite imagery due to its rich information content, global availability, and near uniformity in quality across countries. To evaluate whether additional satellite data sources beyond visual daytime imagery and nighttime imagery might improve HDI estimates, we follow ref. (22) and add to the baseline model three additional index-based features constructed from Sentinel-2A satellite imagery that integrate knowledge of spectral properties of different ground conditions: the Normalized Difference Vegetation Index, Normalized Difference Water Index, and Normalized Difference Built-Up Index. We find that the model performance is essentially unchanged (Table S5), indicating that these additional features do not provide additional information beyond what is already captured by the MOSAIKS and NL features. While it is important to note that there is a limit to how well socioeconomic variables can be predicted using satellite imagery generally, and adding these additional features did not improve performance in this case, future work should nonetheless explore whether incorporating additional imagery sources and/or other ancillary data can improve these estimates (52).

The new Table S5 is reproduced below:

HDI trained at:	Features	Predicted at province level			
		Full variation performance		Within-country performance	
		ρ^2 (1)	R^2 (2)	ρ^2 (3)	R^2 (4)
Within-country (n=1,363)	NDVI+NDWI+NDBI	0.92	0.92	0.03	0.02
	NDVI+NDWI+NDBI+MOSAIKS+NL	0.96	0.96	0.52	0.52
	MOSAIKS+NL (for reference)	0.96	0.96	0.52	0.52

Table S5: Performance for models trained with additional features at the provincial level. Specifically, we use Sentinel 2A imagery downloaded at approximately 500m resolution (67). Following, (22) we process Sentinel 2A imagery to calculate the Normalized Difference Vegetation Index (NDVI), Normalized Difference Water Index (NDWI), and Normalized Difference Built-up Index (NDBI). We then create population-weighted features in the same manner as done with the NL features (see Methods 3.2).

Because the introduction of these additional features did not affect performance, we have chosen to retain the model using MOSAIKS and nighttime light features as our primary specification.

Given that the field combining satellite imagery and machine learning is growing rapidly, we nonetheless hope that future research can build upon this approach and the global fine-resolution estimates of HDI it provides.

2.1. Lack of comprehensive validation

One of the critical weaknesses of this article is its lack of comprehensive validation for the downscaling methods. Validating the granular-level HDI estimates is crucial, as the approach relies on HDI “estimates” data from first-level administrative units as the primary training dataset. These first-level HDI estimates are not ground-truth data but rather modeled outputs derived from extensive household survey data. Consequently, the secondary estimates generated through this approach require robust and rigorous validation to ensure their reliability and applicability.

In Section 2, the authors employ two strategies to validate their method: comparing their HDI estimates with survey data from Mexico and testing their model on two other wealth-related indicators, the International Wealth Index and Nighttime Lights.

As computer vision analysts in the satellite imagery domain, we understand the inherent challenges of evaluating grid-level satellite imagery features due to the scarcity of high-resolution ground-truth datasets across diverse countries. However, these two strategies are insufficient to fully establish the credibility of their results and their utility for policy-making.

First, validating the estimates using data from only one country is inadequate. While high-resolution HDI ground-truth data is rare, there are countries that publish components of the HDI at the municipal level, which could enhance the robustness of the validation. For instance, several East Asian countries report ‘years of schooling’ at the municipal level, and many high-income countries (e.g., Canada) publish municipal-level data on life expectancy at birth. While constructing fully comprehensive ground-truth HDI datasets may not be feasible, these individual indicators can serve as valuable proxies for evaluating the model’s performance more comprehensively across different geographic and socioeconomic contexts.

Second, we are not fully convinced that testing the model on other wealth-related indicators can provide sufficient theoretical or empirical evidence to validate the model’s utility for HDI estimation. While these indicators may correlate with wealth or development, they do not

directly measure the multidimensional components of HDI, such as education, life expectancy, and income. As a result, such tests fail to rigorously demonstrate the model's ability to accurately capture the complexities of HDI and its applicability for real-world policy-making.

We thank the reviewer for this suggestion and agree that further validation of the downscaled estimates would build greater confidence in their reliability and applicability. We also agree that the ideal validation exercise would be to compute HDI from survey data from municipalities globally, and then compare that with our remotely sensed estimates of HDI. However, as the reviewer notes, these data do not exist. Nonetheless, in an effort to address the reviewer's concern, we have obtained two additional large-scale survey-based municipal estimates of HDI for Indonesia and Brazil. We emphasize that identifying and obtaining these data took significant effort; for example, the data from Brazil was obtained through personal contacts at the United Nations Development Programme. The main text now uses these data similarly to how the data from Mexico was previously used. This means we have expanded the population for which we conduct the near-ideal validation from 130 million people (Mexico only) five-fold to 620 million people (adding Brazil and Indonesia). To our knowledge, this now represents the largest dataset of survey-based municipal HDI estimates.

We now discuss model performance comparing our satellite-based estimates with these survey-based estimates on lines 159-177:

Downscaling HDI in Mexico, Brazil and Indonesia

As a direct evaluation of HDI downscaling performance, we compare municipal HDI predictions from the satellite-based model trained on provincial HDI deviations from the country mean to municipal HDI derived from census-based calculations in Mexico (9), Brazil (35), and Indonesia (34) (Methods 2.3). In Mexico, downscaled HDI predictions explain 45% of the municipal HDI variation overall (Figure 2C) and 29% of the within-province variation (Figure 2D). In Brazil, HDI predictions explain 48% of municipal HDI variation overall, and 20% of the within-province variation. And in Indonesia, HDI predictions explain 61% of the municipal HDI variation overall and 53% of the within-province variation. It is encouraging that HDI predictions align better with census-based measures for Indonesia than for Mexico or Brazil because measures for Indonesia are from 2019, which aligns with our satellite-based predictions of within-province HDI variation, while measures for Mexico and Brazil are from 2010. These older HDI measures could differ from our satellite-based predictions in part due to changes in HDI since the measurements were taken. Additionally, some portion of the misalignment

between satellite-based predictions and survey estimates likely arises as a result of errors in the survey data itself, a widely recognized issue (15) that we are unable to assess. Together, these results indicate that our method for SIML-based downscaling improves our understanding of the spatial distribution of HDI in these three example countries; although, it is not a complete substitute for survey-based estimates when such data are available.

We have also now added Brazil and Indonesia to the second row of Figure 2, where previously we only plotted data from Mexico:

Figure 2: MOSAIKS models perform well predicting socioeconomic indicators, including at downscaled resolution. (A) Observed and predicted HDI at the province level. Note that the within-country variation is smaller than the across-country variation, as illustrated by France, in yellow, and Ethiopia, in pink. (B) The same as (A), evaluating within-country variation. Provincial deviations from the country mean for France and Ethiopia are now centered at 0, and the model is evaluated on how well it can differentiate provinces that are relatively well and worse-off within countries. (C-D) Observed and predicted municipal HDI data in Mexico, Brazil and Indonesia. (E-G) Observed and predicted IWI at the DHS cluster level. (H-J) Observed and predicted nighttime lights at the municipality level. The vertical streaking in (H-J) are caused by countries that have very spatially dense municipalities (Supplementary Information S6.3). Predictions are anchored to known country or provincial means (Methods 4.4). All predictions are for the year 2019.

These new results are also now reported in Table S1, which has been modified to accommodate the additional analysis:

Predicted at province level (n=1,381)						
	Full variation performance		Within-country performance			
	ρ^2	R^2	ρ^2	R^2		
HDI trained at:	(1)	(2)	(3)	(4)		
Within-country (n=1,363)	0.96	0.96	0.52	0.52		
Province level (n=1,381)	0.83	0.83	0.44	0.24		
Country level (n=145)	0.74	0.74	0.28	< 0		
Predicted at municipality level in Indonesia (n=505)						
			Within-country performance		Within-province performance	
	ρ^2	R^2	ρ^2	R^2	ρ^2	R^2
HDI trained at:	(3)	(4)	(5)	(6)		
Within-country (n=1,363)			0.62	0.61	0.53	0.53
Province level (n=1,381)			0.5	0.37	0.51	0.37
Country level (n=145)			0.35	< 0	0.36	< 0
Predicted at municipality level in Brazil (n=5,584)						
			Within-country performance		Within-province performance	
	ρ^2	R^2	ρ^2	R^2	ρ^2	R^2
HDI trained at:	(3)	(4)	(5)	(6)		
Within-country (n=1,363)			0.48	0.48	0.33	0.20
Province level (n=1,381)			0.46	0.36	0.31	< 0
Country level (n=145)			0.29	< 0	0.18	< 0
Predicted at municipality level in Mexico (n=2,457)						
			Within-country performance		Within-province performance	
	ρ^2	R^2	ρ^2	R^2	ρ^2	R^2
HDI trained at:	(3)	(4)	(5)	(6)		
Within-country (n=1,363)			0.5	0.45	0.31	0.29
Province level (n=1,381)			0.31	0.31	0.26	0.23
Country level (n=145)			0.16	< 0	0.12	< 0
Predicted at DHS cluster level (n=51,996)						
	Full variation performance		Within-country performance		Within-province performance	
	ρ^2	R^2	ρ^2	R^2	ρ^2	R^2
IWI trained at:	(1)	(2)	(3)	(4)	(5)	(6)
Within-country (n=862)	0.71	0.67	0.56	0.56	0.36	0.36
Province level (n=862)	0.44	0.28	0.21	0.08	0.24	0.1
Country level (n=85)	0.31	< 0	0.13	< 0	0.13	< 0
Predicted at municipality level (n=62,536)						
	Full variation performance		Within-country performance		Within-province performance	
	ρ^2	R^2	ρ^2	R^2	ρ^2	R^2
NL trained only on MOSAIKS at:	(1)	(2)	(3)	(4)	(5)	(6)
Within-country (n=2,852)	0.73	0.7	0.64	0.63	0.61	0.57
Province level (n=2,852)	0.65	0.61	0.57	0.45	0.53	0.35
Country level (n=170)	0.44	0.3	0.36	0.08	0.31	< 0

Table S1: Performance for models trained to predict HDI, IWI, and population-weighted nightlight luminosity (NL). Models for HDI and IWI use a combination of MOSAIKS and population-weighted NL features. We show performance evaluated at the province level for HDI and evaluate downscaled performance for HDI, IWI, and NL. Performance scatters from the within-country models are shown in Figure 2. All predictions are made for the year 2019.

The new data from Indonesia and Brazil are now described in the methods section on lines 493-518:

2.3 Municipality-level HDI

We compare our municipal HDI estimates with census-derived municipal estimates for HDI, where these data exist.

Indonesia Time series estimates of HDI at the municipality level are made publicly available by Badan Pusat Statistik (BPS), the statistics agency of Indonesia (63). We use estimates from 2019, which makes these, to our knowledge, the only available municipal HDI data that come from the same year as our satellite imagery.

Brazil The Human Development Report Office office of the United Nations Development Programme has derived HDI data at the municipality level for Brazil, using census data (35). The most recent year these data are available is 2010, which we use in this analysis. The census-derived data have a different mean than the 2019 HDI data that we use elsewhere in this analysis; though, this has no influence on the within-country or within-province evaluation metrics (Figure 2 C,D) because predictions and observations are demeaned at the country and province level, respectively, before the metrics are calculated.

Mexico Census derived estimates of HDI are also available in Mexico for the year 2010, constructed by ref. (9). As with Brazil, these data have a different mean than the 2019 HDI data; though, again, that has no influence on the within-country or within-province evaluation metrics.

When certain components of HDI are not directly available at the municipal level, producers of these datasets use close proxies. For example the BPS calculates municipal HDI data in Indonesia using data on real expenditure rather than GNIpc (34). And when calculating municipal HDI in Mexico, ref. (9) use the child survival rate rather than life expectancy at birth for the health index and an asset index rather than GNIpc for the standard of living index. Likewise, provincial HDI estimates are also constructed using proxies when direct measurements of HDI components are not available (7). Discrepancies in how HDI is calculated at the municipal and provincial levels may contribute to differences between our satellite-based HDI estimates and these survey-based HDI estimates.

These direct validations of the downscaling approach using observations of subnational HDI from Mexico, and now Indonesia and Brazil as well, reinforce the additional downscaling tests applying the method to nighttime lights and the IWI. In all tests, the approach is able to predict substantial variation in municipal HDI, or related wealth indices, within the provincial units on which the model was trained. This, in combination with the rest of the paper's analyses, provides substantial evidence that the downscaling approach provides new and useful information on HDI at fine resolution globally.

We also agree with the reviewer that IWI and NL differ in important ways from HDI. In the text we write that: "The IWI is an alternative development indicator to HDI that omits measures of education and health." (lines 144-145). And clarify that NL is "a common proxy for economic wellbeing" (line 146).

Nonetheless, to avoid any confusion with readers, we also now clarify (lines 149-153) that:

No test can directly validate the performance for downscaling HDI globally, since the data necessary for such a test do not exist; however, all three of these large-scale tests taken together document the effectiveness of our downscaling strategy in general -- using global socioeconomic data similar to HDI -- and for HDI in particular -- using municipal HDI data for three countries.

Considering our validation exercises together, the provincial tests show that the approach can predict HDI (Fig 2 A-B), as well its components (Table S3), at global scale. The IWI and NL tests show that the approach can downscale socioeconomic indicators similar to, but not the same as, HDI at regional to global scale (Fig 2 E-J). And the municipal HDI tests show that the approach can downscale HDI for three large countries. Thus, we now additionally clarify that (lines 235-238):

While it remains untestable how accurately the approach can downscale HDI globally, we find that the approach can predict substantial global variation in HDI and its components, and that it can downscale both HDI in three countries and variables related to HDI globally with accuracy.

We thank the reviewer for encouraging us to identify and obtain these additional validation data, as it was a major effort that we would not have otherwise

undertaken and we believe that it substantially strengthens our findings and the quality of the analysis.

2.2. Lack of time-series validation

The paper appears to validate the estimates from 2012 to 2021 individually by year (Figure S8). However, since the results span from 2012 to 2021, additional validation is needed to assess how well the model captures temporal changes. Specifically, the validation should assess whether the year-to-year variations in the HDI estimates align with actual HDI data. This involves verifying the increases or decreases in HDI over time against reliable ground-truth data to ensure that the model accurately captures temporal trends and dynamics. As previous studies have argued [1], validating time-series changes and temporal predictions from satellite imagery is inherently more challenging than estimating a single time point. For instance, if the performance at a single time point is 0.6, the performance across two time points may decrease due to textual differences between satellite images taken in different years and months.

Thank you for this comment; we agree with the reviewer's assessment that throughout the literature, changes over time tend to be more difficult to predict using satellite imagery than changes over space. Like the reviewer, we are interested in time-series changes in much of our research; however, we focus this manuscript on estimating fine-resolution *spatial* variation in HDI because existing provincial measures do not resolve within-province differences in human wellbeing, and because spatial variation in HDI is generally 60 times larger than temporal variation. Nonetheless, as described in Methods Section 4.5, we construct fine-resolution global estimates of HDI from 2012-2021 by adding time-constant satellite-based estimates of municipal HDI deviations from the provincial mean onto time-varying survey-based provincial values over this temporal range. That is, we use the satellite imagery to downscale annual first level administrative HDI estimates to higher spatial resolution, but do not predict or evaluate changes in the within-province distribution of HDI over time (lines 683-685):

This approach assumes that the spatial pattern of HDI within provinces does not change over time; relaxing this assumption gives similar results (Supplementary Information S3).

Supplementary Information Section S3 provides further discussion of and analysis on this topic (lines 1076 - 1090):

We focus this manuscript on estimating fine-resolution spatial variation in HDI because existing provincial measures do not resolve within-province differences in human wellbeing, and because spatial variation in HDI is generally larger than temporal variation. We construct fine-resolution global estimates of HDI from 2012-2021 by combining time-varying estimates of provincial HDI from (7) with time-constant estimates of the local (i.e. within-province) distribution of HDI based on satellite imagery. Here, we conduct an additional experiment where we test whether allowing HDI to vary over time locally might improve model performance. Specifically, we compare our primary estimates (Figures 2 A-B and S8 A-B), which are made assuming a time-constant distribution of provincial HDI with another set of estimates that are made allowing for a time-varying distribution of provincial HDI (Figure S8 C-D). We find that these two approaches have near identical performance predicting historical HDI from 2012-2021. This motivates our use of the simpler model that assumes an approximately time-constant distribution of HDI as our primary specification when constructing municipal and grid-level estimates. Details of this analysis are described below...

To further clarify this for readers we have now updated our discussion of this topic in the conclusion to read (lines 371-381):

We have produced global downscaled estimates of HDI for 2012-2021 by combining existing provincial measures with fine-resolution satellite estimates. This approach estimates municipal HDI in each year by adding satellite-based estimates of municipal HDI deviations from the provincial mean, which are time-constant, onto time-varying provincial values calculated from surveys (7). These municipal values are appropriate for applications comparing levels of HDI across locations within a specific moment in time. They cannot be used to evaluate changes in the distribution of HDI within-provinces over time. Current data does not allow us to distinguish performance between our validated approach and an approach that estimates local trends in HDI (Figure S8 and Supplementary Information S3). Developing additional approaches for tracking and validating subnational trends in HDI is an important area for future research.

2.3. Downscaling of the composite HDI index versus individual components

Another issue of concern is the (main) manuscript's methodological focus on downscaling the HDI as a composite index, rather than separately downscaling its individual components and subsequently aggregating them into a single estimate. As evidenced in Table S4 of the

manuscript, the components of the HDI demonstrate low correlations and distinct distributions within countries. For example, the Pearson correlation between expected years of schooling and life expectancy is only 0.1. This indicates that treating the HDI as a unified index may obscure critical spatial variations in its underlying dimensions, potentially reducing both the precision and interpretability of the estimates. While the authors provide the performance of their model for each HDI component in the supplementary materials (Table S3), we argue that these results should be elevated to the main manuscript to emphasize their significance and address these methodological concerns.

Also, in our view, while the HDI index is meaningful in itself, it is even more important to understand the different forms of development across various dimensions. For instance, even with the same HDI score, one region may exhibit a high education index but low life expectancy, while another may show high income but low education scores. These varying patterns of development require different policy responses. Therefore, we believe that producing granular-scale estimates for each individual component of HDI would be more valuable, as it would allow for targeted interventions.

[1] Burke M. et al. (2021). Using satellite imagery to understand and promote sustainable development. Science, 371, eabe8628.

To address the reviewer's concern, we have followed the reviewer's suggestion and moved the discussion of Table S3 into the main text on lines 217-238.

As suggested by the reviewer, we also now create and release data for Mean Years Schooling, Expected Years Schooling, Life Expectancy and GNIpc, using an approach identical to that for HDI. This is now described on lines 244-245:

We also similarly produce and make available estimates of the individual components of HDI.

We also shared the reviewer's intuition that modeling and predicting each HDI component separately and then calculating HDI could potentially improve model performance. However, when we try this, we find that it produces nearly identical results, which is consistent with some prior theoretical results on how MOSAIKS works (12). We now report this result on lines 230-232:

Predictions of HDI made from combinations of its individually predicted components perform nearly identically to the direct predictions of HDI used throughout this analysis.

3.1. Baseline comparisons

The paper assumes that the model predictions explain a fairly significant portion of HDI variations. For example, it reports that model predictions explain 52% of within-country provincial variation in HDI. However, the authors do not discuss whether explaining 52% (or other numbers) of the variation is sufficient to consider the model reliable for policy applications. This omission raises questions about the adequacy of the model's performance for informing HDI-related policies. One key challenge in evaluating this performance is the lack of a baseline comparison with other models capable of estimating HDI or related indices. While there may not be a prior model specifically designed for HDI estimation using satellite imagery and machine learning (SIML), there are existing SIML approaches that could potentially be adapted for HDI prediction. For example, Relative Wealth Index (RWI) in Mexico constructed from Meta [1] can be used for the granular-level baseline comparisons in Table S1. Establishing such a baseline would provide critical context for interpreting the reported performance and determining its practical significance.

[1] Chi, G., Fang, H., Chatterjee, S., & Blumenstock, J. E. (2022). Microestimates of wealth for all low-and middle-income countries. Proceedings of the National Academy of Sciences, 119(3), e2113658119.

Thank you for this thoughtful comment. In our original manuscript, the example policy application in Mexico was designed to test and illustrate how the data we generate could provide actionable information to a policy-maker. We have now expanded this demonstration to Indonesia, building on the micro-data discussed above. In addition, to further clarify key issues raised by the reviewer, we have now added a more general discussion of the utility of our fine-resolution HDI estimates alongside our illustrative application of HDI-based geographic targeting for a policy on lines 297-308:

This application illustrates how fine-resolution HDI estimates can improve targeting of policies that would otherwise use coarser provincial measures, even though these fine-resolution estimates are imperfect. Using provincial measures implicitly assumes no within-provincial variation in HDI, so any positive ability to predict variation within provinces improves understanding of the spatial distribution of HDI. Users of these fine-resolution estimates should consider their specific policy context to determine whether these fine-resolution estimates, which explain 29% of the within-province HDI variation in Mexico, 20% in Brazil, and 53% in Indonesia, provide sufficient additional information to be useful in their setting. These estimates may be particularly valuable in data poor settings, where traditional data sources such as surveys are less available and reliable (15); only about half of the

world's poorest countries have conducted a census in the last decade (16, 26). These estimates may also be useful in cross-country settings where a consistent metric of welfare is desired.

We also agree that understanding the performance of our approach relative to other potential approaches for predicting global HDI at fine resolution is valuable, while also acknowledging that a first global fine-resolution estimate is useful and can serve as a platform for future improvements. We cannot directly apply the approach in Chi et al., (2022) to predict HDI using our data because their approach uses regularly structured data of similar shapes and sizes, and because fine-resolution measurements of HDI do not exist globally for training their approach. Additionally, their approach uses proprietary data which we do not have access to (we have corresponded extensively with the authors in an effort to obtain access in order to respond to this comment). We can, however, compare the performance of our approach to Chi et al. predicting their Relative Wealth Index (RWI), which is identical to what we call "within-country IWI". In general, our performance metrics are slightly higher than Chi et al., although direct comparison is complicated by differences in methods and data. We now add this to the discussion comparing to other prior approaches on lines 318 - 330:

Our strategy is motivated by the limited resolution of available training data for HDI, but our results do not exhibit obvious compromises in performance relative to alternatives that exploit high-resolution labels. A related benchmark in this literature achieves $\rho^2=0.63-0.67$ predicting a wealth index when training and evaluating at the DHS cluster resolution (16, 51). Though we train at the provincial level, not the DHS cluster level, our performance predicting DHS cluster IWI is competitive with this previous analysis $\rho^2=0.7$; Table S1, col 1). Another benchmark in this literature achieves an R^2 of 0.54 predicting RWI, which is conceptually identical to what we call within-country IWI, at the same DHS cluster level (17) (metric taken from a replication of their Figure 3B). Again, despite training at the provincial level rather than the DHS cluster level we achieve a within-country R^2 of 0.56 predicting IWI at the cluster level. Direct comparison to both of these benchmarks, however, is complicated by differences in training and evaluation methodologies including our focus on explaining within-province variation and our corresponding mean-anchoring approach.

We also tried to additionally compare our performance predicting within-province, rather than within-country IWI, which is the preferred metric in our paper because we are interested in explaining fine-resolution variation in HDI within provinces that has

yet to be previously estimated at global scale. Unfortunately, the replication data provided by Chi and co-authors do not allow us to calculate a within-province metric, and correspondence with their team indicates that they no longer have access to the data that would be necessary to do this.

3.2. Ablation study: Excessive Contribution of Nightlight Data to the Full Model's Performance
As described in the ridge regression, MOSAIKS and nightlight are the two primary features used for estimating HDI in a given region. Table S1-S3 showcase the model performance with ablations, where each component is excluded. However, upon reviewing the data in Table S1, it appears that the NL standalone model shows comparable performance to the full model in terms of ρ^2 and R^2 . In some cases the NL model even overperforms the full model. For example, when predicted at municipality level in Mexico, within-country performance of MOSAIKS+NL presents 0.5 ρ^2 (0.45 R^2) while that of NL is 0.58 ρ^2 (0.5 R^2). In light of these results, the authors are encouraged to provide evidence supporting the meaningful contribution of MOSAIKS in the full model.

This is an interesting question and we thank the reviewer for raising it. It is clear throughout the multiple data sets and tests explored in our analysis that both MOSAIKS features and NL features individually contain useful signals and, in principle, the analysis could have been completed using either one and our qualitative findings would have nonetheless been obtained. However, through the process of our work, we have discovered that global performance for HDI using both data sets combined achieves the best results, and thus have proceeded with a focus on this combined model. Nonetheless, as the reviewer points out, the strength of the different signals from these two input data sets are expressed differently for different outcomes in different subsamples. Indeed, the inclusion of both data sets is valuable because it is unknown ex ante what information will be provided by each, and we find that each provides value in different contexts. The single result that the reviewer has isolated is an interesting and puzzling result that we are continuing to investigate in ongoing work; however, we do not yet have a complete explanation that we are satisfied with, and thus we plan to continue future work in this vein. We hope to present a more complete analysis that more fully evaluates how data from many satellite sensors (including others not used in this analysis) can be effectively and efficiently combined in SIML applications. However, this more complete treatment is well beyond the scope of this analysis, and it is not essential to the present work. Thus, throughout the manuscript we have simplified our presentation by focusing on the joint MOSAIKS + NL model that is our primary interest. However, to address the reviewer's concern, we have now added additional discussion of this

to our discussion of the “Value from combining daytime and nighttime imagery” in Supplementary Information Discussion S2. This section now reads:

The MOSAIKS-based approach can use image features from multiple sensors simultaneously when training models, a property that is used throughout this analysis to predict HDI from both daytime and nighttime imagery. Analyzing the performance of MOSAIKS models based on the type of satellite imagery used, we find that daytime and nighttime imagery together explain 7% more variation in provincial HDI deviations from the country mean than does nighttime imagery alone, improving model fit by 16% from $R^2 = 0.45$ to 0.52 (Table S3). This improved performance from using daytime and nighttime imagery together is strongest in regions of low human development (HDI < 0.6) (Figure S7A), consistent with a previous finding that models using daytime imagery outperform models using nighttime imagery when predicting assets of the poorest populations in five African countries (20). Analyzing model performance for each component of HDI, we see that the improved performance predicting HDI using daytime and nighttime imagery stems from improved or comparable performance predicting each component of HDI (largest change in R^2 is 0.12 for expected years of schooling, smallest is no change in R^2 for life expectancy, Table S3).

In exploratory analysis of subsamples, we have observed that in several middle income countries with HDI between 0.7 and 0.8 (including those where we have municipal variation in HDI), within-country R-squared is sometimes higher when models are trained on nighttime imagery alone, relative to models that combine MOSAIKS and nighttime imagery. However, for other parts of the HDI distribution and on average, as described above, MOSAIKS features appear to add performance to the full model. We currently cannot explain why a nightlights-only model performs best for these particular countries when predicting HDI, and we do not know if this phenomena extends to non-HDI outcomes. In future work, we hope to present a more complete analysis that more fully evaluates how data from many satellite sensors, including others not used in this analysis, can be effectively and efficiently combined in SIML applications.

In addition to contributing to the full model performance on average, we also note that the MOSAIKS daytime imagery features on their own have substantial predictive performance, achieving an R^2 of 0.42 predicting within-country provincial HDI variation and an R^2 of 0.44 predicting within-province municipal variation in

Indonesia, the country for which we have the HDI data most closely aligned with the date of our satellite imagery.

4.1. Same weights β for linearization at tile, province, municipality levels.

The model assumes that the weights (β) representing the relationship between satellite imagery features and HDI are consistent across different spatial scales, such as tiles, provinces, and municipalities. However, this assumption—that the same β applies regardless of scale—does not appear to have been explicitly validated in this study. Given the differences in spatial aggregation and scale, it is critical to test whether this assumption holds, as it directly impacts the robustness and reliability of the model's estimates.

We thank the reviewer for suggesting this analysis, which is helpful for both validating the approach and understanding why it works. We have implemented the analysis suggested by the reviewer. However, we note that this test, as suggested by the reviewer and now implemented in our manuscript, is *sufficient* to demonstrate that model weights are transportable across scales of aggregation, but it is not *necessary* to pass this test in order for our main results to hold. Requiring that similar weights are independently recovered across multiple scales is not strictly necessary for one set of weights to be valid when applied to multiple aggregated scales of aggregation. Since our main analysis uses a large number of features relative to the number of training observations, weight selection during the ridge regression step could produce multiple practically-equivalent sets of weights when applied to different samples of data or different scales of aggregation. This is now also explained in the text when we present this test.

We now briefly mention the test in the main text on lines 82-83:

See Supplementary Information S5 for an empirical validation of the scale-invariance of model weights using this approach.

We discuss the test and its results in a supplementary discussion (lines 1202-1257):

S5 Supplementary discussion assessing consistency of model weights across spatial scales

A key feature of our approach is that it can be trained on and make predictions for units of arbitrary shape and size. We employ this approach to train on global provincial HDI data and make predictions of HDI for global municipalities and a 0.1° X 0.1° grid. For this approach to be effective, model

weights estimated at the provincial level, must be able to make skillful predictions at the municipal and grid levels.

One way to understand how this approach works is to see that, in a linear model, the relationship between aggregated outcomes and aggregated features should be similar to the relationship between disaggregated outcomes and features. This is illustrated in Equations 2-3 of the main text. Rolf et al (12) provide additional mathematical explanation for relating predictions made using MOSAIKS features at image and sub-image scales. A primary goal of this manuscript is to propose that this approach can be used to address the challenge of limited training data in remote sensing applications by allowing for training on irregularly structured and sized observations – which is not discussed in ref.(12) – and to empirically test whether this works in practice. The primary evidence supporting this are the downscaling tests reported in Figure 2 and Table S1.

Here, we additionally explore the question of *why* the approach works by empirically testing whether model weights estimated at aggregated and disaggregated scales are similar. To do so, we compare model weights between models of IWI trained at the provincial (N = 862) and DHS cluster (N = 51,996) scales. We use IWI for this experiment because there are a large number of aggregated and unaggregated observations that span the same spatial extent.

A challenge in designing this experiment is that MOSAIKS features are correlated with each other, and there are a large number of features relative to the number of training observations. This means that the same information could load onto different features even when training and retraining at the same scale if we do not introduce additional constraints to the feature set. Put another way, different sets of model weights could give the same predictions and represent the same relationship between the imagery and outcome of interest. This is not an issue in our main application, since the set of weights obtained at an aggregated scale will remain valid if applied to a disaggregated scale, and vice versa. However, there is no guarantee that the same weights will be obtained if models are independently fit at both scales, since there are multiple valid ways to represent the data using the model features. Thus, for this experiment, we first transform our features into an orthogonal basis.

We use Principal Components Analysis (PCA) to project the MOSAIKS features into a feature space with independent (i.e., uncorrelated) features that contain the same information as the original features, following ref. (17). PCA can also be used to reduce the dimensionality of the feature space, which can aid interpretation in this setting by focusing on features that explain most of the variation in the imagery, and thus likely in the outcome of interest. We find that 100 PCA features explain > 99.9% of the variation in the original 4,000 MOSAIKS features in this context, and that a model trained using these 100 PCA features provides essentially identical predictions to a model trained using the 4000 original features (Figure S10A). This PCA model is thus practically identical to our MOSAIKS-based model but with independent features that represent an orthogonal basis. We use these orthogonalized and rotated MOSAIKS features to analyze whether model weights are consistent when training at different levels of aggregation.

Using these orthogonal MOSAIKS features, we find that model weights estimated using the (aggregated) provincial data are very similar to model weights that are independently estimated using (disaggregated) DHS cluster data ($R^2=0.75$, Figure S10A). Note that when calculating R^2 in this setting we weight by the fraction of variance in the MOSAIKS features each component explains, so that greater weight is placed on features that explain more variation in the MOSAIKS features, and likewise, in IWI. The high correspondence between weights estimated at aggregated and disaggregated scales indicates that the same satellite information is being used in the same way to predict IWI at both scales. This helps to explain how our approach is able to achieve skill in the downscaling applications illustrated in Figure 2 and Table S1.”

4.2. Methodological novelty

Another methodological issue is the heavy reliance of this study's model on the MOSAIK framework, which was previously published in Nature Communications [1]. While it is noteworthy that this paper applies the approach to estimating different datasets (i.e., HDI) and provides the first global estimates at the second administrative level, it is difficult to identify significant methodological advancements compared to the earlier work. The primary novelty lies in the application context rather than in the development of the modeling approach itself, raising questions about the extent of its contribution beyond the adaptation to a new target variable.

If the methodological advancements are not substantial, it raises the question of whether the work would be better suited for publication as a Scientific Data paper rather than as a full research article. In such a case, the focus should be on rigorously validating the dataset to ensure its credibility and utility for future research and applications, rather than emphasizing methodological innovation.

[1] Rolf, E. et al. (2021). *A generalizable and accessible approach to machine learning with global satellite imagery. Nature Communications, 12(1), 1-11.*

We respectfully but firmly disagree with the reviewer on this point. We believe that the manuscript makes substantial methodological advancements, and that the present analysis is out of scope for the journal *Scientific Data*.

The aim and scope of *Scientific Data* is “publishing descriptions of research datasets and articles on research data sharing”. This is a different objective than the contributions of the present analysis. For example, in one of our research lab’s prior publications in *Scientific Data* (Krasovich, 2022), we assemble, validate and document a data set of pre-existing water quality measurements from the Mississippi River Basin. This was in alignment with *Scientific Data*’s statement that its Data Descriptor Papers “describe open research datasets in a manner that promotes reuse, without reporting whether datasets support hypotheses or conclusions.” *Scientific Data* expressly prohibits the presentation of any material that goes beyond cataloging new data sets, stating: “Please note that *Scientific Data* does not publish traditional research articles using data to validate regular scientific hypotheses.” Thus, papers that are developing and exploring methods used to make new measurements are not generally published in *Scientific Data*.

Papers, such as our present manuscript, that develop new techniques for measurement, implement large scale measurements (resulting in new data sets), and explore the implications of those new measurements are regularly published in general interest journals such as *Nature Communications*. For example, Hansen et al (“High-Resolution Global Maps of 21st-Century Forest Cover Change”, *Science* 2013); Kruitwagen (“A global inventory of photovoltaic solar energy generating units”, *Nature*, 2021); Chi et al (“Microestimates of wealth for all low- and middle-income countries”, *PNAS*, 2022); and Neugarten et al (“Mapping the planet’s critical areas for biodiversity and nature’s contributions to people”, *Nature Communications*, 2024). Similar to our analysis, these papers develop novel approaches to apply existing methods to a new context, construct valuable new global maps using these techniques, and describe new results that are obtained by inspecting and analyzing these new large-scale measurements.

Our manuscript builds upon the MOSAIKS approach developed in Rolf et al., (2021) and the use of NL to measure economic activity developed in Henderson et al., (2012) to make several substantial methodological contributions. Primarily, it develops an approach that allows for training and predicting on irregularly shaped and sized polygons, which is generally not possible using existing SIML approaches (see lines 45-64 for details). This addresses a key limitation of the existing SIML literature -- the dearth of available training data -- by allowing models to be trained on coarse but plentiful administrative data. Using MOSAIKS in this way was not discussed in Rolf et al., (2021) which applied the approach to only square images of a constant size, following the existing literature.

In addition to proposing and developing this MOSAIKS/NL-based downscaling approach, the manuscript empirically demonstrates that the approach works well in practice (Figure 2), and provides evidence for why it works (Supplementary Information S5). This empirical demonstration of the developed approach is key for broad use of the approach and for trust in the created estimates of HDI.

Developing this approach required a collection of specific methodological advances. For example, we found that to predict fine-scale local variation in the outcome of interest, SIML models should be trained on only fine-scale local variations, rather than the full variation across all the training data (lines 122-133). This combines a fixed-effect regression model specification, often used in econometrics, with a more traditional machine learning approach. The commonly used practice of training on the full variation in the outcome of interest -- both within and across countries -- we show, has substantially lower performance in our setting across all validation tests (Table S3). This led to the development of our "within-country" models, which we use to predict HDI. Another substantial development enabling our approach and its application to HDI was the production of a set of MOSAIKS features across nearly the entire global land surface. In contrast, Rolf et al. (2021), created MOSAIKS features and analysed less than 1% of the global land surface. This manuscript's global dense featurization required roughly \$60,000 of compute, and enabled us to produce HDI estimates across global municipalities, rather than for a small subset of the Earth's surface. These features can be used, in combination with the developed approach, to downscale other variables of interest at large scale.

Reviewer #2 (Remarks to the Author):

We are grateful for your time and your support of early career researchers. Your collective comments have had a tremendous impact on the quality of this manuscript.

Reviewer #3 (Remarks to the Author):

The paper "Global high-resolution estimates of the United Nations Human Development Index using satellite imagery and machine learning" contains the first granular estimates of the UN's Human Development Index. The paper develops a downscaling technique that uses provincial estimates to generate estimates at a more granular level. The paper is very well-written, to the point, and contains numerous relevant and convincing robustness checks. The authors anticipated and addressed many of my questions related to the interpretability of the model, its performance over time, its performance by HDI dimension, and whether the relationship between the features and the outcome is geographically stable. I have three comments, which I think could help improve the relevance and applicability of the model.

We thank the reviewer for their thoughtful and helpful comments. Please find our response to each below.

1. Use of multiple imputation techniques.

The authors point out that the predictions have a much lower variance than the true distribution, which is also evident from several of the panels of Figure 2. If I am not mistaken, this means that the model systematically underpredicts the extent to which non-poor provinces have poor municipalities, as well as the extent to which poor provinces have non-poor municipalities. This seems to be an important issue given that perhaps the main purpose of the paper (as illustrated with the targeting example in Mexico) is to uncover such situations. Multiple imputation and small area estimation techniques have existed for decades to deal with the issue that the variance of the predicted values is too small. Could the authors use such techniques to recover more plausible predictions? I realize that small area estimation applications predict at the same level as which ground-truth data are available, so the authors are in a different situation here. Yet surely there are some assumptions about the error term one could make to recover more realistic predictions, perhaps utilizing the Mexican data where the true distribution of municipal estimates is available. If not attractive, and no attractive alternative exists, please explain why.

We appreciate the reviewer's question and insights with respect to this. Unfortunately, we find that there is no clear way to expand the variance in predictions without reducing the accuracy of our predictions.

Predicted values having lower variance than the true values is a common feature of predictive models and stems from the model not explaining all of the variance in the true values. For example, consider the linear model: $HDI = \beta X + e$. Under this model, $Var(HDI) = Var(X + e)$, but $Var(HDI) > Var(\beta X)$ so long as $Var(e)$ is non-zero. Thus, any

linear model for which the predictions (βX) do not explain all the variance in the outcome (i.e., $R^2 < 1$) will have reduced variance of the predictions relative to the outcome. The more of the variance in the outcome the model can explain, the greater $\text{Var}(\beta X)$ is relative to $\text{Var}(e)$, and the less variance of the predictions will be deflated. Our model, which performs competitively to similar approaches in the literature despite training and evaluating on different scales (lines 318-323), explains roughly half of the within-country and within-province variation in HDI, so it makes sense that its predictions, like those of related models in the literature, have lower variance than the true values. The lower variance of the predictions means that, on average, the model will under-predict how poor relatively poor provinces within countries are and also under-predict how rich relatively rich provinces within countries are. And similarly for municipalities within provinces. While imperfect, these satellite-based predictions are still substantial improvements over the provincial values, which implicitly assume zero municipal variation within provinces.

It is not clear to us how existing multiple imputation and small area estimation techniques could be used to improve upon our current approach. Though multiple imputation can take many forms, it has two general elements that could potentially ameliorate issues related to deflated variance: re-scaling and adding variance. Following ref. (25), we could view the true municipal HDI values as missing, and impute them using the satellite-based HDI predictions calibrated to ground truth municipal HDI values from Indonesia, Mexico and Brazil. To evaluate this type of approach, we could train a calibration model for our estimates using data from Indonesia and Brazil ($\text{HDI} = \beta \hat{\text{HDI}} + e$), and then use this calibration model to correct predicted HDI in Mexico ($\text{HDI}_{\text{calibrated}} = \hat{\beta} \hat{\text{HDI}} + \hat{e}$). This linearly rescales the original HDI estimates to match the ground truth calibration values and adds an error term to better match the variance of predicted values to the true variance. We find that this approach reduces predictive performance in the Mexican data (Within-province R^2 decreases from 0.29 to 0.03, ρ^2 declines from 0.29 to 0.08) and, accordingly, reduces targeting performance (AUC=.86 to AUC=.79). Performance is reduced for two reasons. First, the calibration model learned on the Indonesian and Brazilian data is not a perfect fit for the Mexican data. Similar issues are likely to occur if calibrating on data from Mexico, Indonesia and Brazil and then correcting data from the rest of the world, which is arguably the most feasible application of this approach to the global setting. Second, adding an error term introduces additional noise into the estimates and is thus not preferable when trying to generate a single best estimate of the quantity of interest. While this added noise increases the variance of estimates, it also introduces error into the estimates, making them farther on average from the true values, which reduces targeting performance.

While applying multiple imputation approaches to correct the HDI data is likely to be useful in some settings (25), we see that in this targeting setting it worsens performance. Thus, we maintain our estimates as they are, and allow users to implement multiple imputation using our open source code and freely available data if it would be of use in their particular setting. We note that there are many ways to potentially apply multiple imputation, but we see no reason that general features of rescaling, adding noise, or making multiple predictions of the target variable, are likely to improve our estimates in general.

With regards to small area estimation methods, we now explain how our approach is a type of small area estimation on lines 345-352:

This approach can be viewed as a type of regression-based technique for small area estimation using satellite imagery (53, 54). Though data limitations in this setting – including the lack of fine-resolution global surveys of HDI – preclude the direct implementation of most commonly used small area estimation techniques (e.g., (55)), this approach follows the general small area estimation framework of combining limited measurements of the variable of interest (provincial HDI) with universally available axillary data informing the variable of interest (satellite imagery) to make accurate and precise fine-resolution predictions.

We view improving the model to more fully explain HDI as the best way to address this challenge of reduced variance. To this end, we try adding features from an additional satellite to the model. Unfortunately, this did not improve performance, indicating that there may be a limit to what can be predicted using satellite imagery in this setting. We discuss this on lines 331-344:

Here, we use satellite imagery due to its rich information content, global availability, and near uniformity in quality across countries. To evaluate whether additional satellite data sources beyond visual daytime imagery and nighttime imagery might improve HDI estimates, we follow ref. (22) and add to the baseline model three additional index-based features constructed from Sentinel-2A satellite imagery that integrate knowledge of spectral properties of different ground conditions: the Normalized Difference Vegetation Index, Normalized Difference Water Index, and Normalized Difference Built-Up Index. We find that the model performance is essentially unchanged (Table S5), indicating that these additional features do not provide additional information beyond what is already captured by the MOSAIKS and NL features. While it is important to note that there is a limit to how well

socioeconomic variables can be predicted using satellite imagery generally, and adding these additional features did not improve performance in this case, future work should nonetheless explore whether incorporating additional imagery sources and/or other ancillary data can improve these estimates (52).

2. Discussion of advantages/disadvantages for targeting relative to alternative approaches

The policy relevance of the paper clearly comes from improved within-country targeting of resources, as cross-country targeting of resources tends to rely on country-level data. Yet it is unclear when countries armed with these estimates would use them for targeting resources. Countries with administrative data would probably use those for targeting, while countries with recent census and survey data would probably rely on small-area estimation techniques to generate granular measures of well-being (I realize small area estimation techniques rarely if ever have the HDI on the left-hand side, but they could have any multidimensional measure of well-being that can be recovered from surveys). What are the cases the authors envision the data generated could be applied? Perhaps in countries with no census, no survey data, no admin data? Or do you think this method works better than small-area estimation techniques? If so, what is the evidence to that end? This paper might be of interest for the discussion: <https://doi.org/10.1016/j.jdevec.2024.103352>

Thank you for this question. We now provide a greater discussion of the potential use and value of these estimates for targeting on lines 297-308:

This application illustrates how fine-resolution HDI estimates can improve targeting of policies that would otherwise use coarser provincial measures, even though these fine-resolution estimates are imperfect. Using provincial measures implicitly assumes no within-provincial variation in HDI, so any positive ability to predict variation within provinces improves understanding of the spatial distribution of HDI. Users of these fine-resolution estimates should consider their specific policy context to determine whether these fine-resolution estimates, which explain 29% of the within-province HDI variation in Mexico, 20% in Brazil, and 53% in Indonesia, provide sufficient additional information to be useful in their setting. These estimates may be particularly valuable in data poor settings, where traditional data sources such as surveys are less available and reliable (15); only about half of the world's poorest countries have conducted a census in the last decade (16,

26). These estimates may also be useful in cross-country settings where a consistent metric of welfare is desired.

And we discuss how the approach in this manuscript relates to small area estimation approaches on lines 345-364:

This approach to producing HDI estimates can be viewed as a type of regression-based technique for small area estimation using satellite imagery (53, 54). Though data limitations in this setting – including the lack of fine-resolution global surveys of HDI – preclude the direct implementation of most commonly used small area estimation techniques (e.g., (55)), this approach follows the general small area estimation framework of combining limited measurements of the variable of interest (provincial HDI) with universally available auxiliary data informing the variable of interest (satellite imagery) to make accurate and precise fine-resolution predictions. A key difference between more traditional small area estimation techniques and ours is that we use a global-level auxiliary dataset that does not suffer from the challenge of construct validity across countries, although this comes with a tradeoff of needing to assume some degree of consistency between imagery and conditions on the ground. A recent comparison of poverty maps for Malawi produced using i) a small area estimation approach combining survey data and census data and ii) an alternative approach combining survey data and geospatial indicators including satellite imagery found that the two approaches produced very similar estimates, with a correlation exceeding 0.9 (56). This, paired with the performance of satellite imagery predicting HDI shown here, indicates that satellite imagery can be an effective substitute for census information when producing fine-resolution estimates of human wellbeing in settings where census data is not available. In locations where reliable census data is available, satellite imagery could serve as a complementary source of information.

3. Interpretation of the model's error

The authors conduct numerous different and relevant checks to test the performance of their model. The overarching challenge is that an ideal assessment of the model would require true granular HDI estimates which, except for in one circumstance, do not exist. The one genuine test of the error they can make suggests that the performance is moderate at best (Figure 2D), with an R^2 of 0.29, while the other one that comes close (2G) is likewise not too great. I think the authors ought to place greater emphasis on the point that the model fundamentally cannot

be evaluated in a first-best manner and that second-best approximations suggest that one-third of the relevant variance can be predicted by the model.

We appreciate this thoughtful comment and agree that this is an important point to make clearly. In the text we write (lines 135-136):

We cannot directly evaluate the performance of municipality-level or grid-level HDI predictions worldwide because such highly-resolved estimates have not been previously constructed.

And that (lines 149-153):

No test can directly validate the performance for downscaling HDI globally, since the data necessary for such a test do not exist; however, all three of these large-scale tests taken together document the effectiveness of our downscaling strategy in general -- using global socioeconomic data similar to HDI -- and for HDI in particular -- using municipal HDI data for three countries.

We also now add that (lines 211-212):

Comparisons to municipal data on HDI, IWI, and NL indicate that models trained on provincial data can explain 20-57% of within-province municipal variation.

We have also added two additional direct evaluations of our municipal HDI predictions using data from Indonesia and Brazil. We discuss model performance comparing our predictions to these estimates on lines 159-177 and visualize performance in Figure 2.

Downscaling HDI in Mexico, Brazil and Indonesia

As a direct evaluation of HDI downscaling performance, we compare municipal HDI predictions from the satellite-based model trained on provincial HDI deviations from the country mean to municipal HDI derived from census-based calculations in Mexico (9), Brazil (35), and Indonesia (34) (Methods 2.3). In Mexico, downscaled HDI predictions explain 45% of the municipal HDI variation overall (Figure 2C) and 29% of the within-province variation (Figure 2D). In Brazil, HDI predictions explain 48% of municipal HDI variation overall, and 20% of the within-province variation. And in Indonesia, HDI predictions explain 61% of the municipal HDI variation overall and 53% of the within-province variation. It is encouraging that HDI predictions align

better with census-based measures for Indonesia than for Mexico or Brazil because measures for Indonesia are from 2019, which aligns with our satellite-based predictions of within-province HDI variation, while measures for Mexico and Brazil are from 2010. These older HDI measures could differ from our satellite-based predictions in part due to changes in HDI since the measurements were taken. Additionally, some portion of the misalignment between satellite-based predictions and survey estimates likely arises as a result of errors in the survey data itself, a widely recognized issue (15) that we are unable to assess. Together, these results indicate that our method for SIML-based downscaling improves our understanding of the spatial distribution of HDI in these three example countries; although, it is not a complete substitute for survey-based estimates when such data are available.

The updated Figure 2 is reproduced below:

Figure 2: MOSAIKS models perform well predicting socioeconomic indicators, including at downscaled resolution. (A) Observed and predicted HDI at the province level. Note that the within-country variation is smaller than the across-country variation, as illustrated by France, in yellow, and Ethiopia, in pink. (B) The same as (A), evaluating within-country variation. Provincial deviations from the country mean for France and Ethiopia are now centered at 0, and the model is evaluated on how well it can differentiate provinces that are relatively well and worse-off within countries. (C-D) Observed and predicted municipal HDI data in Mexico, Brazil and Indonesia. (E-G) Observed and predicted IWI at the DHS cluster level. (H-J) Observed and predicted nighttime lights at the municipality level. The vertical streaking in (H-J) are caused by countries that have very spatially dense municipalities (Supplementary Information S6.3). Predictions are anchored to known country or provincial means (Methods 4.4). All predictions are for the year 2019.

And the updated associated Table S1 is reproduced as well, with some modifications to accommodate the additional analysis:

Predicted at province level (n=1,381)						
	Full variation performance		Within-country performance			
	ρ^2	R^2	ρ^2	R^2		
	(1)	(2)	(3)	(4)		
HDI trained at:						
Within-country (n=1,363)	0.96	0.96	0.52	0.52		
Province level (n=1,381)	0.83	0.83	0.44	0.24		
Country level (n=145)	0.74	0.74	0.28	< 0		
Predicted at municipality level in Indonesia (n=505)						
			Within-country performance		Within-province performance	
			ρ^2	R^2	ρ^2	R^2
			(3)	(4)	(5)	(6)
HDI trained at:						
Within-country (n=1,363)			0.62	0.61	0.53	0.53
Province level (n=1,381)			0.5	0.37	0.51	0.37
Country level (n=145)			0.35	< 0	0.36	< 0
Predicted at municipality level in Brazil (n=5,584)						
			Within-country performance		Within-province performance	
			ρ^2	R^2	ρ^2	R^2
			(3)	(4)	(5)	(6)
HDI trained at:						
Within-country (n=1,363)			0.48	0.48	0.33	0.20
Province level (n=1,381)			0.46	0.36	0.31	< 0
Country level (n=145)			0.29	< 0	0.18	< 0
Predicted at municipality level in Mexico (n=2,457)						
			Within-country performance		Within-province performance	
			ρ^2	R^2	ρ^2	R^2
			(3)	(4)	(5)	(6)
HDI trained at:						
Within-country (n=1,363)			0.5	0.45	0.31	0.29
Province level (n=1,381)			0.31	0.31	0.26	0.23
Country level (n=145)			0.16	< 0	0.12	< 0
Predicted at DHS cluster level (n=51,996)						
	Full variation performance		Within-country performance		Within-province performance	
	ρ^2	R^2	ρ^2	R^2	ρ^2	R^2
	(1)	(2)	(3)	(4)	(5)	(6)
IWI trained at:						
Within-country (n=862)	0.71	0.67	0.56	0.56	0.36	0.36
Province level (n=862)	0.44	0.28	0.21	0.08	0.24	0.1
Country level (n=85)	0.31	< 0	0.13	< 0	0.13	< 0
Predicted at municipality level (n=62,536)						
	Full variation performance		Within-country performance		Within-province performance	
	ρ^2	R^2	ρ^2	R^2	ρ^2	R^2
	(1)	(2)	(3)	(4)	(5)	(6)
NL trained only on MOSAIKS at:						
Within-country (n=2,852)	0.73	0.7	0.64	0.63	0.61	0.57
Province level (n=2,852)	0.65	0.61	0.57	0.45	0.53	0.35
Country level (n=170)	0.44	0.3	0.36	0.08	0.31	< 0

Table S1: Performance for models trained to predict HDI, IWI, and population-weighted nightlight luminosity (NL). Models for HDI and IWI use a combination of MOSAIKS and population-weighted NL features. We show performance evaluated at the province level for HDI and evaluate downscaled performance for HDI, IWI, and NL. Performance scatters from the within-country models are shown in Figure 2. All predictions are made for the year 2019.

The new data are described in the methods on lines 493-518:

Municipality-level HDI

We compare our municipal HDI estimates with census-derived municipal estimates for HDI, where these data exist.

Indonesia Time series estimates of HDI at the municipality level are made publicly available by Badan Pusat Statistik (BPS), the statistics agency of Indonesia (63). We use estimates from 2019, which makes these, to our knowledge, the only available municipal HDI data that come from the same year as our satellite imagery.

Brazil The Human Development Report Office office of the United Nations Development Programme has derived HDI data at the municipality level for Brazil, using census data (35). The most recent year these data are available is 2010, which we use in this analysis. The census-derived data have a different mean than the 2019 HDI data that we use elsewhere in this analysis; though, this has no influence on the within-country or within-province evaluation metrics (Figure 2 C,D) because predictions and observations are demeaned at the country and province level, respectively, before the metrics are calculated.

Mexico Census derived estimates of HDI are also available in Mexico for the year 2010, constructed by ref. (9). As with Brazil, these data have a different mean than the 2019 HDI data; though, again, that has no influence on the within-country or within-province evaluation metrics.

When certain components of HDI are not directly available at the municipal level, producers of these datasets use close proxies. For example the BPS calculates municipal HDI data in Indonesia using data on real expenditure rather than GNIpc (34). And when calculating municipal HDI in Mexico, ref. (9) use the child survival rate rather than life expectancy at birth for the health index and an asset index rather than GNIpc for the standard of living index. Likewise, provincial HDI estimates are also constructed using proxies when direct measurements of HDI components are not available (7). Discrepancies in how HDI is calculated at the municipal and provincial levels may contribute to differences between our satellite-based HDI estimates and these survey-based HDI estimates.

These direct validations of the downscaling approach using observations of subnational HDI from Mexico, and now Indonesia and Brazil as well, supplement the additional downscaling tests applying the method to nighttime lights and the IWI. In all tests, the approach is able to predict substantial variation in municipal HDI, or related wealth indices, within the provincial units on which the model was trained. This, in combination with the rest of the paper's analyses, provides substantial evidence that the downscaling approach provides new and useful information on HDI at fine resolution globally.

We do acknowledge, however, that these estimates are far from perfect and that "there is a limit to how well socioeconomic variables can be predicted using satellite imagery generally" (lines 340-341). We now provide additional discussion of the potential utility of these imperfect estimates on lines 297-308:

This application illustrates how fine-resolution HDI estimates can improve targeting of policies that would otherwise use coarser provincial measures, even though these fine-resolution estimates are imperfect. Using provincial measures implicitly assumes no within-provincial variation in HDI, so any positive ability to predict variation within provinces improves understanding of the spatial distribution of HDI. Users of these fine-resolution estimates should consider their specific policy context to determine whether these fine-resolution estimates, which explain 29% of the within-province HDI variation in Mexico, 20% in Brazil, and 53% in Indonesia, provide sufficient additional information to be useful in their setting. These estimates may be particularly valuable in data poor settings, where traditional data sources such as surveys are less available and reliable (15); only about half of the world's poorest countries have conducted a census in the last decade (16, 26). These estimates may also be useful in cross-country settings where a consistent metric of welfare is desired.

We also note that our model estimates, while imperfect, perform competitively with prior approaches in the literature despite training on coarse provincial data (see the discussion on lines 318-330 for details).

Reviewer #4 (Remarks to the Author):

Summary

The manuscript by Sherman et al. presents a novel method to downscale the United Nations Human Development Index (HDI) using a combination of high-resolution daytime and nighttime satellite imagery and machine learning. The study addresses the critical issue of aggregation bias in traditional HDI measurements by demonstrating that a significant fraction of the global population is misclassified when relying solely on country- or province-level data. Validation using survey-based estimates in Mexico and comparisons with alternative socioeconomic indicators (e.g., the International Wealth Index and nighttime lights) support the robustness of the proposed approach. Overall, the work offers a promising avenue to support more spatially precise policymaking and contributes substantially to the literature on satellite-based socioeconomic measurements.

We thank the reviewer for their thoughtful and constructive comments. Please find our response to each below.

Major Comments

1. Introduction:

The paper assumes how HDI is constructed at the national and municipal levels, but it would be beneficial to explicitly mention the range of HDI (0 to 1), its variation across countries, and the weights assigned to each component (education, health, and income).

We add this information on lines 468-479:

The United Nations Human Development Index is a composite measure used to assess a country's average achievements in three key dimensions of human development: health, education, and standard of living (4). Health is measured by life expectancy at birth; education is evaluated using a combination of mean years of schooling for adults and expected years of schooling for children; and standard of living is assessed through gross national income (GNI) per capita, adjusted for purchasing power parity (PPP). The HDI applies a logarithmic transformation to GNI to account for the diminishing benefits of increased income. Each of these components is normalized on a scale from 0 to 1, and the HDI is calculated as the geometric mean of the three dimension indices, giving equal weight to each dimension. This method allows the HDI to reflect not just economic wealth, but also

broader aspects of well-being. In 2019, the average HDI across countries was 0.72, with a standard deviation of 0.15, a minimum of 0.36 in Somalia and a maximum of 0.96 in Switzerland. (Figure 3A).

In Mexico, municipal-level HDI estimation relies on the population census for education and health, but the income variable is estimated using small area estimation techniques (Elbers, Lanjouw, and Lanjouw 2003). These estimations are inherently uncertain. It would strengthen the paper to contrast the proposed method with these techniques, highlighting not only the limitations of census data for small areas but also the challenges in obtaining statistically valid income estimates at the municipal level.

We now discuss the uncertainty in the Mexican municipal HDI estimates on lines 510-518:

When certain components of HDI are not directly available at the municipal level, producers of these datasets use close proxies. For example the BPS calculates municipal HDI data in Indonesia using data on real expenditure rather than GNIpc (34). And when calculating municipal HDI in Mexico, ref. (9) use the child survival rate rather than life expectancy at birth for the health index and an asset index rather than GNIpc for the standard of living index. Likewise, provincial HDI estimates are also constructed using proxies when direct measurements of HDI components are not available (7). Discrepancies in how HDI is calculated at the municipal and provincial levels may contribute to differences between our satellite-based HDI estimates and these survey-based HDI estimates.

We also now discuss how our approach fits within the range of small area estimation techniques on lines 345-364:

This approach to producing HDI estimates can be viewed as a type of regression-based technique for small area estimation using satellite imagery (53, 54). Though data limitations in this setting -- including the lack of fine-resolution global surveys of HDI -- preclude the direct implementation of most commonly used small area estimation techniques (e.g., (55)), this approach follows the general small area estimation framework of combining limited measurements of the variable of interest (provincial HDI) with universally available axillary data informing the variable of interest (satellite imagery) to make accurate and precise fine-resolution predictions. A key difference between more traditional small area estimation techniques and

ours is that we use a global-level auxiliary dataset that does not suffer from the challenge of construct validity across countries, although this comes with a tradeoff of needing to assume some degree of consistency between imagery and conditions on the ground. A recent comparison of poverty maps for Malawi produced using i) a small area estimation approach combining survey data and census data and ii) an alternative approach combining survey data and geospatial indicators including satellite imagery found that the two approaches produced very similar estimates, with a correlation exceeding 0.9 (56). This, paired with the performance of satellite imagery predicting HDI shown here, indicates that satellite imagery can be an effective substitute for census information when producing fine-resolution estimates of human wellbeing in settings where census data is not available. In locations where reliable census data is available, satellite imagery could serve as a complementary source of information.

2. Methodological Clarity:

While the authors discuss the MOSAIKS method, a clearer explanation with a more accessible example would enhance understanding. Figure 1 provides a visual representation, but the role of colors and the integration of socioeconomic variables could be explained more explicitly. A simplified schematic or a step-by-step breakdown of Figure 1 would aid readers unfamiliar with MOSAIKS in grasping how the method links coarse provincial data with fine-resolution predictions.

We thank the reviewer for this helpful comment. To improve clarity of the methods we have expanded the schematic of our methods and their description to include a discussion not only of how the features capture information from the color and texture of the imagery, but how this information about color and texture is used to predict HDI. The updated Figure S1 is reproduced below and also now referenced in the caption to Figure 1.

Figure S1: **An illustration of generating MOSAIKS features for two cartoon images, and using the features to predict HDI and other example outcomes.** (A) MOSAIKS random convolutional features capture the information within satellite imagery by measuring how similar each image is to a fixed set of small patches of imagery. Similarity is measured mathematically using a moving-window dot product, or “convolution.” The parts of each image that are similar to each patch are shown, with greater similarity leading to larger feature values. The green patch, for example, is similar to the parts of the imagery containing green trees. Image 1 has a greater feature value for this patch because it has more trees than image 2. Collectively, MOSAIKS features capture information on the color and texture of the imagery, which represent the content of the imagery (e.g., trees, roads, and lakes). (B) Features associate differently with different outcomes: images that are more similar to the green patch tend to have higher forest cover, and images that are more similar to the grey patch tend to have more roads. Regressing HDI onto these features learns how higher or lower feature values associate with higher or lower HDI, and in turn, how to predict HDI using these features. Each dot in each scatter represents an image, and the arrow represents the direction in the feature space of increasing forest, roads, or HDI. The direction of the arrow is learned by the regression. For more details on MOSAIKS features and how they can be used to predict a broad range of outcomes see ref. (12).

The paper could delve deeper into the comparison between the proposed method and small area estimates (SAE). It would be valuable to discuss whether the ML-based downscaling approach offers advantages over SAE, particularly in terms of variance and accuracy.

As mentioned in the response to a previous comment, we now discuss how our approach fits within the range of small area estimation techniques on lines 345-364:

This approach to producing HDI estimates can be viewed as a type of regression-based technique for small area estimation using satellite imagery (53, 54). Though data limitations in this setting – including the lack of fine-resolution global surveys of HDI – preclude the direct implementation of most commonly used small area estimation techniques (e.g., (55)), this approach follows the general small area estimation framework of combining limited measurements of the variable of interest (provincial HDI) with universally available auxiliary data informing the variable of interest (satellite imagery) to make accurate and precise fine-resolution predictions. A key difference between more traditional small area estimation techniques and ours is that we use a global-level auxiliary dataset that does not suffer from the challenge of construct validity across countries, although this comes with a tradeoff of needing to assume some degree of consistency between imagery and conditions on the ground. A recent comparison of poverty maps for Malawi produced using i) a small area estimation approach combining survey data and census data and ii) an alternative approach combining survey data and geospatial indicators including satellite imagery found that the two approaches produced very similar estimates, with a correlation exceeding 0.9 (56). This, paired with the performance of satellite imagery predicting HDI shown here, indicates that satellite imagery can be an effective substitute for census information when producing fine-resolution estimates of human wellbeing in settings where census data is not available. In locations where reliable census data is available, satellite imagery could serve as a complementary source of information.

We also expand our comparison with other related approaches in the literature that seek to estimate measures of human wellbeing at fine resolution on lines 318-330:

Our strategy is motivated by the limited resolution of available training data for HDI, but our results do not exhibit obvious compromises in performance relative to alternatives that exploit high-resolution labels. A related benchmark in this literature achieves $\rho^2=0.63-0.67$ predicting a wealth index

when training and evaluating at the DHS cluster resolution (16, 51). Though we train at the provincial level, not the DHS cluster level, our performance predicting DHS cluster IWI is competitive with this previous analysis ($\rho^2=0.7$; Table S1, col 1). Another benchmark in this literature achieves an R^2 of 0.54 predicting RWI, which is conceptually identical to what we call within-country IWI, at the same DHS cluster level (17) (metric taken from a replication of their Figure 3B). Again, despite training at the provincial level rather than the DHS cluster level we achieve a within-country R^2 of 0.56 predicting IWI at the cluster level. Direct comparison to both of these benchmarks, however, is complicated by differences in training and evaluation methodologies including our focus on explaining within-province variation and our corresponding mean-anchoring approach.

The paper does not expand on the consistency of estimates in well-being over time. Specifically, it would be insightful to investigate instances where the official HDI shows an increase, while alternative measures like SIML suggest a decrease (or viceversa). Are there such cases?

Thank you for this comment. We focus this manuscript on estimating fine-resolution *spatial* variation in HDI. We do so because existing provincial measures do not resolve within-province differences in human wellbeing, and because spatial variation in HDI is generally 60 times larger than temporal variation. As described in Methods Section 4.5, we construct fine-resolution global estimates of HDI from 2012-2021 by adding time-constant satellite-based estimates of municipal HDI deviations from the provincial mean onto time-varying survey-based provincial values over this temporal range. Thus, we do not specifically predict or evaluate changes in the within-province distribution of HDI over time. All temporal changes in our estimates stem from temporal changes in the provincial values that the municipal estimates are anchored to. Supplementary Information Section S3 provides further discussion of and analysis on this topic.

To clarify this for readers we have updated our discussion of this topic in the conclusion to read:

We have produced global downscaled estimates of HDI for 2012-2021 by combining existing provincial measures with fine-resolution satellite estimates. This approach estimates municipal HDI in each year by adding satellite-based estimates of municipal HDI deviations from the provincial mean, which are time-constant, onto time-varying provincial values calculated

from surveys (7). These municipal values are appropriate for applications comparing levels of HDI across locations within a specific moment in time. They cannot be used to evaluate changes in the distribution of HDI within-provinces over time. Current data does not allow us to distinguish performance between our validated approach and an approach that estimates local trends in HDI (Figure S8 and Supplementary Information S3). Developing additional approaches for tracking and validating subnational trends in HDI is an important area for future research.

Reviewer #4 (Remarks on code availability):

The link works and seems to be very informative:

github.com/Global-Policy-Lab/hdi_downscaling_mosaiks

I did not install the files, but it seems that everything is reproducible.

Thank you for reviewing our code and data. We are committed to ensuring that all data are available and that all analyses are replicable.

Replies to Reviewers

Reviewer #1 (Remarks to the Author)

I appreciate the authors' efforts and the potential value of high-resolution HDI-related data. I have carefully evaluated the authors' rebuttal and revised materials in light of my initial review. While I appreciate the authors' efforts, the key concerns remain insufficiently addressed. In my judgment, addressing these issues would require substantial new work and re-framing that goes beyond a reasonable revision and would effectively constitute a new study.

Our Reply

We thank the reviewer for their previous thoughtful comments, as they have significantly improved the quality of the manuscript.

However, we respectfully but firmly disagree with the reviewer's new assessment of the methodological elements of the manuscript. As we detail below, the Reviewer's new assertions that support this assessment do not accurately reflect the content of the manuscript, the analysis, alternative data sources, and the prior literature. Nonetheless, we have, where possible, revised the manuscript to avoid confusion among readers on the issues raised by the Reviewer. Please find the reviewers comments below in plain text, and our responses in bold.

Here are three issues that I am not still convinced by the manuscript and authors' responses.

1. Low model performance

Model performance at the target spatial scales is too low to support the main claims and intended uses. The manuscript validates municipality-level predictions using province-level HDI data from three countries (Indonesia, Brazil, Mexico). The authors focus on "explaining within-province variation" (manuscript, line 330), and this is the "preferred metric in (their) paper because (they) are interested in explaining fine-resolution variation in HDI within provinces that has yet to be previously estimated at global scale" (response, pp. 15-16).

However, the reported within-province predictive performance averages about $R^2 \approx 0.34$ (Brazil 0.20, Mexico 0.29, Indonesia 0.53). This is substantially below what is commonly seen and expected in the satellite-based socioeconomic prediction literature for policy-relevant applications (typically $R^2 \geq 0.5$). Given this level of performance and the scale mismatch in validation, the headline claims—particularly regarding widespread misclassification of HDI quintiles and improved decision-making—are not sufficiently substantiated.

Our reply

We respectfully but firmly disagree with both the accuracy and validity of the Reviewer's statements.

First, the assertion that novel research in this domain is only publishable if $R^2 \geq 0.5$ is both arbitrary and inaccurate. In our years of work in this field, we have never heard of nor encountered such a standard, and the suggestion that such an implicit standard exists is not correct. Indeed, the canonical paper in this field by Jean et al. (*Science*, 2016) reported numerous R^2 for consumption expenditures across five countries that were universally below 0.5, ranging from 0.19-0.46, with the outlying exception of one model trained and evaluated in Tanzania ($R^2=0.56$). Moreover, the work published in this journal by Rolf et al. (*Nature Communications*, 2021) that the current manuscript builds upon presented a top R^2 for income of 0.45. In contrast to the reviewer's claim, there is no implicit or explicit standard in the field, and we are aware of no objective R^2 -based standard for signal detection. Different tasks in this field have different degrees of difficulty and different metrics are used in a range of use-cases, each of which is generally evaluated based on context, including the performance of comparable recent work. It is not appropriate to compare our within-province performance to model performance in the literature evaluated using within-country or full-variation performance because within-province performance is more difficult to estimate.¹ Further, instead of training and predicting at fine resolution, as is typically done, we are training at coarse resolution and predicting at fine resolution, which is a novel and more difficult task that is necessary because fine-resolution HDI measures do not exist globally. In the case for the present manuscript, the downscaling of HDI at these resolutions has never been achieved, and thus the prior R^2 benchmark would be zero.

Second, the Reviewer is presenting the incorrect R^2 values when evaluating claims presented in the paper, whereas the correct values actually exceed the Reviewer's threshold of $R^2 \geq 0.5$. The Reviewer's concern about the headline claims about misclassification of HDI quintiles and improved decision-making are based on *within-country* HDI distributions and have an accuracy that should be evaluated based on the "*within-country* performance" R^2 metric. The Reviewer incorrectly presents the "*within-province* performance" R^2 metric, which is not the most relevant metric for evaluating within-country HDI distributions, particularly when comparing to prior literature that has been evaluated using a within-country performance metric. When the appropriate R^2 metric is referenced, the average $R^2=0.51$ for Brazil, Mexico, and Indonesia and $R^2=0.56$ for DHS clusters (see Table S2, "*Within-country* performance" column). Thus, when evaluated with a metric that is comparable to that used in the literature (e.g., Chi et al., Yeh et al.) we achieve a performance level that exceeds the Reviewer's $R^2 \geq 0.5$ threshold for acceptable performance. Of course, we think this threshold is not appropriate, but if the Reviewer prefers it as stated, then they should have concluded

¹ For example, Yeh et al., 2020 achieve a full-variation $R^2=0.67$, but an R^2 of 0.32 and 0.40 differentiating values within rural and urban contexts -- a related, though not identical, task to explaining local variation within provinces (see their Figure 3d).

that the manuscript's headline claims were substantiated. Comparisons of performance with the literature were explicitly discussed in the original manuscript, which stated in the second paragraph of the Conclusion:

[O]ur results do not exhibit obvious compromises in performance relative to alternatives that exploit high-resolution labels. A related benchmark in this literature achieves $\rho^2=0.6-0.67$ predicting a wealth index when training and evaluating at the DHS cluster resolution (Yeh et al., 2020; Yeh et al., 2021). Though we train at the provincial level, not the DHS cluster level, our performance predicting DHS cluster IWI is competitive with this previous analysis ($\rho^2=0.71$; Table S1, col 1). Another benchmark in this literature achieves an R^2 of 0.54 predicting RWI, which is conceptually identical to what we call within-country IWI, at the same DHS cluster level (Chi et al., 2022) (metric taken from a replication of their Figure 3B). Again, despite training at the provincial level rather than the DHS cluster level we achieve a within-country R^2 of 0.56 predicting IWI at the cluster level.

Nonetheless, the Reviewer's comment highlights to us that the large number of R^2 values we present to evaluate different components of the data distribution can be confusing. Thus, to avoid similar confusion among readers, we have now added text to the figure captions related to misclassification of HDI quintiles and improved decision-making reporting the appropriate R^2 associated with the figure results. The caption to Figure 4 now reads (new text in orange):

“Note: in validation tests, average within-country $R^2=0.51$ and $\rho^2=0.53$ for municipal HDI in Brazil, Mexico, and Indonesia.”

Finally, the Reviewer's suggestion that the model performance presented is of too low quality to be of interest or use to policy-making applications is not correct. This work is of intense interest to the policy-makers that developed and use the HDI metric. For example, these models and results are already in use by the UNDP and were reviewed in detail by the UNDP Statistical Advisory Panel on February 27, 2025. This work was codeveloped and coauthored with researchers at the UNDP (as documented in Supplementary Appendix S1) and they deeply understand its functionality and limitations. Policy-makers who understand the technique and estimates we present have already evaluated and adopted it.

In addition, a simple, available baseline outperforms the authors' approach in at least one country. In my first-round review, I requested a within-province baseline using Meta's Relative Wealth Index (RWI) for Mexico (Chi et al., 2022). The authors responded that it was not possible “to calculate a within-province metric” (response, p. 16). Alternatively, I simply computed the within-province relationship between municipal MHDl and the mean RWI by municipality in Mexico (N = 2,456; excluding Bacalar due to missing MHDl). MHDl data was accessed through

the website of authors. This baseline achieved $R^2 = 0.364$ at the municipality level, exceeding the authors' reported value for Mexico ($R^2 = 0.29$). This indicates that a simple, readily available baseline can match or outperform the proposed approach on the stated task.

Our reply

We apologize that we misunderstood the Reviewer's previous question regarding the Chi et al. data. We interpreted the reviewer's initial comment as a suggestion to directly compare performance for the DHS-level wealth data that both we and Chi et al. use to evaluate our respective approaches. We worked diligently and in good faith to try and address what we understood to be the Reviewer's question and consulted extensively with the Chi et al. authors about how to calculate this metric and make appropriate comparisons, which we added to the manuscript (we are coauthors with the researchers on that team). We had not implemented the test that the Reviewer describes above because it did not seem to be a valid comparison (which we detail below).

Based on the Reviewer's updated example calculation, which clarifies the Reviewer's original request, we respectfully but firmly point out that (1) the Reviewer's proposed test is not a valid comparison to the downscaling procedure presented in the manuscript; (2) the proposed alternative solution is not viable globally with real-world data; and (3) even as stated, the proposed test was not fairly implemented or accurately evaluated in the above comment.

First, the Reviewer's proposed comparison is to regress municipal HDI in Mexico directly on the RWI constructed by Chi et al., thereby training and evaluating a model on the municipal data. This, however, is incomparable with all of our estimates because the Reviewer's procedure skips the difficult-but-essential *downscaling* part of the problem. The accuracy of the manuscript's downscaling analysis is the focus of metrics cited by the Reviewer, and demonstrating that this downscaling works is a key contribution of the manuscript. In contrast, the Reviewer's test bypasses the downscaling problem and assumes that they can observe the municipal-level data, but this is forbidden by the setup of the downscaling problem. The test in the manuscript that the Review is comparing against acknowledges that municipal data is generally not observable (Mexico is a special case) and only uses the municipal data from Mexico to evaluate how well the downscaling model performs -- even when it has not seen the municipal data during training. From the perspective of the problem that the manuscript is solving, the Reviewer's proposed test "cheats" by training the model directly on the unobservable municipal data. Nowhere in the manuscript do we train on this target data, since it is only used for validation. And if we do train at the municipal level, we substantially outperform

models based on the Chi et al data.² The output of the Reviewer's proposed procedure is not a valid baseline because it does not solve the downscaling problem.

Second, the Reviewer states that their approach represents a “readily available baseline”, but this statement is not accurate because the necessary data is not available – their suggested baseline is viable for only 1.7% of the countries in our sample. The Reviewer's proposed baseline procedure trains a model directly on municipal HDI data to predict municipal HDI. Thus their approach can be implemented only in the few countries where municipal HDI data exists. In response to the Reviewer's previous request, we conducted extensive searches for such municipal data, including leveraging the statistical network of the UNDP, and we were able to recover municipal HDI data for only Mexico, Indonesia and Brazil (all discussed in the manuscript). This means that the Reviewer's proposed procedure would not be implementable in the other 176 countries where we produce HDI estimates. Thus the proposed baseline is not valid since it cannot be implemented globally, and the explicit purpose of the paper is to produce global fine-resolution estimates of HDI, which currently do not exist.

Third, if the Reviewer's proposed approach were assumed to represent a valid comparison (which we do not believe is correct), then its implementation by the Reviewer is neither complete nor fair. A complete implementation would include both Brazil and Indonesia, in addition to Mexico, and a fair implementation would compare the Reviewer's R^2 measure to the ρ^2 measure we report in all of our tables, since the Reviewer's proposed procedure implicitly allows for arbitrary linear rescaling of predictions. Replicating the procedure proposed by the Reviewer results in a within-province performance of $R^2 = 0.50 / 0.26 / 0.37$ for predicting municipal HDI in Indonesia / Brazil / Mexico using Meta's RWI, which should be compared to our performance of $\rho^2 = 0.53 / 0.33 / 0.31$. Thus, implementing the Reviewer's proposed test verbatim, our approach outperforms Meta's RWI in two out of three countries with available data and on average across the three, which contrasts with the Reviewer's statement. Of course, the margin here is small, and thus we believe that a fair characterization would be to say that we achieve comparable performance using this test, similar to how we achieved comparable performance in the other comparisons to the literature we present in the main text. However, we refrain from introducing this comparison into the main article because it is not a valid solution to the downscaling

² If we were to use our satellite-based features but follow the Reviewer's procedure, i.e. train a model directly on the municipal HDI data from Mexico, we substantially outperform the R^2 that the Reviewer reports for the Chi et al. data. We obtain an out-of-sample within-province $R^2=0.54$ with image features compared to an out-of-sample within-province $R^2=0.36$ when trained using the municipal RWI values from Chi et al. This comparison suggests, based on the logic laid out in the Reviewer's comment, that our features substantially out-perform the baseline that the Reviewer proposes. However, we do not present this kind of argument in the paper because the test is not appropriate for the downscaling problem that the manuscript is solving.

problem and it is not generally applicable in real world data, as mentioned above, and so its presence would likely confuse readers.

Cognizant of the miscommunication that occurred on this point in the last round of revision, we also note that it is possible that the Reviewer intended a different interpretation of their above comment. The language of the comment leaves open the possibility that the Reviewer believes the wealth estimates in Chi et al. should be used directly as a wholesale substitute for our estimates of HDI (rather than a predictor of them). Our response to this alternative interpretation would be fourfold.

First, this approach would represent a complete reversion from using HDI, the more comprehensive measure of wellbeing that measures health, human capital and living standards, to a single wealth measure. This is an apples-to-oranges substitution that would not address the needs of many users who are concerned about broader measures of well-being (a distinction that we note in both the abstract and the first paragraph of the original manuscript).

Second, we acknowledge that for strict targeting contexts, there exist other useful datasets that could be used, which we did not explicitly mention in the manuscript previously. To address this, we have now adjusted our language around targeting in the main text to now state on line 301:

“Moreover, we acknowledge that there exist other datasets that could alternatively be used for wealth- or income-based targeting (e.g., Chi et al., 2022).”

Third, the Chi et al. estimates are not global, and only cover low and middle income countries, which limits the applications that they can be used for. For example, we learned through our communication with Dr. Blumenstock’s lab that some researchers there have adopted our estimates of HDI in their own work, instead of using the Chi et al. estimates of RWI that the lab produced, explicitly because they need full global coverage.

Fourth, the estimates in Chi et al. are not reproducible outside of Meta (formerly Facebook) because they directly rely on proprietary Facebook data, and unfortunately Meta appears unlikely to update the estimates in Chi et al. in the future or make the underlying data publicly available. The research arm within Meta that produced these estimates is being dismantled, with researchers, including Dr. Guanghua Chi himself, being reallocated to work that is more central to Meta’s core business. This was communicated to us directly by Dr. Chi. This severely limits the future practical applications of the Chi et al. data and suggests it would be unwise to invest in the further development and reliance on these estimates as a baseline measure of global wellbeing. In contrast, our estimates are fully reproducible using the publicly released code and data.

2. Lack of Validation for time-series estimates

During the first round of review, I asked the authors to provide proper validation of time-series estimates on HDI. However, the authors stated that “we use the satellite imagery to downscale annual first level administrative HDI estimates to higher spatial resolution, but do not predict or evaluate changes in the within-province distribution of HDI over time. (Response, p. 11)” and “This approach assumes that the spatial pattern of HDI within provinces does not change over time; relaxing this assumption gives similar results (Supplementary Information S3).“

Nevertheless, the manuscript proposes to release annual maps for 2012–2021. In practice, when this manuscript is published in Nature Communications, there are high possibilities for some readers to use these data for time-series comparisons and policy evaluation despite the authors’ guideline in the manuscript. Without any temporal validation (e.g., consistency across years, sensitivity to shocks, stability of rankings, calibration of year-to-year changes), this might create a substantial risk of misuse and misinterpretation. Given that the study’s primary objective is to characterize fine-scale spatial variation in HDI, what is the rationale for releasing annual maps for 2012–2021?

Our reply

We agree with the Reviewers' concern regarding potential misuse and misinterpretation, and have removed these time-series estimates from the manuscript. These time-series were not included in the original version of our manuscript and were only introduced in response to an explicit reviewer request (at Redacted). However, we have also had some concern about misinterpretation, even with explicit guidelines in the paper, and have thus removed the time-series estimates from the manuscript.

3. Methodological novelty

I raised the heavy reliance of this study’s model on the MOSAIK framework, which was previously published in Nature Communications (Rolf et al., 2021). However, the authors disagree with my concern with several methodological contributions they assert. The authors argue novelty in training and predicting on irregularly shaped and sized polygons to leverage coarse administrative labels. “Training and predicting on irregularly shaped and sized polygons, which is generally not possible using existing SIML approaches. This addresses a key limitation of the existing SIML literature -- the dearth of available training data -- by allowing models to be trained on coarse but plentiful administrative data. (response p. 22)”. Unfortunately, the similar ideas have been already proposed such as Han et al. (2020).

Other touted advances—training on fine-scale local variation, using a fixed-effects specification, and expanding MOSAIKS features from ~Y1% to near-global coverage (response p. 22)—appear to be incremental refinements and engineering scale-up rather than a substantive

methodological contribution. The manuscript's empirical demonstrations do not, in my view, establish a new methodological paradigm beyond what has been introduced in the literature.

Our reply:

We strongly disagree with the reviewer's assertion that the analysis by Han et al. supersedes the contribution of this study. Respectfully, the Reviewer's statement above sounds as if the two studies make the same contribution, but this is only because the statement does not fairly represent the content and contribution of both studies.

The contribution of our manuscript is to address the problem of downscaling administrative data that originates from irregular polygons. We believe this is clear in the manuscript, since it is described in the title, abstract, introduction, it is the focus of Figures 1-3, and it is repeatedly tested throughout the text. However, the Reviewer's statement above is removed from context and only describes the training based on irregular polygons as if that were the only point of the paper, ignoring the difficult-but-central downscaling. This incomplete portrayal of our contribution sounds as if it aligns with the contribution of Han et al., since Han et al. focus on training a model using polygons and stop there.

Han et al. neither develop a downscaling procedure nor present how one might work. Han et al. do not produce downscaled estimates of any outcome and they do not present any results validating that their model could do so accurately. In a single paragraph of the discussion of their manuscript, Han et al. speculate in words that downscaling might be possible using their approach, stating that "[t]he potential to utilize such micro-level inference is magnificent," but they do not test this conjecture and they provide no theory nor evidence to support it. As part of this discussion, Han et al. present one non-quantitative graphical illustration of this conjecture (pasted below), but it is not even clear from the associated text whether this conceptual figure depicts actual model results or is a qualitative illustration (e.g. there are not quantitative values associated with the color bar):

[Figure Redacted]

Figure 6 from Han et al.

In contrast, nearly our entire manuscript is dedicated to rigorously testing whether our approach to downscaling is viable in many real world data sets. Thus, it seems inaccurate to suggest that the methodological contribution of Han et al. precludes our contribution. Rather, it would be accurate to state that the paper by Han et al. ends with a discussion speculating that it might be possible to do what we have done in our analysis.

Nonetheless, to address the Reviewer’s concern, we now clarify the relationship between Han et al. and our work by adding a reference to Han et al. at the appropriate point in the Introduction, where we describe learning an outcome from data aggregated over irregular polygons. The text now states:

“Our approach builds on recent advances in machine-learning (Rolf et al., 2021; Han et al., 2020) to develop a general method that learns the relationship between imagery and an outcome of interest (here, HDI) using data from any set of political boundaries.”

References

Chi et al. (2022). Microestimates of wealth for all low-and middle-income countries. Proceedings of the National Academy of Sciences 119, e2113658119

Han, S. et al. (2020). Lightweight and Robust Representation of Economic Scales from Satellite Imagery. Proceedings of the AAAI Conference on Artificial Intelligence, 34(01), 428-436.
<https://doi.org/10.1609/aaai.v34i01.5379>

Rolf, E. et al. (2021). A generalizable and accessible approach to machine learning with global satellite imagery. Nature Communications, 12(1), 1-11.

(Remarks on code availability)

Reviewer #3 (Remarks to the Author)

Thank you to the authors for replying to my last set of questions. I have just two follow-up questions which I think can be handled relatively easily.

Our reply:

We thank the reviewer for their thoughtful comments, as they have significantly improved the quality of the manuscript.

1. I am still struggling a bit with the fact that the ranking of predicted values within a country will be biased against poor municipalities in non-poor regions and vice versa. If I am not mistaken, this systematic bias would not arise if the predictions were made in levels instead of deviations from mean values. I understand such predictions would perform worse in terms of R^2 , but perhaps the exclusion/inclusion errors for targeting would be better? It should be easy for the authors to reject this hypothesis with their Mexican, Brazilian, and Indonesian data. If the AUC indeed is worse under this approach, I don't think any further changes are needed to the manuscript on this topic.

Our reply:

We thank the Reviewer for raising this interesting suggestion. If we understand correctly, the Reviewer is hypothesizing that targeting performance will improve if we use a model that is trained on and makes predictions for HDI in levels ("Province level" in Table S1), rather than using a model that is trained on and makes predictions for HDI anomalies from country means ("Within-country" in Table S1). In the manuscript, we show that the models trained "within-country" have higher predictive performance, as measured by R^2 than models trained using the provincial HDI levels directly, as the Reviewer refers to.

Here, we test the Reviewer's hypothesis that using a model that is trained on and makes predictions for HDI levels will have better targeting performance, evaluated using AUC, than the "within-country" model. Replicating the targeting analysis for Mexico depicted in Figure 5, we find that the "within-country" model has higher targeting performance (Municipal AUC = 0.86) than the model that predicts HDI in levels (Municipal AUC = 0.80). This rejects the hypothesis, and thus we do not make any related changes to the manuscript, as recommended by the Reviewer.

2. Thanks to the authors for clarifying when the proposed data can be useful for targeting. I agree that the application illustrates how fine-resolution HDI estimates can improve targeting of policies that would otherwise use coarser provincial measures. Yet I am still unaware of any cases where coarser HDI estimates were used for targeting. It would be helpful if the authors could cite a handful of such cases to document that this is tackling a widespread policy issue.

Even if such cases do not exist, I think the fine-resolution HDI estimates have tremendous value, but perhaps more in terms of providing vital data for the research community and increasing our understanding of human wellbeing globally, but less for targeting purposes.(Remarks on code availability)

Our reply:

HDI has been used at the national level to set prices and prioritize development aid. For example, the pharmaceutical company Merck has given 90% discounts on some drugs to countries with 'low' HDI, and Ireland has focused development aid towards countries with 'low' HDI (Wolff et al., 2011, ref. (6) in the manuscript). We are not aware, however, of any policy use-cases where HDI-based targeting has occurred below the national level. However, we do not find this to be especially surprising since only provincial HDI data would have been available to-date. We hope that the finer-resolution HDI estimates provided here can enable and encourage targeting at fine spatial scales using HDI, as well as provide vital data to researchers and policymakers. Our targeting exercise is intended to illustrate the merits of finer-resolution data, rather than to imitate any specific real world program. While the section header "Illustrative application: targeting policy in Mexico" was intended to convey this idea, we have clarified the language used to describe this experiment.

The first sentence of the section is now (line 281):

*"To explore how our new HDI measures could improve the efficiency of development policies, we conduct an **illustrative** targeting exercise within Mexico."*

The language in the final paragraph of Results Section 5 (starting line 296) now reads:

*"This application illustrates, **in a hypothetical scenario**, how fine-resolution HDI estimates **could** improve targeting of resources, even though these fine-resolution estimates are imperfect. Users of these fine-resolution estimates should consider their specific policy context to determine whether these fine-resolution estimates, which explain 29% of the within-province HDI variation in Mexico, 20% in Brazil, and 53% in Indonesia, provide sufficient additional information to be useful in their setting. **Moreover, we acknowledge that there exist other datasets that could alternatively be used for wealth- or income-based targeting** (e.g., Chi et al., 2022). Still, these estimates may be particularly valuable in data poor settings, where traditional data sources such as surveys are less available and reliable (Burke et al., 2021); only about half of the world's poorest countries have conducted a census in the last decade (Yeh et al., 2020; Aiken et al., 2023). These estimates may also be useful in cross-country settings where a consistent metric of welfare is desired."*

Referenced material:

Wolff, Hendrik, Howard Chong, and Maximilian Auffhammer. "Classification, detection and consequences of data error: evidence from the human development index." *The Economic Journal* 121.553 (2011): 843-870.

REVIEWERS' COMMENTS

Reviewer #1 (Remarks to the Author):

I appreciate the authors' considerable efforts in addressing my comments and for providing well-prepared and detailed responses. The exchange during the review process has been very insightful, and I have gained a clearer understanding of the authors' objectives and interpretations within the manuscript. Their thoughtful replies have also clarified several points of potential misunderstanding on my part. Nonetheless, some points of concern remain, which I believe arise from differences in interpretation and perspective between the authors and myself. I raised three concerns: low model performance, lack of validation for time-series estimates, and methodological novelty. I appreciate that the authors have addressed the second concern by removing the time-series estimates from the manuscript, which might be a difficult decision. On the first and third concerns, I can partly agree with the authors' position. I acknowledge that this is largely a difference in perspective rather than a factual dispute. I believe this issue from the first and last concerns could be addressed constructively in the manuscript itself. I recommend that the authors explicitly describe the potential limitations and concerns in the Discussion section.

1a. Low Model Performance (general performance)

Authors' responses:

- "the assertion that novel research in this domain is only publishable if $R^2 \geq 0.5$ is both arbitrary and inaccurate." (Second Responses, p. 2)
- "The Reviewer incorrectly presents the "within-province performance" R^2 metric, which is not the most relevant metric for evaluating within-country HDI distributions, particularly when comparing to prior literature that has been evaluated using a within-country performance metric." (Second Responses, p.2)
- "Finally, the Reviewer's suggestion that the model performance presented is of too low quality to be of interest or use to policy-making applications is not correct." (Second Responses, p.2)

Reviewer's comments:

Firstly, my intention was not to suggest an official or rigid threshold (e.g., $R^2 \geq 0.5$) as a requirement for publication. Rather, my point is that within today's mature SIML research landscape, performance metrics at or above this level have become commonplace, and are thus a natural reference point for evaluating new contributions.

The examples cited by the authors (Jean et al., 2016; Rolf et al., 2021; Yeh et al., 2020) are indeed important milestones in the SIML domain, despite their modest performance metrics. However, it is important to recognize that these studies were pioneering works, published at a time when the combination of satellite imagery and machine learning was still emerging. Their significance lay primarily in opening new research directions rather than in achieving high predictive accuracy per se. In recent literature, the performance achieved is typically higher—for example, Ahn et al. (2023) reported Spearman's $\rho = 0.77$ for grid-level economic development estimates and $R^2 = 0.84$ at a district level (comparable to Figure 2.C of the present manuscript).

Similarly, Zheng et al. (2025) reported R^2 values between 0.62–0.83 for within-country wealth index predictions across multiple African nations.

Given the maturity the SIML field has now achieved, expectations for model performance and robustness are naturally higher, especially in high-impact venues. Novel contributions today might reasonably be held to more stringent performance and validation standards than the early works from the 2010s or early 2020s.

Secondly, I acknowledge that my earlier comparison between the authors' within-province performance metric and the within-country performance metrics used in prior literature may have been inappropriate, given the differences in spatial resolution and study objectives. I accept that direct comparison across these metrics can be misleading.

However, in this case, the manuscript's stated novelty lies in its downscaling approach and its ability to explain fine-resolution, "within-province variation" in HDI. If this level of granularity represents the core methodological contribution, then it is reasonable to expect the within-province performance itself to be sufficiently strong to justify publication in a high-impact journal such as Nature Communications. While earlier studies evaluated their models using "within-country metrics" because that matched their research scope, they achieved performance levels that were competitive for their chosen resolution and metric at the time. Similarly, if the focus here is at the "within-province level", then comparable competitiveness should be expected within that resolution context—both in terms of predictive accuracy and demonstration of robustness.

My biggest concern still lies in the cases such as Brazil with 0.2 within-province R^2 although the average for countries is 0.34 (within-province) or 0.51 (within-country). This means the model might explain only about 20% of the variance at this resolution in a certain country. I am concerned that in policy contexts, such low performance might limit practical usefulness or lead to misinterpretation if the uncertainty is not adequately communicated.

Lastly, I did not mention that the model performance is too low to be of interest to policy-making applications. I mentioned it is "too low to support the main claims and intended uses." (Second review comment 1) In fact, this research topic is highly pertinent, and I agree it could be of significant interest to institutions such as UNDP and other stakeholders. The manuscript clearly addresses a socially important issue, and the potential impact in guiding development policy is substantial. However, while institutional interest (e.g., UNDP's collaboration) is encouraging for dissemination and uptake, it does not, in itself, validate the model's accuracy or suitability for the stated applications without further performance improvement or careful qualification of limitations.

In summary for the first concern (low performance), in the context of recent SIML studies, I would generally expect higher performance than what is reported in the manuscript. While the 0.5 threshold I have referred to in earlier comments is not an official requirement, current literature suggests that this is not an unusually high bar to meet. That said, I acknowledge the authors' perspective that performance at the "within-province" resolution can still be acceptable depending on research aims and context.

On balance, I believe the level of performance reported here is acceptable, provided that the Discussion section explicitly addresses potential issues of interpretation that may arise from these accuracy levels. In particular, I recommend that the manuscript comment on the practical risks and limitations in policy-relevant cases such as Brazil, where the within-province R^2 is approximately 0.20.

References

Ahn, D., Yang, J., Cha, M. et al. A human-machine collaborative approach measures economic development using satellite imagery. *Nat Commun* 14, 6811 (2023).

<https://doi.org/10.1038/s41467-023-42122-8>

Zheng, Z., Burke, M., Ermon, S., Kilic, T., Lee, R., Lobell, D.B., Newhouse, D. and Wu, T., 2025. Dynamic, High-Resolution Wealth Measurement in Data-Scarce Environments. World Bank.

1b. Low Model Performance (comparison with Chi et al.)

Authors' responses:

- "the Reviewer's proposed comparison is to regress municipal HDI in Mexico directly on the RWI constructed by Chi et al., thereby training and evaluating a model on the municipal data. This, however, is incomparable with all of our estimates because the Reviewer's procedure skips the difficult-but-essential downscaling part of the problem. [...] From the perspective of the problem that the manuscript is solving, the Reviewer's proposed test "cheats" by training the model directly on the unobservable municipal data." (Second Responses, p. 4)

Reviewer's comments:

I appreciate the authors' detailed response to my suggestion regarding comparison with Chi et al.'s RWI in Mexico. I understand that from the authors' methodological perspective, using RWI to estimate municipal-level HDI may be considered to "cheat" the difficult but essential downscaling step, and is therefore not directly comparable to the approach presented in the manuscript.

However, I would like the authors to also consider this issue from the perspective of an end user, such as a policymaker in Mexico. In a real policy setting, the priority is often to use the most accurate and validated local-level statistic available — whether produced by downscaling or more direct regression — in order to identify disadvantaged regions and allocate resources effectively. For example, RWI is available at the grid level and can serve as a strong predictor of municipal HDI without requiring downscaling in contexts where municipal data are already available.

In countries such as Mexico, Brazil, and Indonesia, where municipal-level HDI data are accessible, it is entirely feasible and valid for end users to use direct regression-based approaches or other high-correlation proxies without addressing the downscaling problem. In contrast, in countries without municipal-level HDI, the quality and trustworthiness of downscaled estimates become far more important. In my view, downscaled HDI in such countries should be considered for policymaking only if adequate validation is presented, comparable to the validation done for Mexico, Brazil, and Indonesia.

I recognize that this represents a difference in perspective: the authors argue that validation in three countries is sufficient to infer reliability in others without municipal data, whereas my position is that validation should be demonstrated country-by-country before such data are used for policy purposes in a certain country. I believe that this concern needs to be mentioned at somewhere in the Discussion, so that readers understand the potential limitations.

Authors' responses:

- "Reviewer's proposed procedure would not be implementable in the other 176 countries where we produce HDI estimates. Thus the proposed baseline is not valid since it cannot be implemented globally, and the explicit purpose of the paper is to produce global fine-resolution estimates of HDI, which currently do not exist." (Second Responses, p. 5)

Reviewer's comments:

I acknowledge the authors' point that my proposed RWI-based procedure would not be implementable in the other 176 countries for which they produce HDI estimates, whereas their downscaling approach enables the creation of global, fine-resolution HDI estimates — a product that, to my knowledge, does not currently exist.

However, I would like to underline that this represents a difference in perspective rather than a disagreement over technical details. From the authors' viewpoint, producing global coverage in itself is a sufficient justification for the value of the dataset, even if country-specific validation is limited to a subset of nations. From my perspective, global fine-resolution HDI estimates are of limited policy utility in any given country unless they are accompanied by adequate validation for that specific national context.

Authors' responses:

- "Third, if the Reviewer's proposed approach were assumed to represent a valid comparison (which we do not believe is correct), then its implementation by the Reviewer is neither complete nor fair. [...] Replicating the procedure proposed by the Reviewer results in a within-province performance of $R^2 = 0.50 / 0.26 / 0.37$ for predicting municipal HDI in Indonesia / Brazil / Mexico using Meta's RWI" (Second Responses, p. 5)

Reviewer's comments:

I appreciate the authors' clarification regarding my earlier suggestion and the replication of the procedure I proposed. As noted, my second-round R^2 results were based solely on municipal-level HDI estimates in Mexico, as that was the only case for which I had access to the necessary data. The authors' replication demonstrates that, in Mexico, R^2 using Meta's RWI is higher ($R^2 = 0.31$) than with the authors' approach. However, when considering the average within-province R^2 values across the three countries assessed — 0.38 for RWI compared to 0.39 for the authors' method — I acknowledge that, on balance, the authors' approach yields slightly better average performance than this potential baseline. Therefore, I agree that, in the context of these three countries, the presented method performs at least as well or marginally better than the proposed RWI-based alternative.

Moreover, I agree with the authors that using RWI for HDI estimation can be considered an "apples-to-oranges substitution" (Second Responses, p.6), since it bypasses the downscaling

process central to the manuscript's methodology. At the same time, I would note that all SIML-derived HDI estimates — regardless of modeling approach — inherently represent “apples-to-oranges” substitutions when compared with official HDI statistics constructed from household survey data. Both RWI-based and other SIML-based methods are proxies, and performance differences are therefore best understood as degrees of alignment to survey-based ground truth, not as direct measurements.

Authors' responses:

- “Third, the Chi et al. estimates are not global, and only cover low and middle income countries, which limits the applications that they can be used for.” (Second responses, p. 6)

Reviewer's comments:

I respectfully disagree with the authors' view that the Chi et al. estimates have limited utility because they are not global and cover only low- and middle-income countries. As noted in much of the SIML literature, satellite-based socioeconomic estimates are often most valuable precisely in low- and middle-income countries, where conventional statistics are sparse or outdated. In such contexts, these proxy measures can fill critical data gaps and support evidence-based decision-making.

Moreover, as I mentioned in my first-round review, there are municipal-level data for HDI-related indicators (e.g., literacy rate, life expectancy) in several high-income countries, such as South Korea. Other countries in the Global North typically have both the resources and institutional capacity to conduct municipal- or district-level surveys for HDI or component statistics. As such, the added value of satellite-based downscaled HDI estimates may be greatest in settings where official local-level data do not already exist.

2. Lack of Validation for time-series estimates

Thank the authors again for removing the time-series estimation component from the manuscript. I recognize that this may have been a difficult decision, given the potential appeal of time-series data. I believe this revision improves the clarity and focus of the paper in its current form. I also hope that the authors will consider developing and validating time-series HDI estimates in future studies, as such work could make a valuable contribution to the field once robust validation methods are available.

3. Methodological novelty.

Authors' responses:

- “The Reviewer's statement above is removed from context and only describes the training based on irregular polygons as if that were the only point of the paper, ignoring the difficult-but-central downscaling.” (Second response, p. 8)

Reviewer's comments:

The authors state in their response that my earlier comment “is removed from context” and portrays training on irregular polygons as the only point of the paper, “ignoring the difficult-but-central downscaling.” (Second Response, p. 8) I respectfully disagree with this

characterization. In the main manuscript, the authors themselves clearly present the ability to utilize irregularly shaped administrative data as a key methodological advance:

“This limitation is partly due to the design of modern SIML methods, since large quantities of administrative data are available, but existing systems are generally not designed to make use of them.” (lines 47 – 48)

“This restriction has caused prior studies to rely on coarse approximations for linking irregularly shaped labels to corresponding imagery, for example, by averaging polygon labels that overlap with the square image” (lines 54 – 56)

In their First Response (p. 22), the authors again emphasize:

“Training and predicting on irregularly shaped and sized polygons, which is generally not possible using existing SIML approaches. This addresses a key limitation of the existing SIML literature -- the dearth of available training data -- by allowing models to be trained on coarse but plentiful administrative data.” (First response p. 22)

Across these statements, the irregular-polygon approach is positioned as a methodological innovation in its own right, not solely as a supporting detail within the downscaling framework. My earlier comments on novelty were therefore based on the authors' own framing of this capability as a central contribution, alongside the downscaling methodology.

After reading the Second Response, I acknowledge that Han et al.'s aim and the authors' aim differ: the authors seek “to address the problem of downscaling administrative data that originates from irregular polygons” (Second Response, p. 8), while Han et al. aim to “obtain essential spatial representation for any given district from high-resolution satellite imagery based on deep neural networks” (Han et al., 2020, abstract). Nonetheless, Han et al. appear to address the challenge of training (at least) on irregularly shaped and sized polygons—not necessarily within a downscaling approach—through “learning the critical spatial characteristics of arbitrary size areas and representing such characteristics in a fixed-length vector.” (Han et al., abstract)

Finally, while the authors also mentioned other advances—training on fine-scale local variation, use of a fixed-effects specification, and expansion of MOSAIKS features from ~Y1% to near-global coverage (First Response, p. 22)—I still regard these as incremental refinements and engineering scale-up, rather than major methodological breakthroughs.

Again, I acknowledge this difference in view remains primarily a matter of perspective. At this stage, I believe further back-and-forth would likely restate the same arguments, and I therefore leave it to the editor or other reviewers to determine whether the manuscript meets the journal's standards for publication.

Han, S. et al. (2020). Lightweight and Robust Representation of Economic Scales from Satellite Imagery. *Proceedings of the AAAI Conference on Artificial Intelligence*, 34(01), 428-436.
<https://doi.org/10.1609/aaai.v34i01.5379>

We thank the reviewer for their time, thorough engagement, and valuable feedback and suggestions.

Based on these comments, we have added a paragraph to the Discussion that explicitly discusses these HDI data from a policymakers perspective and acknowledges that policymakers face tradeoffs in electing to use new sources of data.

“Policymakers will face a number of complex considerations before using these data in practice. Broadly, policymakers can benefit from algorithmically-informed decisions that are transparent and explainable (56) and there may be costs to using data for decision-making that is fundamentally harder to interpret and explain than alternative decisions made using traditional survey data. In addition, policymakers will have to decide whether and how it is appropriate to use a global dataset that we have only been able to directly validate in three countries. Third, policymakers may benefit from technical evaluations that interpret the performance here so that they can assess the marginal benefits of using these data for their specific use-case, recognizing that reasonable people may disagree about whether the data provided here achieve high enough performance to merit its use for specific applications. Future work should investigate improvements in remote sensing model interpretability that can support transparent policy-making and aid in understanding potential sources of systematic bias (55,57,58).”

We have also added more strength to our statement about the continuing need for survey data in Results Section 2.

“We emphasize that these data are not a complete substitute for survey-based estimates and that our model could not be trained or evaluated without ground-truth surveys.”